# Protein degradation by human 20S proteasomes elucidates the interplay between peptide hydrolysis and splicing

Wai Tuck Soh[1,10], Hanna P. Roetschke[1,2,3,10], John A. Cormican[1,10], Bei Fang Teo[2,3,4], Nyet Cheng Chiam[1], Monika Raabe[5], Ralf Pflanz[5], Fabian Henneberg[6], Stefan Becker[7], Ashwin Chari[8], Haiyan Liu[4], Henning Urlaub[5,9], Juliane Liepe[1,11] ✉ & Michele Mishto[2,3,11] ✉

If and how proteasomes catalyze not only peptide hydrolysis but also peptide splicing is an open question that has divided the scientific community. The debate has so far been based on immunopeptidomics, in vitro digestions of synthetic polypeptides as well as ex vivo and in vivo experiments, which could only indirectly describe proteasome-catalyzed peptide splicing of full-length proteins. Here we develop a workflow—and cognate software - to analyze proteasome-generated non-spliced and spliced peptides produced from entire proteins and apply it to in vitro digestions of 15 proteins, including well-known intrinsically disordered proteins such as human tau and α-Synuclein. The results confirm that 20S proteasomes produce a sizeable variety of *cis*-spliced peptides, whereas *trans*-spliced peptides are a minority. Both peptide hydrolysis and splicing produce peptides with well-defined characteristics, which hint toward an intricate regulation of both catalytic activities. At protein level, both non-spliced and spliced peptides are not randomly localized within protein sequences, but rather concentrated in hotspots of peptide products, in part driven by protein sequence motifs and proteasomal preferences. At sequence level, the different peptide sequence preference of peptide hydrolysis and peptide splicing suggests a competition between the two catalytic activities of 20S proteasomes during protein degradation.

In eukaryotic cells, most cytoplasmic proteins — e.g., transcription factors, obsolete, damaged or wrongly transcribed proteins — are processed by proteasomes and changes in the proteolytic activity of these proteases have been associated with many pathological conditions. Proteasomes are the core of the Ubiquitin Proteasome System (UPS). The most active proteasome isoforms are 26S/30S proteasomes, wherein the 20S proteasome core is bound to one or two 19S multi-subunit complexes, respectively, and degrade poly-ubiquitylated proteins[1]. Proteins are poly-ubiquitylated through a cascade of E1, E2 and E3 enzymes, which activate, conjugate and transfer multiple ubiquitin moieties to protein substrates for proteasomal degradation[2]. Ubiquitinated proteins can be processed by both 20S and 26S proteasomes[3]. Another well-studied regulatory complex, PA28αβ, binds the 20S proteasome core and can degrade non-polyubiquitinated proteins[4,5]. The 20S proteasome can be expressed in different isoforms, depending on the catalytic subunit content, e.g., standard-, immuno- and thymo-proteasomes[6]. The 20S proteasome isoforms have different preference for peptide sequences and substrates, which can affect various metabolic pathways, response to stimuli and Human Leucocyte Antigen class I (HLA-I) immunopeptidomes[4,7–16]. The latter are pools of peptides, which are mainly generated by proteasomes[1,17], that are presented by HLA-I

molecules to CD8[+] T cells and recognized by T Cell Receptors (TCRs). The 20S proteasomes are also active as such, both in the intracellular and extracellular space[18–21]. They preferentially process intrinsically disordered proteins (IDPs), which can contain large unstructured segments or even completely lack a defined tertiary structure in their native state, although they can adopt a fixed tertiary structure after binding to other macromolecules[22]. The transient lack of an ordered three-dimensional structure allows IDPs to dynamically bind to diverse interaction partners, accelerate interactions and chemical reactions between bound partners and thus influence many biological processes[23,24]. IDPs have been estimated to represent up to 30% of the intracellular proteome[23,25]. Aggregation of IDPs−e.g., human α-Synuclein and tau−into insoluble deposits is the hallmark of neurodegenerative diseases such as Parkinson's and Alzheimer's disease. 20S proteasomes can efficiently degrade these two proteins[26–29] as well as many other IDPs in absence of poly-ubiquitination of the protein targets[25]. The exact mechanism of IDP selection by 20S proteasomes – in absence of the entire UPS – is still unclear, although a family of proteins named catalytic core regulators has been shown to regulate 20S proteasome activity through an allosteric modification of the proteasome gates without affecting its catalytic subunits[30]. The characteristics of the disordered region have been hypothesized to play a role in the substrate selection by 20S proteasomes[31,32]. IDPs with more protein binding partners and post-translational modifications (PTMs) seem to be preferentially processed by 20S proteasomes[25]. Peptide fragments of IDPs−e.g., osteopontin−produced by 20S proteasomes can have regulatory activities in cells[33,34].

Proteasomes can cleave proteins and release the peptides produced by peptide hydrolysis (Fig. 1a), as well as ligate two non-contiguous peptide fragments (i.e., splice-reactants) of either the same molecule (cis-spliced peptides; Fig. 1b), or two distinct molecules of the same protein (homologous trans-spliced peptides; Fig. 1c), or

molecules from two distinct proteins (heterologous trans-spliced peptides; Fig. 1d) in a process called Proteasome-Catalyzed Peptide Splicing (PCPS)[6]. The mere existence of PCPS has been questioned by part of the scientific community since its discovery in 2004[35,36], with the assumption that even if PCPS could occur it would be very inefficient[37]. In particular, the biochemical process of transpeptidation, which was originally proposed for PCPS by Vigneron and colleagues[38] and confirmed by others[39,40], has been recently disputed[37,41], although it has been also described for other proteases[42,43]. The existence and presentation of proteasome-generated cis-spliced peptides is supported not only by biochemical but also immunological evidence. Indeed, these noncanonical peptides can target CD8[+] T cell responses against bacterial antigens in vivo, in a mouse model of Listeria monocytogenes infection, wherein the potential cross-recognition by CD8[+] T cells originally primed against canonical non-spliced peptides was excluded for two specific cis-spliced epitopes[44]. For other cis-spliced epitopes derived from Listeria monocytogenes and human immunodeficiency virus, a CD8[+] T cell response can be stimulated through cross-recognition ex vivo[45,46]. Self cis-spliced epitopes associated with type 1 diabetes (T1D) are recognized by CD8[+] T cells in the pancreas of T1D patients[47]. Potential cases of T1D-associated viral-human epitope mimicry are possible[48], although cis-spliced peptides may not play a special role in CD8[+] T cell tolerance[49]. Cis-spliced peptides can carry cancer-specific mutations[50,51], and in the peripheral blood of melanoma patients, CD8[+] T cells and cytotoxic T lymphocytes (CTLs) recognize melanoma-associated cis-spliced epitopes[52,53]. The potential relevance of cis-spliced epitopes as therapeutic targets in cancer was clear since their first discovery because CTLs and tumor-infiltrating lymphocytes (TILs) specific for cancer-associated cis-spliced epitopes were isolated and used for the epitope validation[35,38,54–56]. In particular, in many of these studies, specific CTLs were used in controlled in vitro conditions that limited the confounding factor of TCR

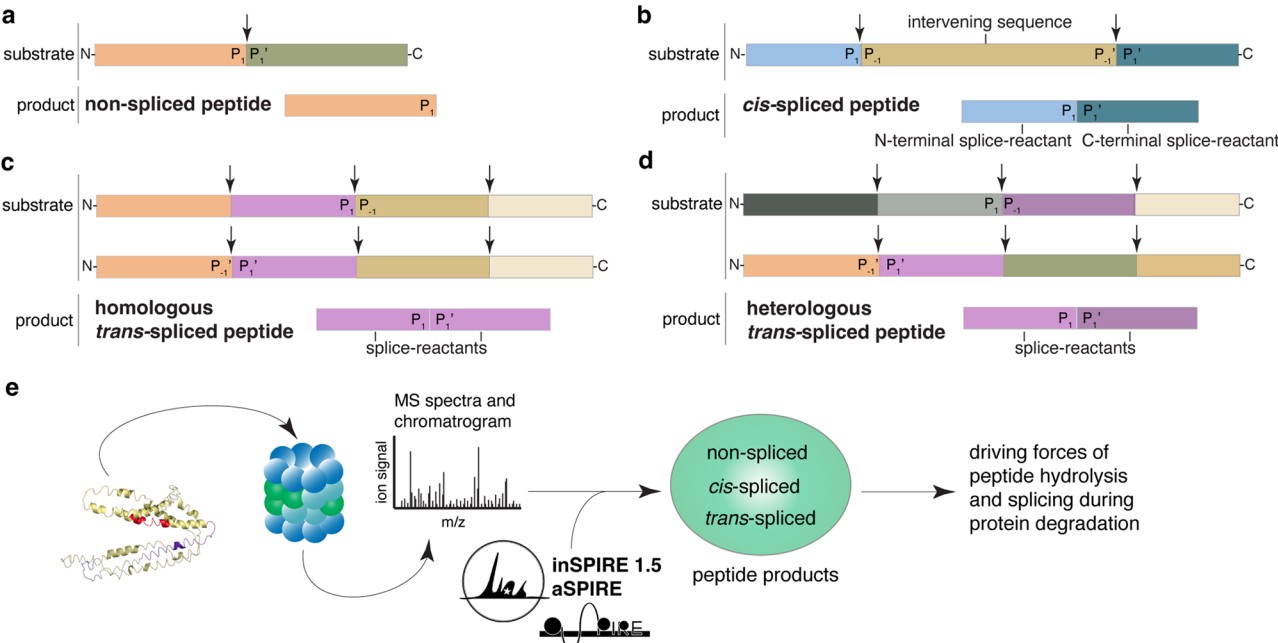

**Fig. 1 | Proteasome-generated non-spliced and spliced peptides.** Proteasomes generate: (**a**) non-spliced peptides via peptide hydrolysis, (**b–d**) spliced peptides through ligation of two non-contiguous splice-reactants either derived from the same molecule (cis-spliced peptides, **b**) or from two distinct molecules of the same protein (homologous trans-spliced peptides, **c**) or two distinct proteins (heterologous trans-spliced peptides, **d**). In (**b**), peptide fragment ligation can occur in forward order, i.e., following the orientation from N- to C-terminus of the parental protein (forward cis-peptide splicing) or in reverse order (reverse cis-peptide splicing). The two fragments, ligated by PCPS, are named splice-reactants, and their junction is named splice-site. The C-terminus of the first (N-terminal) splice-reactant is named P$_1$, while the N-terminus of the second (C-terminal) splice-reactant is named P$_1$'. The sequence segment between two splice-reactants is called intervening sequence. Arrows represent the substrate cleavage/splice sites used by proteasome catalytic Thr1. **e** Graphical representation of the experimental design of the study. The protein structure was adopted from[33].

degeneracy thereby pinpointing spliced epitope production and presentation[38,44,52,54,56]. As further direct evidence supporting the potential translational relevance of *cis*-spliced epitopes, a melanoma patient with metastasis has been cured through adoptive T cell therapy using an autologous TIL clone, which was proved, in a later study, to be specific for a *cis*-spliced epitope derived from a melanoma-associated antigen[57,58]. In contrast, the evidence supporting the biological relevance of *trans*-spliced peptides is scarcer. Homologous *trans*-spliced peptides have been identified in both in vitro experiments with purified proteasomes[14,39,40,50,59,60] as well as *in cellula*[54]. Heterologous *trans*-spliced peptides have been described so far only in a study on HLA-I immunopeptidomes[61]. Their immunological relevance in the context of HLA-I antigen presentation is still an enigma. In contrast, heterologous *trans*-spliced peptides called Hybrid Insulin Peptides, which are produced by cathepsins rather than proteasomes[62,63], have been identified and extensively investigated as T1D-associated epitopes presented by HLA-II molecules[64,65].

The systematic identification of spliced peptides in HLA-I immunopeptidomes started in 2016, in a seminal, although controversial, study that led to the following development of many methods for *cis*-spliced peptide identification[66]. Depending on advance bioinformatics methods based on mass spectrometry (MS) and applied to HLA-I immunopeptidome datasets, the estimations of the frequency of these noncanonical peptides dramatically diverged between 0.1% to over 30% of the HLA-I immunopeptidome sequence variety[53,61,66–72], and many of the spliced peptides identified in the original study[66] have not been confirmed. Many factors could impinge upon the identification of non-spliced and spliced peptides in HLA-I immunopeptidomes[73–76]. Among them, the huge size of the theoretical *cis*-spliced peptide database derived from the human proteome poses statistical issues, which we speculate are common to other noncanonical peptides such as cryptic peptides and PTM-tagged peptides. Another issue is the potential multi-origin of a given peptide sequence, since the theoretical peptide sequence variety derived from the human proteome renders difficult the assignment of a given peptide sequence to a specific peptide type rather than another. These issues are barely present in the MS-based analysis of in vitro digestions of synthetic polypeptides by purified proteasomes because the reference database size of spliced peptides is relatively small and cryptic peptide origin can be excluded since the substrate sequence is known. In this type of experiment, proteasomes produce a similar frequency of non-spliced, *cis*- and *trans*-spliced peptides[59,60], although non-spliced peptides are on average produced in larger amount[50,77]. Correspondence between these in vitro experiments with synthetic polypeptides and *in cellula* and in vivo experiments has been demonstrated in various immunological studies[10,11,38,44,52,55,56,58,78–83]. Nonetheless, although the in vitro processing of synthetic polypeptides by proteasomes can be informative in dissecting the details of peptide sequence preferences of proteasomes and the generation of specific antigenic peptides, it might neglect proteins' transport dynamics, conformation, and steric effects within the proteasome chamber[7,84,85].

In this work, we investigate PCPS during the in vitro digestion of a collection of 27 proteins and identify a pool of proteins that is efficiently processed by purified human 20S proteasomes. As for the in vitro processing of synthetic polypeptides, also the in vitro digestion of entire proteins by purified human 20S proteasomes avoids the confounding factor of potential multiple origins of noncanonical peptides, because they could only derive from PCPS. However, proteins are longer than polypeptides, which is linked to an increased search space size and increases the statistical challenges during MS2 spectrum assignment. To tackle this problem, we identify peptide products by developing an MS-based method called inSPIRE 1.5 (IN silico Spectra Predictor Informed RE-scoring)[86], coupled to aSPIRE (Abundance of Spliced and non-spliced Peptides Incorporating Relative Quantification)[87], thereby allowing peptide label-free quantification (Fig. 1e). According to the results, PCPS is a biochemically unique and tightly regulated process, and in this study, we shed some light into its driving factors.

## Results

### In vitro degradation of proteins by 20S proteasomes

We preliminarily investigated whether a collection of 27 purified proteins could be processed by 20S proteasomes in vitro, and could therefore be used as substrate models for our study. The panel of purified proteins included a wide range of sources including bacteria, yeast, frog (*Xenopus laevis*), chicken, mouse and human (Supplementary Table 1, Supplementary Data 1). We used 20S proteasomes purified from different human sources, as well as various experimental conditions and buffers that were routinely used in our laboratories (without any proteasome activator such as Sodium Dodecyl Sulfate, SDS). Through Western blot and Coomassie blue staining of SDS-PAGE blots, we monitored the protein degradation process. If the latter was not conclusive, we then evaluated the number of identified peptide products by MS and considered a protein as efficiently degraded when there were more than 300 peptide products detected in the overall digestion by 20S proteasomes in vitro. Thereby, we identified 15 proteins that were efficiently processed by 20S proteasomes, i.e., annexin A1, α-Synuclein, calmodulin (CaM), Ffh, H2A, H2B, H3, H4, IF2, IL-37b, LEDGF, RF1, tau, Ube2K and Ube2S (Fig. 2a, Supplementary Fig. 1).

To understand the difference between degraded and non-degraded proteins by 20S proteasomes, we calculated their degree of intrinsically disordered characteristics using SLIDER[88] and IUPred3[89] predictors. We found that the 15 degraded proteins had a significantly greater number of long-disorder segments and disordered regions than the non-degraded proteins (Fig. 2b). Among them, some were archetypes of IDPs, such as human tau and α-Synuclein, others only contained some predicted disordered regions and could have a flexible form in solution[27,90,91]. Together, these suggest that the proteins degraded by 20S proteasomes have, on average, a greater predicted structural disorder than non-degraded proteins in solution and in absence of interaction partners. A similar pattern was also described in an independent analysis of nuclear proteins processed by mouse 20S proteasomes[25].

Since proteasomes are the main proteases generating HLA-I-presented antigenic peptides, the physiological activity of 20S proteasomes in degrading a broad range of IDPs could be detectable in the form of an overrepresentation of IDP-derived peptides in HLA-I immunopeptidomes. We investigated this hypothesis in the IEDB antigenic peptide database considering HLA-A and HLA-B separately. Antigens represented in both pools of HLA-I complexes had a significantly greater predicted number of long-disorder segments than the group of proteins that were not represented in HLA-I immunopeptidomes (Fig. 2c). The difference in median SLIDER score between groups was comparable to what was observed in the pool of in vitro digested and non-digested proteins (Fig. 2b). This suggests an immunologically relevant role of 20S proteasome-mediated processing of proteins with disordered regions in HLA-I antigen presentation.

### Development and benchmarking of inSPIRE 1.5 - aSPIRE for the identification and quantification of peptides produced by proteasome-catalyzed processing of entire proteins

The 15 proteins that were efficiently degraded by 20S proteasomes in our pilot assays had a length between 103 and 890 residues (Supplementary Table 1), and a broad range of disordered region length (Fig. 2b), which could allow an investigation of proteasome-catalyzed peptide hydrolysis and splicing not limited to a single protein example. To this end, we performed kinetic digestions of these 15 proteins using purified human 20S standard proteasomes and a proteasome to target molar ratio between 1:5 and 1:100, based on our preliminary outcomes of the protein degradation rates. In our experimental set-up, the

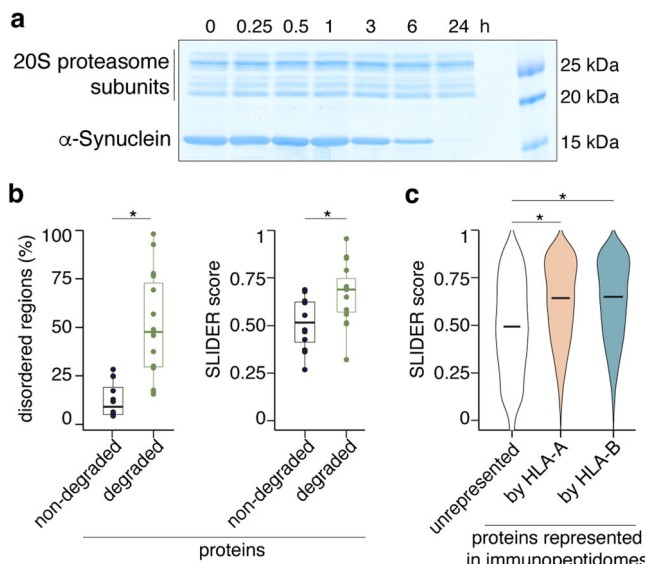

**a**

**b**

**c**

**Fig. 2 | Proteolysis and features of target protein candidates processed by 20S proteasomes. a** Coomassie blue stained SDS-PAGE gel of α-Synuclein processed by human 20S standard proteasomes. A representative gel of the two biological replicates is shown. Both α-Synuclein and 20S standard proteasomes subunits are visible. **b** The fraction of intrinsically disordered regions in the protein sequences (computed via fraction of the sequence that had a IUPred3 score ≥0.4) and the long disorder segments reported as SLIDER score differ between efficiently degraded ($n = 15$) and non-degraded ($n = 12$) proteins ($p = 1.1 \cdot 10^{-5}$ and $p = 0.009$ for IUPred3 and SLIDER scores, respectively). Most of the degraded proteins had a SLIDER score >0.538 that is referred as the threshold for long disorder segments. Box limits represent the first (Q1) and third (Q3) quartiles, with horizontal lines depicting the median, and the interquartile range (IQR) ranging from Q1 to Q3. Whiskers denote the Q1/Q3±1.5·IQR. **c** The distribution of the long disorder segments reported as SLIDER score among proteins represented in the HLA-A ($n = 13,689$) and HLA-B ($n = 13,755$) immunopeptidomes differs from that of unrepresented proteins in HLA-I immunopeptidomes ($n = 4767$), according to the IEDB database ($p = 5 \cdot 10^{-211}$ for HLA-A compared to unrepresented, $p = 9 \cdot 10^{-211}$ for HLA-B compared to unrepresented, $p = 0.13$ for HLA-A compared to HLA-B). In (**b, c**) the horizontal lines represent the median in the plots. Statistically significant differences between groups are labeled with * (two-sided Wilcoxon rank sum test with continuity correction; $p$ value < 0.05). Source data are provided as a Source Data file.

peptides produced in the in vitro digestions were measured by MS and searched against custom reference databases containing all possible non-spliced, *cis*- and homologous *trans*-spliced peptide sequences of the respective protein substrate. Although the digestion of full proteins might represent a remarkable progress in the understanding of PCPS as they should better mimic 20S proteasome activity in living cells, the size of the theoretically possible spliced peptide sequence database introduces statistical challenges in spliced peptide identification. This issue has been demonstrated for HLA-I immunopeptidomes[76], whereas it was less relevant for studies focused on in vitro digestions of synthetic polypeptides shorter than 50 residues, which have been the main source of information to decipher PCPS dynamics so far[39,40,50,55,59,60,82,92,93]. For the analysis of peptide products of the latter dataset, we recently proposed and validated the invitroSPI method[60]. For the analysis of more complex systems such as HLA-I immunopeptidomes and tryptic proteome digestions, we recently proposed and benchmarked the inSPIRE method, which utilizes Prosit spectral and retention time prediction for enhanced peptide identification via Percolator[72,94,95]. To study in vitro protein digestions, we modified the original inSPIRE and developed inSPIRE 1.5 (Supplementary Fig. 2) providing: (i) compatibility between inSPIRE 1.5 and MSFragger, facilitating the use of MSFragger[96] over Mascot as search engine, which allowed for a performant open access solution;

(ii) the origin of the peptide as a feature (i.e., a flag indicating whether a peptide matched to an MS2 spectrum was spliced or non-spliced), allowing inSPIRE 1.5 to deal with the expanded number of spliced candidates through automatic penalization of spliced peptide spectrum matches (PSMs) in the Percolator trained models (Supplementary Fig. 3). Details on further method improvements are provided in the Methods section.

We demonstrated the advantages of inSPIRE 1.5 over invitroSPI for the analysis of entire protein digestions by analyzing the in vitro degradation of three of them (i.e., H4, IF2, and RF1) with both tools. We identified 3973 and 1108 unique peptides either with inSPIRE 1.5 or invitroSPI, respectively, with 714 overlapping peptides (Supplementary Fig. 4a). This difference was confirmed at PSM level (Supplementary Fig. 4b). The higher peptide yield by applying inSPIRE 1.5 rather than invitroSPI was due to a higher recall of peptides, particularly spliced peptides, rather than a drop in the precision, since the distribution of the spectral angle between measured and Prosit-predicted MS2 spectra for *cis*- and *trans*-spliced peptides assigned by inSPIRE 1.5 was significantly higher as compared to invitroSPI (Supplementary Fig. 4c). Spectral angles between experimental and predicted MS spectra are commonly used as indicator of peptide identification quality[76,94,97,98].

While no specialized MS de novo tools exist for spliced peptide identification in this setting, a number of tools employ MS de novo strategies for spliced peptide identification in immunopeptidomics[46,61,69,99] and in in vitro polypeptide digestions[77,92]. Hence, we also benchmarked inSPIRE 1.5 against two simple MS de novo strategies, de novo 1 (DN1) and de novo 5 (DN5) methods (described in detail in the Methods section). Briefly, DN1 was designed to be a more stringent tool, i.e., more accurate in identification but less sensitive, while DN5 was designed to be more sensitive but with potential greater risk of false PSM assignments.

We then compared DN1 and DN5 to inSPIRE 1.5 using iBench 2.0, which is the latest update of our iBench benchmarking software[100]. iBench 2.0 takes high confidence peptide identifications from previously measured MS data and: (i) generates a constructed dataset containing only the PSMs originally assigned and (ii) embeds the peptide sequences in an in silico-only proteome as spliced or non-spliced peptides, which is used as constructed reference database. The constructed MS dataset can then be analyzed with the constructed reference database, thereby representing a ground-truth dataset, to enable benchmarking of an identification method. In particular, precision-recall (PR) curves are generated, comparing the fraction of PSMs identified which are correct and the fraction of all PSMs assigned by a method. Due to the complexity of embedding overlapping spliced and non-spliced peptides in a single substrate protein, for this application, iBench 2.0 used the database of peptide products identified by invitroSPI from multiple polypeptide digestions measured by MS (where the search space is dramatically reduced). The reference database was then generated by concatenating all polypeptide sequences to provide an in silico-only substrate protein as reference database (see Supplementary Fig. 5a and Methods for more details). The inSPIRE 1.5, DN1 and DN5 methods were then applied to the (pseudo) ground-truth dataset containing the non-spliced and spliced peptides produced in the digestions, thereby allowing the evaluation of precision—i.e., number identified peptides labeled correct over number identified peptides—and recall—i.e., number identified peptides labeled correct over the total pool of peptides—of all methods in the identification of both non-spliced and spliced peptides. To note, in this analysis, the non-spliced and spliced peptide product database that was used to define the correct sequence assignment was derived from the invitroSPI-mediated peptide product identification in the in vitro polypeptide digestions. The invitroSPI precision was not 100%[60]. Therefore, we considered this a (pseudo) ground-truth dataset.

The performance of both inSPIRE 1.5, DN1 and DN5 for non-spliced peptide identification was similarly high according to the iBench 2.0 analysis (Supplementary Fig. 5b), whereas inSPIRE 1.5 showed better performance for *cis*-spliced peptide identification (Supplementary Fig. 5c), particularly in terms of precision at the global false discovery rate (FDR) benchmark (indicated by the large dot of the precision-recall curve) and in its ability in differentiating correct and incorrect PSMs (indicated by the shape of the PR curve in Supplementary Fig. 5c). The performance of inSPIRE 1.5 for homologous *trans*-spliced peptide identification was intermediate between DN1 and DN5 methods (Supplementary Fig. 5d).

As a further test of inSPIRE 1.5 performance in spliced peptide identification, we generated a spliced peptide-free ground-truth dataset by applying iBench 2.0 to a large dataset of synthetic peptides[72,100,101]. The iBench 2.0 constructed a reference database represented by a protein sequence through the concatenation of 390 overlapping synthetic peptide sequences (Supplementary Fig. 6a; see Methods for details) and an MS dataset containing only the MS2 spectra of these 390 peptides. Since these peptide sequences were contiguous in the constructed reference database, a perfect method would identify only the non-spliced peptides in the ground-truth dataset and no spliced peptides should be identified.

We then applied inSPIRE 1.5 to this ground-truth dataset, and determined the PR-curve for all peptides identified (Supplementary Fig. 6b). At the Percolator estimated 1% FDR cut-off used in this study, we identified 250 non-spliced peptides (249 of which were assigned with the correct sequence) and wrongly identified 2 spliced peptides, only 0.7% of the identified peptides (Supplementary Fig. 6c), which we could consider as the margin of error in spliced peptide identification by inSPIRE 1.5 in a spliced peptide-free dataset.

Because of the high performance for *cis*-spliced peptide identification of inSPIRE 1.5 shown in Supplementary Figs. 4–6, we proceeded with the analysis of all 15 protein digestions by applying inSPIRE 1.5, and coupling it to the aSPIRE method, which added a quantitative dimension to our analysis (Supplementary Fig. 2). The method aSPIRE was developed by integrating Skyline[102] as an MS1 label-free quantification tool, thereby yielding quantitative information on all time points of the digestion kinetics of the 15 proteins (see Methods for more details). As demonstrated elsewhere[61,66,103], although MS1 label-free quantification cannot be used to directly compare individual peptides, it is a reliable strategy when distributions of hundreds of peptides are compared. In addition, aSPIRE contained a subsequent filtering step for the removal of substrate-derived contaminants, which here referred to peptides present in the purified substrate protein prior to processing by 20S proteasomes. The addition of quantitative information from aSPIRE to the inSPIRE 1.5 peptide list led to a further reduction of the final list of identified peptides, which contained only those peptides that were identified by inSPIRE 1.5 in at least one time point of the kinetics digestions and reliably quantified by aSPIRE at a given time point of the kinetics digestions (see Methods for details). Examples of the quantitative kinetics and the reproducibility between biological replicates are shown in Supplementary Fig. 7, and all kinetics are provided in the online repository (see Source Data Section).

Therefore, the inSPIRE 1.5 - aSPIRE pipeline provides a reliable peptide identification at a single peptide level and a quantification at the bulk peptide level, with a careful removal of contaminants from the analysis. The inSPIRE 1.5 - aSPIRE pipeline produces a variety of tabular and graphical outputs, e.g., a table with full annotation of each detected and quantified peptide, graphics of generation kinetics for each peptide, total ion chromatograms, coverage and residue maps. Both tools are available on GitHub, and a detailed explanation can be found in the online repository (see Source Data Section).

## Spliced peptides are efficiently produced by 20S proteasomes when processing proteins

We applied the inSPIRE 1.5 - aSPIRE pipeline to the digestion kinetics of the 15 proteins. We identified and quantified 16,219 non-spliced peptides (87%) and 2428 spliced peptides (13%; 2341 *cis*-spliced and 87 homologous *trans*-spliced peptides; Fig. 3a) setting a 1% FDR cut-off and a peptide length cut-off between 7 and 30 residues (see Methods for details). Since only a handful of proteins have been successfully digested in vitro by 20S proteasomes and measured by MS until today, we were not able to compare our results for the 15 proteins, efficiently digested by human proteasomes and here investigated, with previously published MS data. For human α-Synuclein and tau we could perform a comparison only by neglecting the proteasome origin. Indeed, for human α-Synuclein, Alvarez-Castelao et al.[104] identified 25 non-spliced peptides produced by rat 20S proteasomes. In our assay, we confirmed 24 out of the 25 non-spliced peptides in addition to 1228 previously undescribed non-spliced peptides (Table 1). For human tau, ref. 27, identified 64 cleavage sites used by *Thermoplasma acidophilum* 20S proteasomes. In our assay, we confirmed 63 out of the 64 of these cleavage sites in addition to 334 previously undescribed cleavage sites. Of course, we could not compare the protein-derived spliced peptide products with the literature because this has never been attempted before.

The total number of peptide products, identified by applying the inSPIRE 1.5 - aSPIRE pipeline to the digestion kinetics of the 15 proteins, varied between 350 and over 2900, and the frequency of the spliced peptides varied between 3% and 21% among the degraded proteins (Table 1). We observed a moderate correlation between the protein length and the total number of peptides detected (Supplementary Fig. 8a). However, no correlation between protein length and relative frequency of spliced peptides was observed (Supplementary Fig. 8b). Since inSPIRE 1.5 - aSPIRE could not only identify peptides but also estimate their abundance, we investigated the quantity of the non-spliced and spliced peptides identified in the 15 protein digestions. Overall, individual non-spliced peptides were on average more abundant than individual spliced peptides (Fig. 3b, Supplementary Fig. 9a), which was in agreement with what was observed in the in vitro digestions of synthetic polypeptides[40,50,77]. Indeed, non-spliced peptides accounted for 89.7% of all molecules, whereas *cis*-spliced and homologous *trans*-spliced peptides accounted for 9.8% and 0.5%, respectively (Fig. 3c). This result confirmed the higher rate of peptide hydrolysis over splicing.

The rate of peptide generation and the relative ratio between the quantity of non-spliced, *cis*- and homologous *trans*-spliced peptides did not significantly change over time in the in vitro digestion kinetics (Supplementary Fig. 9b), indicating that the likelihood of peptide splicing is equally high at early compared to late digestion time points and not an artifact due to peptide product accumulation over time. Furthermore, this suggests that peptide hydrolysis and splicing dynamics were relatively conserved during the progression of the protein degradation, in our experimental conditions.

To further test the quality of spliced peptide identification, we compared the spectral angle, Spearman correlation and iRT error distributions of the PSMs of the non-spliced and spliced peptides identified in the 15 protein digestion kinetics and compared them to the Prosit-predicted MS2 spectra (Fig. 3d, Supplementary Fig. 10, Supplementary Data 2–4, and Methods for more details). The spectral angle and Spearman correlation distributions were significantly higher and the iRT error was significantly smaller for spliced than non-spliced peptides, thereby further validating the spliced peptide identification by inSPIRE 1.5. The values reported in Fig. 3 and following figures referred to the inSPIRE 1.5 analysis done using a global 1% FDR cut-off, which is the most stringent cut-off used by most peptidomics studies. A variation of the global FDR cut-off would lead to a variation of the number of identified non-spliced and spliced peptides, of the relative

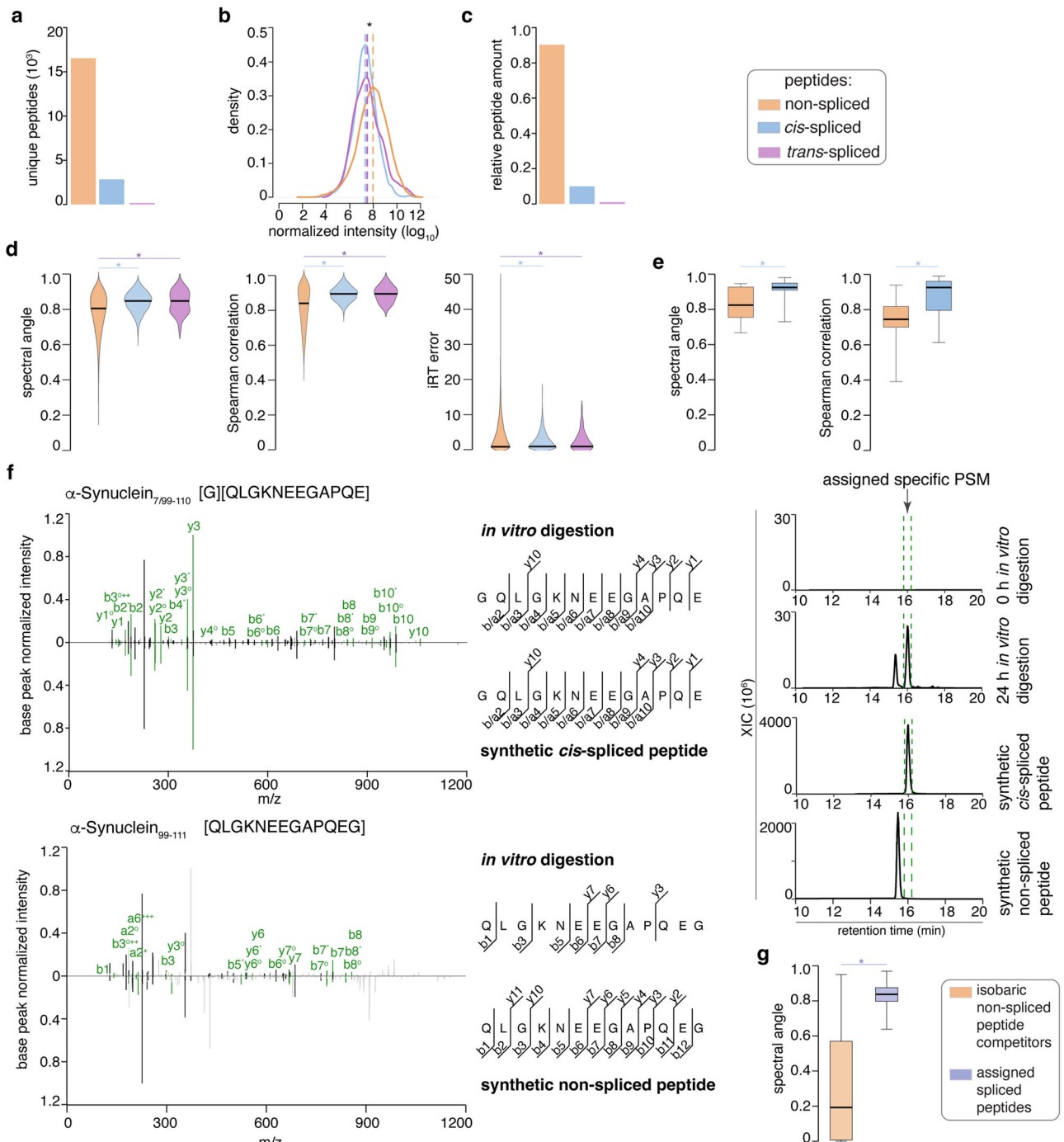

**Fig. 3 | Prevalence and validation of spliced peptides produced by 20S proteasomes from the 15 proteins efficiently degraded. a–g** refers to database of peptide products generated by 20S proteasome degradation of the 15 proteins efficiently degraded. **a** Number of unique non-spliced and spliced peptides. **b** Distribution of abundance of non-spliced and spliced peptides, which is significantly larger than of spliced peptides ($p = 2 \cdot 10^{-218}$ and $3 \cdot 10^{-29}$ for *cis-* and homologous *trans*-spliced peptides). **c** Relative abundance of non-spliced and spliced peptides produced. Abundance of peptides refers to the 4 h digestion time point (3 h for α-Synuclein), which is representative of the kinetics (Supplementary Fig. 9b). **d** Spectral angle ($p = 3.6 \cdot 10^{-127}$ and $p = 4.4 \cdot 10^{-6}$) Spearman correlation ($p = 1.7 \cdot 10^{-197}$ $p = 1.2 \cdot 10^{-9}$) and iRT prediction error ($p = 9.1 \cdot 10^{-90}$ and $p = 2.0 \cdot 10^{-9}$) between measurements and Prosit-predictions of non-spliced ($n = 17,265$), *cis*-spliced (n = 2341) and homologous *trans*-spliced ($n = 87$) peptides ($p$ values compare *cis-* and homologous *trans*-spliced to non-spliced). **e** Spectral angle ($p = 0.029$) and Spearman correlation ($p = 0.0096$) distribution between MS2 spectra of non-spliced ($n = 10$) and *cis*-spliced ($n = 23$) peptides compared to cognate synthetic

peptides. **f** MS2 spectra of the representative *cis*-spliced peptide [G][QLGKNEE-GAPQE] identified in α-Synuclein digestion, its synthetic peptide, and the synthetic peptide of its isobaric non-spliced peptide competitor [QLGKNEEGAPQEG]. Potential y-, b-, or a-ions matched between spectra are in green. Matched peaks of unknown origin are in black. Peaks not matched are in gray. Charge is shown by number $^+$ symbols. Ions' loss of water or ammonia is symbolized by $^o$ and $^*$. In the right panel, ion chromatograms (m/z range = 678.8163–678.8299) of the *cis*-spliced peptide in 0 and 24 h digestion, cognate synthetic peptide, and the synthetic isobaric non-spliced peptide are plotted. Lines delimit the range of the peak corresponding to the MS2 spectra in the left panel. **g** Spectral angle of the PSMs, assigned to spliced peptides ($n = 699$), compared to alternative isobaric non-spliced peptide competitors for the same spectra ($p \approx 0.0$). **e, g** Box limits represent first and third quartiles (Q1, Q3), horizontal lines the median, whiskers the Q1/Q3 ± 1.5·IQR. **b, d, e, g** Statistically significant differences are labeled with * (two-sided Wilcoxon rank-sum test with continuity correction). Source data are provided as a Source Data file.

**Table 1 | Number and frequency of peptide products identified and quantified in the 15 protein digestions by inSPIRE 1.5 - aSPIRE**

| Protein | Peptide products | | |
|---|---|---|---|
| | Non-spliced | Cis-spliced | Homologous trans-spliced |
| Annexin A1 | 859 (90.5%) | 88 (9.3%) | 2 (0.2%) |
| α–Synuclein | 1252 (97.1%) | 31 (2.4%) | 7 (0.5%) |
| CaM | 342 (97.2%) | 8 (2.3%) | 2 (0.6%) |
| Ffh | 2424 (81.7%) | 526 (17.7%) | 17 (0.6%) |
| H2A | 712 (92.6%) | 53 (6.9%) | 4 (0.5%) |
| H2B | 757 (91%) | 67 (8.1%) | 8 (1%) |
| H3 | 350 (79.2%) | 89 (20.1%) | 3 (0.7%) |
| H4 | 827 (87.5%) | 109 (11.5%) | 9 (1%) |
| IF2 | 1464 (91.9%) | 128 (8%) | 1 (0.1%) |
| IL-37b | 599 (85.7%) | 100 (14.3%) | 0 (0.0%) |
| LEDGF | 1994 (82.3%) | 417 (17.2%) | 13 (0.5%) |
| RF1 | 1019 (92.9%) | 76 (6.9%) | 2 (0.2%) |
| tau | 2372 (83.7%) | 456 (16.1%) | 6 (0.2%) |
| Ube2K | 646 (83.2%) | 120 (15.5%) | 10 (1.3%) |
| Ube2S | 602 (88.8%) | 73 (10.8%) | 3 (0.4%) |

Number of non-spliced, cis-spliced, and homologous trans-spliced peptide products identified and quantified in the kinetic digestions of the 15 proteins efficiently processed by human 20S proteasomes in vitro. The relative frequency of each peptide type is reported in parentheses.

frequency of the latter among the peptide products, and of the spectral angle distributions for both identified non-spliced and spliced peptides (Supplementary Data 2). According to inSPIRE 1.5 analysis, the identified spliced peptides had a higher spectral angle than non-spliced peptides, when we used a 2% global FDR cut-off; when we used a 1% global FDR cut-off, the identified spliced peptides had a similar spectral angle mean to the non-spliced peptides assigned in the golden-standard analysis of immunopeptidomics data by ref. [95], where the search space spanned non-spliced peptides from 85,919 proteins with unspecific cleavages (Supplementary Data 2; see Methods for details).

Like most of the other MS analytical methods, inSPIRE 1.5 computed the global FDR by comparing the searches of the whole target and decoy database. The inSPIRE 1.5 could also compute the mean posterior error probabilities from each peptide group, which allowed an estimation of the FDR per peptide group. By using this function, we could compute that, at the global 1% FDR cut-off, the estimated group FDR for non-spliced peptides was 0.7% and for spliced peptides was 3.2% (Supplementary Data 2).

As a further step of validation, we selected a pool of 24 cis-spliced and 10 non-spliced peptides and compared the experimental MS2 spectra of the synthetized peptides, of the corresponding PSMs of the in vitro digestions and of the Prosit MS2 spectrum prediction (Supplementary Data 5–7). The match between MS2 spectra of the non-spliced and spliced peptides identified in the protein digestions and the cognate synthetic peptides was high (both spectral angle and Spearman correlation medians were larger than 0.7; Fig. 3e, Supplementary Data 5, 6), although it was significantly higher for spliced peptides (Fig. 3e), thereby confirming the validation of both non-spliced and spliced peptides and the more stringent approach toward spliced peptide identification.

We also investigated all cases where a spliced peptide was clearly identifiable in the in vitro digestions, despite the existence of an isobaric non-spliced peptide in the reference database. Their assignment was confirmed by the comparison with the synthetic peptides' MS2 spectra for two representative cis-spliced peptides (Fig. 3f, Supplementary Data 7), thereby showing a higher spectral angle of the

MS2 spectra of the protein digestion kinetics with the synthetic cis-spliced peptides compared to the isobaric non-spliced peptide competitors (Supplementary Data 7). In the example shown in Fig. 3f, we can also appreciate the match in the retention time between the MS2 spectrum assigned to the cis-spliced peptide [G][QLGKNEE-GAPQE] identified in in vitro digestion of the human α-Synuclein and its cognate synthetic peptide. In the right panel of Fig. 3f, we can see a first peak in the chromatogram of the 24 h in vitro digestion, which corresponded to the isobaric non-spliced peptide [QLGKNEE-GAPQEG], thereby showing a perfect example of isobaric peptides both produced by proteasome and having different MS2 spectra and retention time, both assigned by applying inSPIRE 1.5 to the correct peptide sequence. The generation kinetics of these two peptides is reported in Supplementary Fig. 7a, b. Similarly, for the other 697 spliced peptides that had an isobaric non-spliced peptide competing for the same MS2 spectra, a significantly higher spectral angle and Spearman correlation between the measured and Prosit-predicted MS2 spectra of the spliced peptides compared to the isobaric non-spliced peptide competitors was observed (Fig. 3g, Supplementary Fig. 11a, Supplementary Data 7, 8). This phenomenon was driven by the isobaric peptide filter step introduced in inSPIRE 1.5 (Supplementary Fig. 2), which could efficiently discriminate between spliced peptide and isobaric non-spliced peptide sequences competing for a given MS2 spectrum. The assigned spliced peptides and the isobaric non-spliced peptide competitors differed in the iRT error, which was significantly smaller for the assigned spliced peptides (Supplementary Fig. 11b), as a further confirmation of the reliability of the spliced peptide identification by inSPIRE 1.5.

As last step of validation, we investigated if the proteasome to target molar ratio that we used in our in vitro digestions could impinge upon the frequency of spliced peptides identified. We selected the Ffh substrate, which gave a relative frequency of spliced peptides of 18.3%, in the condition of proteasome to target molar ratio 1:25 (Table 1). For this substrate, we performed in vitro digestions keeping the 20S proteasome concentration constant and reducing the proteasome to target molar ratio from 1:25 to 1:12.5 and 1:6.25 (Supplementary Fig. 12a, b). Reaction volumes containing similar amount of substrate were measured by MS - to avoid a bias in the peptide product identification due to the different peptide/protein amounts loaded into the MS – and the analysis was carried out by inSPIRE 1.5 – aSPIRE. Ffh degradation was measured by quantification on an SDS-gel and showed a similar degradation rate among the different conditions (Supplementary Fig. 12c). This suggests a $V_{max}$ state in our experimental conditions, which varied over time, in agreement with a study carried out with short fluorogenic peptide substrates[7]. The frequency of unique non-spliced and spliced peptides did not considerably vary among the different conditions although a trend of higher non-spliced peptide frequency was observed by decreasing the proteasome to target ratio in the example of Ffh digestion (Supplementary Fig. 12d). There was a similar coverage of the substrate sequence by the amount of non-spliced and spliced peptide products among the different conditions (Supplementary Fig. 12e). All these analyses suggest that the proteasome to target molar ratio should not have dramatically impinged upon the catalytic activities of 20S proteasomes in our experimental setting.

## PCPS in synthetic polypeptide and protein digestions

Existing information about the sequence preferences and driving forces of PCPS so far came from the analysis of in vitro digestions of synthetic polypeptides by proteasomes[39,40,50,55,59,60,77,93]. By applying the inSPIRE 1.5 - aSPIRE pipeline to the 15 protein digestions we were able to compare the characteristics of non-spliced and spliced peptides produced from proteins with those obtained from synthetic polypeptides. To this end, we compared the largest database of non-spliced, cis-spliced and homologous trans-spliced peptides

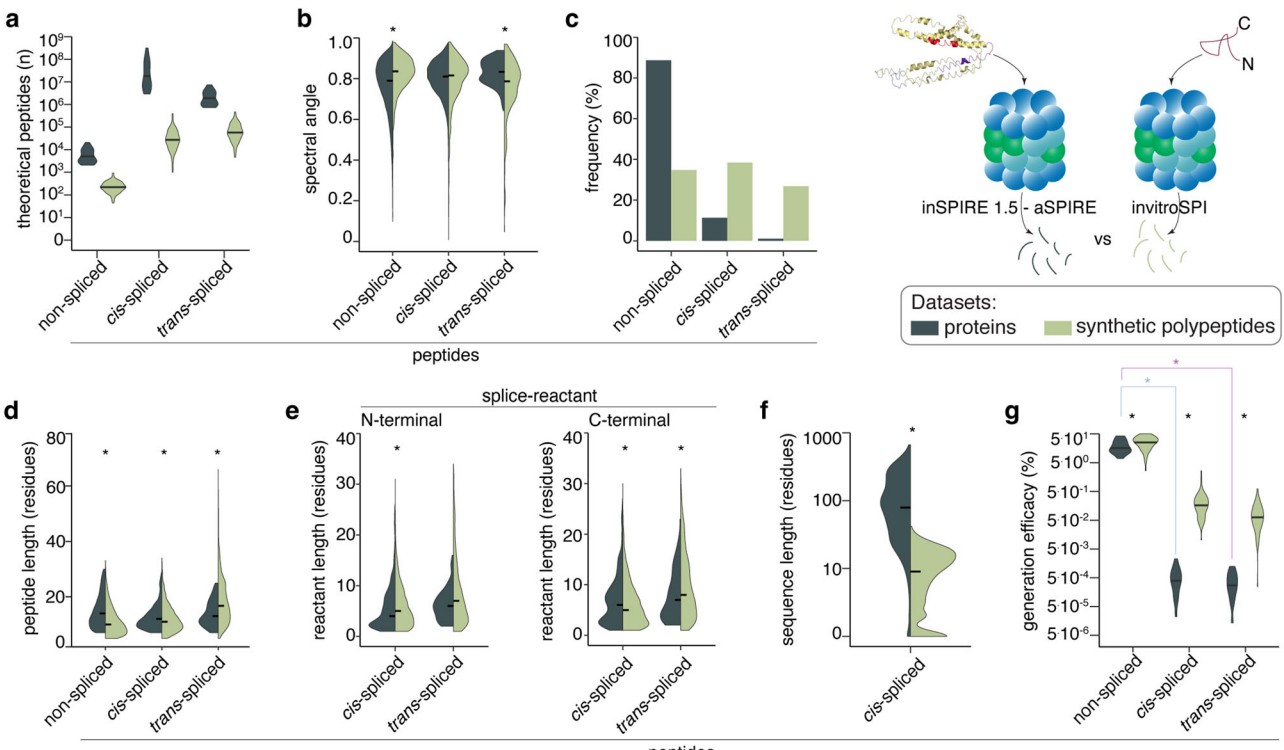

**Fig. 4 | Comparison of peptides produced in in vitro degradation of synthetic polypeptides and proteins. a** Size of theoretical peptide sequence space. Number of theoretically possible non-spliced and spliced peptide sequences that could be derived from the 15 efficiently digested proteins of this study and the 80 synthetic polypeptides in sequence-agnostic fashion. **b** Spectral angle distribution computed between experimentally measured and Prosit-predicted MS2 spectra of non-spliced and spliced peptides identified in protein and polypeptide digestions ($p = 5.4 \cdot 10^{-81}$ and $p = 6 \cdot 10^{-11}$ for non-spliced and *trans*-spliced PSMs, respectively). Only PSMs of peptides shorter than 13 amino acids are included in the analysis because of the dependency of Prosit performance on peptide length. For polypeptide digestions, only those substrates that were measured on high-precision MS ($n = 25$) are included in this panel for a fair comparison with the protein digestions measured with high-precision MS. **c** Relative frequency of non-spliced, *cis*- and homologous *trans*-spliced peptides among all peptide products. **d** Peptide length distribution of non-spliced and spliced peptides ($p = 0$, $1.2 \cdot 10^{-4}$, $1.2 \cdot 10^{-7}$ for non-spliced, *cis*- and *trans*-spliced peptides). **e** Splice-reactant lengths among *cis*- and *trans*-spliced peptides

(N-terminal: $p = 3.4 \cdot 10^{-9}$, $5.2 \cdot 10^{-2}$, C-terminal: $p = 7.8 \cdot 10^{-8}$, $1.7 \cdot 10^{-2}$ for *cis*- and *trans*-spliced peptides). **f** Intervening sequence lengths among *cis*-spliced peptides ($p = 0$). **g** Efficacy of generation of non-spliced and spliced peptides, computed as number of detected peptide products over the number of all theoretically possible peptide products ($p = 0.03$, $9.4 \cdot 10^{-10}$, $4.9 \cdot 10^{-9}$ for non-spliced, *cis*- and *trans*-spliced peptides comparing polypeptide and protein generation efficacies; $p = 1.3 \cdot 10^{-8}$, $2.6 \cdot 10^{-8}$ comparing generation efficacies of non-spliced to *cis*- and *trans*-spliced peptides in the protein database). In (**a**), peptide product databases generated in the in vitro digestions of either proteins ($n = 15$) or synthetic polypeptides ($n = 80$) by 20S standard proteasomes were reported. In (**b**, **d**–**g**) statistically significant difference between the peptide product databases of proteins vs synthetic polypeptides are labeled with * (two-sided Wilcoxon rank sum test with continuity correction). In the violin plots, horizontal black lines represent the median. Source data are provided as a Source Data file. The protein structure in the cartoon corresponding to (**d**) was adopted from[33].

($n = 5435$, 6005 and 4198, respectively) derived from the processing of synthetic polypeptides ($n = 80$)[60], with the here described database of 16,219, 2341 and 87 non-spliced, *cis*-spliced and homologous *trans*-spliced peptides, respectively, derived from the processing of 15 proteins. Both peptide product databases referred to in vitro digestions with human 20S standard proteasomes. The substrate length varied from 103 residues to 890 residues (Supplementary Table 1) among the 15 proteins and from 13 to 47 residues among the polypeptide digestions[60]. As expected, the size of the theoretical sequence search space of both non-spliced and spliced peptides was larger for the 15 degraded proteins dataset than the synthetic polypeptide dataset (Fig. 4a), with statistical implications that were approached by applying invitroSPI in the synthetic polypeptides and inSPIRE 1.5 in the 15 protein dataset. The spectral angles between the experimentally measured and the Prosit-predicted MS2 spectra depicted high quality PSMs in both peptide product databases (Fig. 4b). The most apparent difference between the two experimental systems was the frequency of homologous *trans*-spliced peptides, which was 0.5% in the protein digestions and 27% in the synthetic polypeptide digestions. Also, *cis*-spliced peptides had a lower frequency in the protein than the polypeptide digestions, although they still were a sizeable 12.6% of the

peptides detected in the protein digestions (Fig. 4c). In the latter dataset, we identified significantly longer non-spliced and *cis*-spliced peptide products, whereas the homologous *trans*-spliced peptides were shorter than in polypeptide digestions (Fig. 4d). In addition, among *cis*-spliced peptides, N-terminal splice-reactants were significantly shorter and C-terminal splice-reactants were significantly longer in protein than in polypeptide digestions (Fig. 4e). A striking difference between the intervening sequence length of *cis*-spliced peptides produced in protein and synthetic polypeptide digestions also emerged (Fig. 4f), which was likely driven by the restriction of intervening sequence length in the synthetic polypeptide digestions due to a very limited substrate length.

Therefore, although peptide products were identified with relative high confidence and partially showed similar features in both experimental systems, the initial analysis of the spliced peptides produced by 20S proteasomes from the 15 proteins showed important differences from what was known so far from the digestion of synthetic polypeptides. A similar scenario appeared when we computed the peptide generation efficacy, which could be described as the ratio between the number of non-spliced and spliced peptides identified in both experimental systems over the cognate number of theoretically

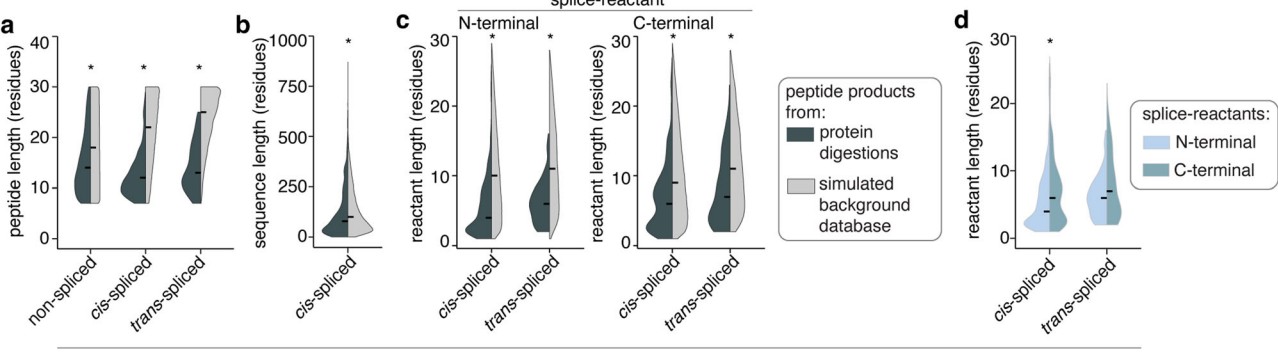

**Fig. 5 | Characteristics of non-spliced and spliced peptides produced by 20S proteasomes from the 15 proteins.** In (a–c), the results observed in vitro were compared to those derived from a simulated background database in silico, computed from the same substrate sequences. **a** Peptide length distribution of non-spliced and spliced peptides ($p = 0$, 0, $3.2 \cdot 10^{-37}$ for non-spliced, cis-spliced and trans-spliced peptides). **b** Intervening sequence lengths of cis-spliced peptides ($p = 1.5 \cdot 10^{-22}$). **c** Splice-reactant lengths of cis- and trans-spliced peptides (N-terminal: $p = 8.5 \cdot 10^{-251}$, $3.7 \cdot 10^{-15}$, C-terminal: $p = 3.1 \cdot 10^{-109}$, $9.4 \cdot 10^{-9}$ for cis- and trans-spliced peptides). **d** N- and C-terminal splice-reactants length of spliced peptides ($p = 3.2 \cdot 10^{-22}$, 0.2 for cis-spliced and homologous trans-spliced peptides). In (a–d), statistically significant difference between pairs is labeled with * (two-sided Wilcoxon rank sum test with continuity correction) and qualitative analysis, i.e., counting the number of unique peptides produced by 20S proteasome whilst degrading the 15 proteins, was used. In the violin plots, horizontal black lines represent the median. Source data are provided as a Source Data file.

possible peptides (the latter is reported in Fig. 4a). In both experimental systems, a higher generation efficacy for non-spliced peptides compared to cis-spliced peptides and homologous trans-spliced peptides was observed, although for all peptide types the generation efficacy in the synthetic polypeptide digestion was significantly higher than in protein digestions (Fig. 4g). Based on the latter observation, we could speculate that in vitro digestions of synthetic polypeptides depict a greater variety of peptides produced by 20S proteasomes than in vitro protein digestions, including those peptides that would have very low abundance and hence little relevance in physiological systems.

## Driving forces of non-spliced and spliced peptide generation by human 20S proteasomes when processing proteins

We believe that the generation efficacy of non-spliced and spliced peptides deserves particular attention because it could better frame the controversy about the putative biochemical efficacy of PCPS[37,41]. In the 15 protein digestions, the identified non-spliced peptides were 1 out of 6 possible non-spliced peptides that could be derived from these proteins. Such a frequency was considerably lower for cis-spliced peptides (1 out of 262,652) and homologous trans-spliced peptides (1 out of 335,505), thereby confirming that, in terms of qualitative peptide product variety, the efficacy of peptide hydrolysis was significantly higher than PCPS (Fig. 4g). Despite the number of homologous trans-spliced peptides identified in this study was much lower than the number of identified cis-spliced peptides, the generation efficacy of cis- and homologous trans-spliced peptides did not significantly differ because the theoretical homologous trans-spliced peptide sequence space was a tenth of the theoretical cis-spliced peptide sequence space of the 15 degraded proteins (Fig. 4a, and Methods for details).

If peptide hydrolysis and peptide splicing catalyzed by 20S proteasomes were not driven by any catalytic specificity, the characteristics of an experimentally measured peptide product and a product of a simulated background database, wherein the peptide sequences were generated in silico by randomly picking and combining peptide fragments, should not significantly differ. Therefore, we performed an analysis of the characteristics of the non-spliced and spliced peptides produced in the 15 protein digestions at the qualitative level—i.e., considering the number of unique peptides rather than their quantity—and verified the outcome using a simulated background database. In contrast to the null hypothesis that peptide hydrolysis and peptide splicing were unspecific ("random") processes, both non-spliced and

spliced peptides were shorter than what would have been expected from a random process. This conclusion emerged by comparing the lengths of identified peptide products to those of a simulated background database derived in silico from the 15 degraded proteins (Fig. 5a). Similarly, both the intervening sequences and the splice-reactants of the spliced peptide products in the protein digestions were shorter than what would have been expected by a random process (Fig. 5b, c). These observations were confirmed in the digestions of synthetic polypeptides (Supplementary Fig. 13a–c). Because the distributions of the length of the N- and C-terminal splice-reactants derived from a simulated background database did not significantly differ (Fig. 5c), the fact that the experimentally identified N-terminal spliced reactants were, on average, significantly shorter than their C-terminal counterpart (Fig. 5d) suggests that the former underlay more strongly a length constraint than the latter.

On top of studying the peptide products detected in the 15 protein digestions, we also investigated the protein sequence properties that might favor peptide splicing or hydrolysis. In the absence of any catalytic specificity, we would expect a uniform distribution of the number of peptide molecules produced by 20S proteasomes along the protein sequence at a given time point. In contrast, a quantitative evaluation of the localization of the peptide molecules produced during the processing of the 15 proteins suggests that there were specific areas within the proteins that were preferentially processed by 20S proteasomes. These "hotspot" regions, where the peptide product molecules were preferentially derived from, differed among proteins, captured both non-spliced and spliced peptide products in a similar fashion, and did not seem to be correlated to neither the presence of predicted disordered protein segments nor predicted protein secondary structure elements (Fig. 6a, Supplementary Figs. 12e, 14a, b). The location of 'hotspots' in the in vitro protein digestions remained stable over the digestion time points (Fig. 6b, Supplementary Figs. 12e, 14b), thereby suggesting a preference of the proteasome for a specific substrate region that was consistent during the course of the reactions.

We then zoomed in the 15 degraded protein sequences and focused on the specific local substrate sequence environment. We used the quantity estimation of aSPIRE and computed the usage of each substrate residue for peptide hydrolysis (SCS-$P_1$) and peptide splicing (PSP-$P_1$). Details are described in the Methods and Supplementary Fig. 15. The SCS-$P_1$ and PSP-$P_1$ analysis did not show a direct correlation between the quantitative usage of individual substrate residues for peptide hydrolysis and peptide splicing (Fig. 6c,

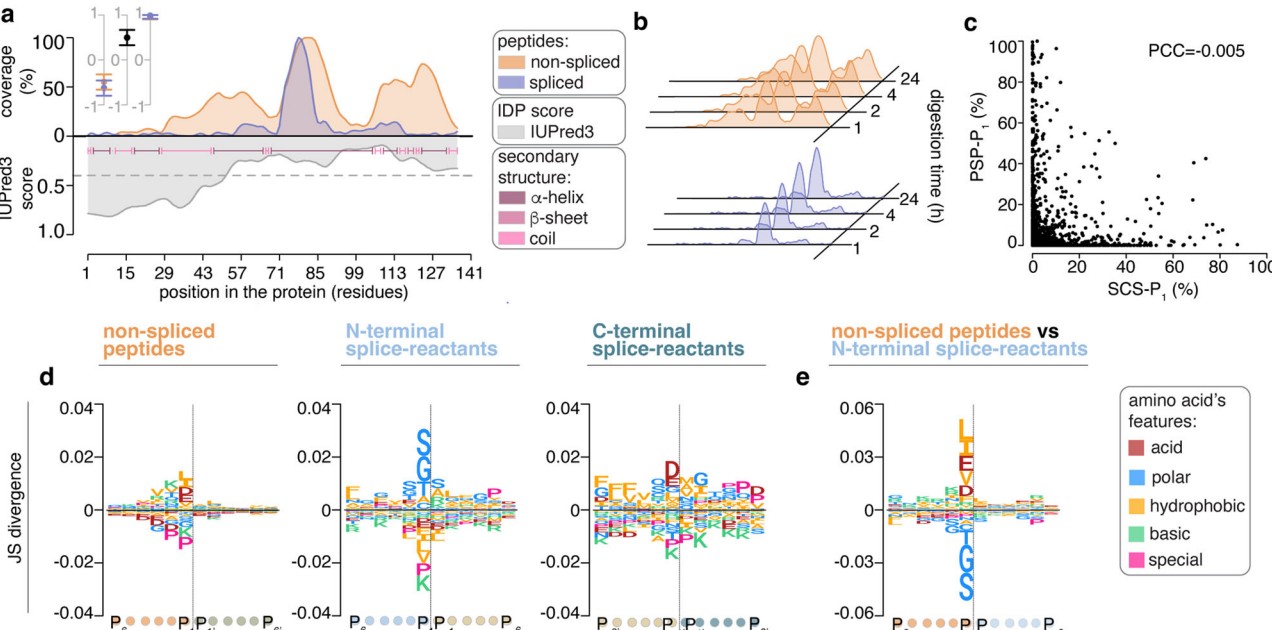

**Fig. 6 | Peptide products' hotspots and sequence preferences. a, b** Quantitative protein coverage profiles formed by non-spliced and spliced peptide products detected in in vitro digestion of the representative protein H3. In (**a**) the coverage is compared to the IUPred3 score predicting the presence of disordered protein segments. The gray dashed line represents the score = 0.4, which is used as threshold for a disordered segment. Predicted secondary structure elements are depicted as pink bars. The Spearman correlation coefficients and confidence intervals in the inlets denote (from left to right, respectively) correlation between: (i) spliced / non-spliced sequence coverage and IUPred3 score, (ii) spliced and non-spliced sequence coverages and (iii) spliced and non-spliced sequence coverage profiles across all time points of the digestion kinetics. Dots represent estimated Spearman correlation coefficients and error bars denote the confidence interval. Coverage profiles were normalized for spliced/non-spliced peptides separately. In

(**b**) the protein coverage profiles at different digestion time points are shown. Coverage profiles were normalized for spliced/non-spliced peptides across all time points. **c** Scatter plot of all SCS-$P_1$ and PSP-$P_1$ computed for the 15 proteins upon in vitro digestion by 20S proteasomes. A dot represents a single protein substrate residue. The Pearson's product moment correlation coefficient (PCC) is reported. **d** The Jenson-Shannon (JS) divergence of all non-spliced and spliced peptide sequences compared to a simulated background database is depicted. **e** The JS divergences between non-spliced and spliced peptide sequence motifs are shown. In (**a**, **b**) peptide amount was aggregated per substrate residue to yield quantitative coverage (see Methods for details). In (**c**, **d**, **e**) quantitative analysis, i.e., estimating the peptide amount, of the peptide products of the 15 efficiently digested proteins has been used. Source data are provided as a Source Data file.

Supplementary Fig. 16). A similar approach was applied to the in vitro digestions of synthetic polypeptides in previous studies and arrived at similar conclusions[40,50]. Both outcomes point toward diverse (and potentially competing) driving forces and peptide sequence preferences of peptide hydrolysis and peptide splicing catalyzed by proteasomes.

To understand what these peptide sequence preferences were, we investigated which amino acids were predominantly present around cleavage and splice sites, using quantitative information obtained through aSPIRE, and compared them to a simulated background database (see Methods for details). The latter step was introduced to account for the amino acid content in the 15 efficiently degraded protein sequences and avoid artefacts describing the protein substrate sequence motifs rather than the peptide products' sequence motifs. For the non-spliced peptides, the sequences at the C-terminus of the peptide, next to the cleavage site, appeared to bear the most critical residues, with increasing divergence from the sequence background as they were getting closer to the $P_1$ position, where acidic and hydrophobic amino acids were preferentially represented (Fig. 6d). Similarly, the residues at the $P_1$ position of spliced peptides showed the highest divergence from the sequence background, although with polar and small amino acids being dominant (Fig. 6d). The $P_{1'}$ position, which is supposed to perform the nucleophilic attack on the $P_1$ residue, showed less divergence from the background as compared to $P_1$ and $P_{-1'}$ positions (Fig. 6d), perhaps suggesting a secondary role of the C-terminal splice-reactant as compared to the N-terminal splice-reactant in PCPS. Proline (P) and lysine (K) were disfavored for both peptide splicing and hydrolysis at $P_1$ compared to the sequence

background (Fig. 6d). In opposite, several other amino acids were preferentially used for either peptide hydrolysis or splicing, as emerged by the direct comparison of the sequences of non-spliced and spliced peptide products, wherein the amino acids at $P_1$ seemed to be the main driver of the different fate of a peptide to be released upon peptide hydrolysis or splicing (Fig. 6e). Historically, the proteolytic activity of proteasomes has been divided into trypsin-like (K, R), chymotrypsin-like (hydrophobic amino acids: W, F, Y, L, I, V, A or M) and caspase-like (D, E) activities based on short fluorogenic substrate assays[105–108]. More recent studies have suggested that the correlation between these assays and the substrate sequence preference of proteasomes in cleaving longer peptides is weak[8,109,110], and that proteasomes can cleave after any amino acid of a polypeptide and not only after those amino acids defined by the trypsin-, chymotrypsin- and caspase-like activities[59,60]. In our data, the most frequently used amino acids at the $P_1$ position of peptide hydrolysis would fit into the chymotrypsin- and caspase-like patterns, but not the trypsin-like motifs. A similar pattern is observed at the $P_{-1'}$ position of spliced peptides, which must be hydrolyzed in order to generate the N-terminus of the C-terminal splice-reactant (Fig. 6d). None of the three classical proteasome activities would well describe the frequency of amino acids at the $P_1$ position of spliced peptides (Fig. 6d, e).

At last, we investigated whether these local amino acid preferences could be related to "hotspots" observed at a global substrate sequence level. To separate hotspot from non-hotspot regions, we computed the quantitative coverage of each residue used by peptide hydrolysis and splicing and defined a coverage cut-off, so that half of all detected $P_1$ positions - those with higher coverage - fell within

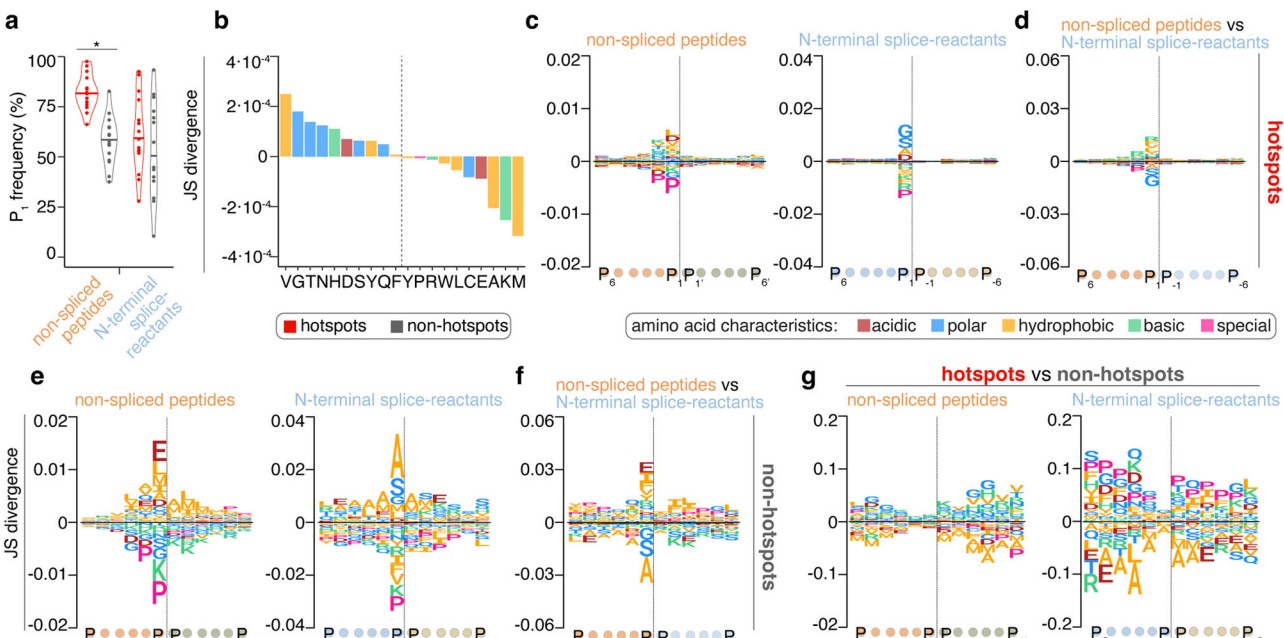

**Fig. 7 | Features of hotspot and non-hotspot regions. a** The frequency of residues used as $P_1$ position by peptide hydrolysis and splicing in the hotspot and non-hotspot regions of the 15 efficiently degraded proteins. The statistically significant difference between pairs is labeled with * (two-sided Wilcoxon rank sum exact test; $p$ value = 1.8·10$^{-6}$). In the violin plots, horizontal black lines represent the median and dots the single proteins. **b** Amino acid frequency in hotspot and non-hotspot regions relative to the overall amino acid frequency among the 15 efficiently degraded proteins. Positive JS divergence indicates a dominance of these amino acids in hotspot regions, negative divergence indicates a dominance in non-hotspot regions. Peptide sequence motifs of non-spliced peptides and N-terminal splice-reactants either in hotspots (**c**, **d**) or non-hotspots (**e**, **f**). In JS divergence between the sequence motifs of either non-spliced peptides or N-terminal splice-reactants and the protein sequence background, inside (**c**) and outside (**e**) the hotspot regions. Positive values represent an overrepresentation in the specific region. In JS divergence between the sequence motifs of non-spliced peptides and N-terminal splice-reactants, inside (**d**) and outside (**f**) the hotspot regions. Positive values represent an overrepresentation among non-spliced peptides. **g** The JS divergence of the sequence motifs of either non-spliced peptides or N-terminal splice-reactants comparing hotspot vs non-hotspot regions. Positive values represent a prevalence in hotspots. Source data are provided as a Source Data file.

hotspot regions, and half of all detected $P_1$ positions—those with lower coverage – fell within non-hotspot regions (see Methods for details). Hotspots were independently identified for non-spliced and spliced peptide products, although they partially overlapped as shown in Fig. 6a and Supplementary Fig. 14. For PCPS, we focused on N-terminal splice-reactants for a better comparability to non-spliced peptides since they both formed an acyl-enzyme intermediate with the threonine 1 of the proteasome catalytic subunits.

We then computed the density of residues used as $P_1$ position, which was significantly higher in hotspots than non-hotspots of peptide hydrolysis. In hotspots, a median of 82% and 59% of the residues were used as $P_1$ residues by peptide hydrolysis and splicing, respectively, in opposite to the 58% and 45% observed in the non-hotspots of peptide hydrolysis and splicing, respectively (Fig. 7a). This showed that not only was a larger amount of peptide products generated, but also substrate residues were more often used for peptide hydrolysis in hotspots than non-hotspots.

This phenomenon could, in part, be explained by the fact that hotspot regions had a relatively higher frequency of amino acids that were preferentially observed at the $P_1$ position of both spliced and non-spliced peptides (V, D, F, Y, H for peptide hydrolysis and G, T, S for peptide splicing). Non-hotspot regions, on the other hand, had a relatively higher frequency of amino acids that were disfavored by both catalytic processes (e.g., K, P; Fig. 7b).

In hotspots, the residues at the $P_1$ position were the most different compared to a background sequence distribution derived from a randomization of the protein sequence, as represented by the Jensen-Shannon (JS) divergence, thereby pointing to a strong preference of $P_1$ residues for the catalysis than the surrounding residues. The more they diverged, the more distinct a sequence pattern was from the background. This phenomenon occurred for both peptide hydrolysis and

splicing hotspots (Fig. 7c), and confirmed the highest divergence at the $P_1$ position between the sequence preferences of peptide hydrolysis and splicing (Fig. 7d). A similar behavior was observed outside the hotspots, although the JS divergence was even higher than in the hotspots, with the emergence of influential amino acids in the positions surrounding the peptide hydrolysis and splicing reaction sites (Fig. 7e). Similarly, the peptide sequence motifs of the non-spliced peptides and the N-terminal splice reactants diverge more in non-hotspots (Fig. 7f) than in hotspots (Fig. 7d). When we directly compared the peptide sequence motifs of the peptide products inside and outside the hotspots, we observed the smallest divergence at the $P_1$ position for both peptide hydrolysis and splicing, which progressively increased in the more distal positions (Fig. 7g). This scenario is compatible with the hypothesis that proteins have regions prevalent with amino acids preferred by peptide hydrolysis and splicing. In these regions (hotspots), a larger portion of residues are used as $P_1$ sites (Fig. 7a), and, on average, each of them is used more frequently by proteasomes - generating a larger amount of peptide products (definition of hotspots) - compared with other regions (non-hotspots), where residues such as P and K, which disfavor the reactions, are more prevalent. Hence, in non-hotspots, not only there is a $P_1$ site preference but also other amino acids motifs at the surrounding are needed to support peptide hydrolysis and splicing, thereby compensating the overall amino acid environment for peptide hydrolysis and splicing outside the hotspots.

## Discussion

Proteasomes are amongst several proteases that can catalyze peptide splicing. Indeed, this endergonic reaction is also catalyzed by cathepsins, connectases, plant legumains and other enzymes[42,43,62,111,112]. The barrel conformation of proteasomes with tight regulation of the gates

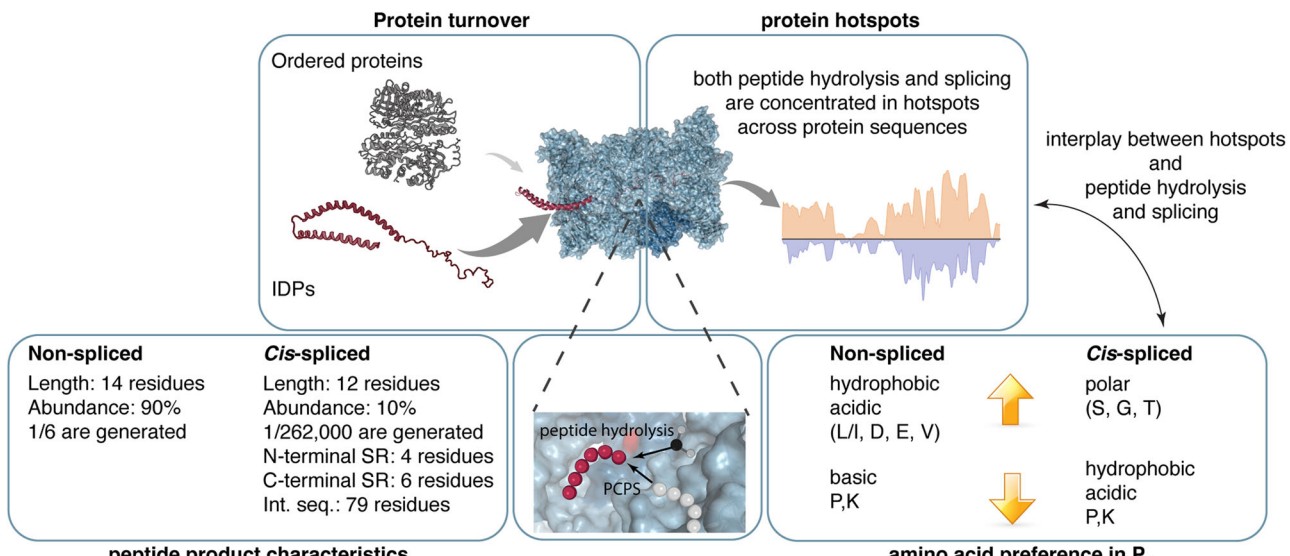

**Fig. 8 | Characteristics of peptide hydrolysis and peptide splicing catalyzed by human 20S proteasomes.** Characteristics of 20S proteasome-mediated protein digestions, resulting in non-spliced and spliced peptide products with a biochemically distinct signature. SR: splice-reactant. Int. seq.: intervening sequence. Medians of the various characteristics are reported.

and the hydrophobic inner environment are structural factors that can particularly promote peptide splicing in proteasomes as compared to the other enzymes[43]. Nonetheless, especially for proteasomes, some scientists argued that "peptide splicing is, at most, an extremely rare event and likely does not happen at all"[37,41]. This first insight in the protein processing by 20S proteasomes contradicted this statement, since *cis*-spliced peptides represented a sizeable portion of the variety of peptides produced in our experimental system. For their identification, we developed the inSPIRE 1.5 – aSPIRE workflow and cognate software, which could address the large-search-space problem, identify spliced peptides by assigning PSMs with high confidence (as confirmed by the high spectral angle and Spearman correlation and low iRT prediction error) and resolve technical pitfalls such as the discrimination between spliced and isobaric non-spliced peptide competing for the same PSM. This latter function could lead to the identification of both spliced and isobaric non-spliced peptides, in different part of the MS chromatograms (e.g., see Fig. 3f), thereby increasing the overall peptide identification recall and confirming the importance of the RT feature for PSM assignment, in agreement with other studies[69,70,94,96,113].

Despite the sizeable variety of *cis*-spliced peptides detected in the protein digestions, they represented only a small fraction of the spliced peptides that could have been produced by 20S proteasomes during protein processing, and were produced on average in smaller amounts than non-spliced peptides, therefore confirming that 20S proteasomes preferentially catalyze peptide hydrolysis.

The results obtained from both protein and synthetic polypeptide digestions confirmed the hypothesis that both peptide hydrolysis and peptide splicing are driven by catalytic factors rather than being random processes, generate shorter peptide products than would be expected from a random process, and are driven by peptide sequence preferences that diverge between the two processes (Fig. 8). Spliced peptides also seem to favor short N-terminal splice-reactants, which might well fit with a model of transpeptidation, which has been proposed for several other proteases such as sortases, butelase and asparaginyl endopeptidases[42,43]. Indeed, proteases are catalysts for the hydrolysis of a scissile bond. Although such a process is theoretically reversible, the presence of water makes the reverse reaction kinetically unfavorable. The concept of transpeptidation refers to the attack of the unstable substrate-protease acyl intermediate bond by the N-terminal amine group of a peptide rather than by a nucleophilic water molecule. The energy barrier of amidation can be overcome by proteases that drive amidation by retaining the substrate in their S'-pockets, thereby blocking water access to the substrate-protease acyl intermediate. This, in turn, can allow for the subsequent nucleophilic attack by peptides in close proximity. In 20S proteasomes, this process is catalyzed by the catalytic threonine at the residue 1 of the six proteasomal catalytic subunits and it has been described in mammals and yeasts[40,52]. Transpeptidation as a mechanism catalyzing peptide splicing in proteasomes has been proposed and confirmed by Van den Eynde lab in several studies by performing experiments with modified synthetic peptides in vitro and mini-genes *in cellula*[38,54,56,58], and later further confirmed by other researchers via biochemical experiments with either heavy water or nuclear magnetic resonance spectroscopy[40,114]. One might speculate that the stability of the acyl intermediate and the characteristics of the peptide fragment determine its catalytic fate of either peptide splicing or hydrolysis. This could, for instance, involve steric and physicochemical features of the N-terminal splice-reactant. Short fragments bound to the proteasome catalytic threonine 1 could affect catalysis and intermediate stability in a way that favors transpeptidation and disfavors peptide hydrolysis. The comparison of the peptide product's sequence motifs (shown in Figs. 6–7) suggests that the residue that impacts most on peptide hydrolysis and splicing is located in the $P_1$ position, and hence forms the acyl-enzyme intermediate with the proteasomal threonine 1. That residue not only drives the efficacy of the catalysis but also the outcome of the process, with hydrophobic and amino acids that preponderate toward peptide hydrolysis whereas polar amino acids toward peptide splicing (Fig. 8).

Our study on PCPS during protein digestion also provided a first general estimation of the intervening sequence length distribution among *cis*-spliced peptides, unbiased by experimental design and limitations in identification methods present in other studies[59,60]. The intervening sequences had a median of 79 residues and a quartile of the distribution equal to 30 residues (Fig. 8). Although intervening sequences were shorter than expected by chance – in line with the initial hypothesis of Van den Eynde lab[54] – the mild difference from the simulated background might indicate that this factor is not among the main drivers of PCPS and that distances between splice-reactants could cover the entire length of a protein.

Among spliced peptides, our study of protein digestions by 20S proteasomes also shed light on homologous *trans*-PCPS, which represented only a minor fraction of the peptide products in protein digestions. Nonetheless, in both polypeptides and protein digestion datasets, the generation efficacy of *cis*- and homologous *trans*-spliced peptides did not differ (Fig. 4g), suggesting that both processes might be equally likely to occur in a scenario where only a single substrate is available for proteasomal digestion. *In cellula*, where proteasomes have access to thousands of proteins, homologous *trans*-peptide splicing events should be very rare because they would require a concomitant or successive processing of two molecules of a given protein by the same proteasome complex. If the generation efficacy of homologous and heterologous *trans*-PCPS were comparable, we might expect the generation of a pool of heterologous *trans*-spliced peptides characterized by a very large sequence variety and a very small amount. To date, their presentation by MHC-I molecules has only been shown in ref. 61, and their biological and immunological relevance remains to be proven. We speculate that the higher generation efficacy of *cis*- and homologous *trans*-spliced peptides observed in in vitro polypeptide digestions compared to protein digestions could be due to a higher sampling depth in synthetic polypeptide digestion measurements, thereby allowing the detection of even low abundant peptide products[40,59,60], and potentially overestimating the generation efficacy of *cis*- and *trans*-spliced peptides physiologically occurring *in cellula*.

The analysis of the 15 protein digestions also suggests that the proteasome processes different protein sequence regions with different efficiencies and that peak regions of proteasomal processing – i.e., hotspots – are partially shared between non-spliced and spliced peptides. These hotspot regions remain stable over the course of the protein digestions and do not correlate with the disordered regions or secondary structure elements of proteins. A preliminary analysis of the peptide sequence preferences in hotspots as compared to non-hotspot regions pointed toward a general prevalence of amino acids preferred by peptide hydrolysis and splicing in hotspots in contrast to a unfavorable amino acid environment for proteasomal catalysis in the non-hotspots, although this factor might not be the only driver of this phenomenon.

## Methods

### Protein purification

Mature form of mouse IL-1α (130-094-051), human IL-1α (130-093-894), mouse IL-1β (130-101-682) and human IL-1β (130-095-374) were purchased from Miltenyi Biotec. Recombinant human mature IL-37b (#7585-IL-025/CF), recombinant human Annexin A1 (#3770-AN), and recombinant human tau (#SP-495-100) were purchased from R&D Systems. Enolase 1 (#E6126), and chicken Ovalbumin (#2512) were purchased from Sigma. Human α-Synuclein was recombinantly expressed in *E. coli* strain BL21(DE3) and purified as previously described[115]. Briefly, the cells from 1 L minimal medium expression culture were lysed by three freeze-thaw cycles followed by sonication, boiled for 15 min and centrifuged at $48.000 \times g$ for 45 min. Contaminating DNA was precipitated and removed from the supernatant by adding streptomycin (10 mg/ml) to the ice-cold stirred solution. After centrifugation, α-Synuclein was precipitated from the supernatant by adding ammonium sulfate. The protein pellet was resuspended in 25 mM Tris/HCl, pH 7.7 and further purified by anion exchange chromatography on a 30 ml POROS HQ column (PerSeptive Biosystems). The protein was further purified by gel filtration on a HiLoad 16/60 Superdex 75 column (Cytiva) equilibrated with 50 mM HEPES, pH 7.4, 100 mM NaCl. To remove any aggregated protein, the eluted monomeric α-Synuclein was centrifuged at $106,000 \times g$ for 1 h at 4 °C and filtrated through 0.22 μm ULTRAFREE-MC centrifugal filter units (Merck Millipore). The final protein concentration was adjusted to 0.3 mM. Recombinant CaM, EF-G, EF-Ts, Ffh, H2A, H2B, H3, H4,

HUWE1, IF2, LEDGF, LRP130, PDF, RF1, UbcH7, Ube2K, and Ube2S were prepared according to previously published protocols[112,116–123].

The standard 20S proteasomes used in this study were obtained from different sources. The mouse 20S proteasomes used for the degradation of mature form of mouse IL-1α and IL-1β were purchased from R&D Systems (E-355; see Supplementary Fig. 1). The human 20S proteasomes used for the degradation of mature forms of human IL-1α, IL-1β and IL-37b were purchased from Enzo Life Sciences (BML-PW8720-0050; see Supplementary Fig. 1). The human 20S proteasomes used for the degradation of Annexin A1, α-Synuclein, CaM, EF-G, EF-Ts, Ffh, Enolase1, H2A, H2B, H3, H4, HUWE1, IF2, IL-37b, LEDGF, LRP130, Ovalbumin, PDF, RF1, tau, UbcH7, Ube2K, and Ube2S (Figs. 2–6; Supplementary Figs. 1, 12) were purified from HeLa cells as described elsewhere[124]. In brief, the clarified HeLa cells extract was subjected to several rounds of differential precipitation with Poly-EthyleneGlycol400 (PEG400) and 10–30% sucrose gradient ultra-centrifugation. The purified 20S proteasomes were stored in buffer containing 5% (w/v) sucrose, 5 mM DTT and 0.01% (w/v) Lauryl Maltose Neopentyl Glycol (LMNG). Proteasome concentration was determined by Bradford assay using BSA as a standard and verified by SDS-PAGE and Coomassie staining. The purity of the 20S proteasome preparation obtained with this method (Supplementary Fig. 17) is comparable to previous purification methods[9,14]. The 20S proteasomes subunits typically migrate at around 20–30 kDa on SDS-PAGE. In our purification, we sometimes observed some protein bands migrated at between 37 and 100 kDa. We identified the protein bands by in-gel trypsin digestion followed by MS measurement (Supplementary Fig. 17) and none of them were proteases that were able to catalyze either proteolysis or transpeptidation (Supplementary Data 9).

### In vitro digestions of proteins by 20S proteasomes

For the initial identification of which proteins could be efficiently degraded by 20S proteasomes in our in vitro system (Fig. 2a, Supplementary Fig. 1), the following conditions have been used (final concentrations are reported in parentheses):

1. Mature form of mouse IL-1α (5.71 μM) and IL-1β (5.71 μM), human IL-1α (5.71 μM), IL-1β (5.71 μM) and IL-37b (9.52 μM) digestion were carried out by incubating the respective protein substrate with either mouse (for mouse proteins) (1.47 μM) or human (for human proteins) 20S standard proteasomes (4 μM) in 50 μl TKMD buffer (50 mM Tris/HCL-pH 7.8, 20 mM KCl, 5 mM MgAc, 0.5 mM DTT) for 0, 4 and 24 h at 37 °C. The reactions were stopped by acidification for cytokine detection via Western blot.

2. Annexin A1 (5 μM), α-Synuclein (7 μM), CaM (5 μM), EF-G (5 μM), EF-Ts (5 μM), Enolase1 (5 μM), Ffh (5 μM), H2A (5 μM), H2B (5 μM), H3 (5 μM), H4 (5 μM), HUWE1 (5 μM), IF2 (5 μM), LEDGF (5 μM), LRP130 (5 μM), Ovalbumin (5 μM), PDF (5 μM), RF1 (5 μM), tau (5 μM), UbcH7 (5 μM), Ube2K (5 μM), and Ube2S (5 μM) digestions were carried out by incubating the respective protein substrate with 20S standard proteasomes (from HeLa cell lines; 0.05–1 μM) in 20 μl HKM buffer (50 mM HEPES, pH 7.2, 20 mM KCl, 5 mM MgCl₂) for 0 and 24 h at 37 °C. The reactions were stopped by SDS loading buffer (for SDS-PAGE visualization).

The conditions used in (2) were then chosen as the optimal conditions to be applied in the 0–24 h kinetic digestions of the 15 proteins efficiently digested by 20S proteasomes. Thus, we performed the 0–24 h kinetic digestions of the following proteins: Annexin A1 (5 μM), α-Synuclein (7 μM), CaM (5 μM), Ffh (5 μM), H2A (5 μM), H2B (5 μM), H3 (5 μM), H4 (5 μM), IF2 (5 μM), IL-37b (5 μM), LEDGF (5 μM), RF1 (5 μM), tau (5 μM), Ube2K (5 μM), and Ube2S (5 μM). In these digestions, the amount of proteasome and substrate used can be recapitulated in the following proteasome to substrate molar ratio: Annexin A1 (1:5), α-Synuclein (1:35), CaM (1:100), Ffh (1:25), H2A (1:100), H2B (1:100), H3 (1:25), H4 (1:25), IF2 (1:25), IL-37b (1:5), LEDGF (1:25), RF1 (1:25), tau

(1:25), Ube2K (1:25), and Ube2S (1:25). The reactions were stopped by acidification (for MS measurement). All kinetics digestion experiments were performed with the same batch of 20S standard proteasomes (from HeLa cell lines) in two independent assays (biological replicates) and measured by MS2 (see below; in duplicate, i.e., 2 technical replicates for each biological replicate).

## Evaluation of protein degradation

The kinetics experiments involving the degradation of mouse IL-1α and IL-1β, human IL-1α, IL-1β and IL-37b in presence of purified 20S proteasomes were measured via Western blots as follows: in vitro degradation samples were separated in 12% polyacrylamide SDS gel and transferred on PVDF membrane (Bio-Rad). Membrane was blocked by 5% non-fat dry milk – TBST solution for 1 h at room temperature to prevent unspecific binding. Degraded substrates were detected by incubating the membrane with its respective primary antibody overnight at 4 °C: mouse IL-1α (1: 1000 Abcam), human IL-1α (1:1000 Abcam), mouse IL-1β (1: 1000 R&D Systems), human IL-1β (1: 1000 Cell Signaling Technology) and human IL-37b (1:1000 Abcam). 20S proteasome was detected using anti-proteasome α4 antibody (1: 1000 Santa Cruz Biotechnology). A secondary anti-mouse (1: 20,000 Thermo Fishers Scientific), anti-goat (1: 5000 Santa Cruz Biotechnology) or anti-rabbit (1: 10,000 Thermo Fishers Scientific) horseradish peroxidase-conjugate antibodies was incubated for 1 h at room temperature followed by ECL detection (Thermo Fisher Scientific).

Annexin A1, α-Synuclein, CaM, EF-G, EF-Ts, Enolase1, Ffh, H2A, H2B, H3, H4, HUWE1, IF2, IL-37b, LEDGF, LRP130, Ovalbumin, PDF, RF1, tau, UbcH7, Ube2K, and Ube2S degradation in kinetics experiments in presence of purified 20S proteasomes was monitored using SDS-PAGE and Coomassie blue staining.

For all cases where the outcome of the above-described assays was not conclusive, we measured the in vitro degradation samples via MS. We considered 'degraded proteins', all proteins from which we identified at least 300 peptide products in in vitro degradation experiments.

## In-gel digestion

Protein bands excised from the gel were first washed with water for 5 min and followed by a second wash with acetonitrile (ACN) for 15 min, at 26 °C, 1050 rpm. The gel pieces were then dried in a vacuum centrifuge for 5 min and incubated at 56 °C for 50 min in 100 µl of ammonium bicarbonate buffer containing 10 mM DTT. After the DTT reducing step, the solution was replaced with 150 µl of ACN and further incubated for 15 min. Next, the solution was replaced with 100 µl of ammonium bicarbonate buffer containing 55 mM iodoacetamide and incubated for another 20 min at 26 °C, 1050 rpm. Following that, the gel pieces were subjected to 3 washes. The first wash was done using 150 µl of ammonium bicarbonate buffer and the following two washes were performed using 150 µl of ACN, respectively. All washes were performed at 26 °C for 15 min, 1050 rpm. Subsequently, the gel pieces were dried again in a vacuum centrifuge before being rehydrated at 37 °C for 16 h in 100 µl of digestion solution (100 mM ammonium bicarbonate, 100 mM calcium chloride) containing 0.1 trypsin µg/µl. After the overnight tryptic digestion, a 3-step peptide collection was performed. The first and third steps were done using 50 µl of ACN, whereas the second step was carried out using 50 µL of 5% (v/v) formic acid. Each step required a 15 min incubation, at 37 °C, 1050 rpm before supernatant from the respective step was collected. The supernatant fractions collected from all steps were pooled and dried in a vacuum centrifuge. The dried peptide sample was resuspended in 2% ACN, 0.05% TFA prior to LC-MS measurements on Thermo Scientific Q Exactive mass spectrometer. The resulting raw files were searched against human proteomes using PEAKS v10.6 with a tolerance of 6 ppm for precursor mass and 0.03 Da for fragment ion mass.

## MS measurements and data processing

In vitro digestions were measured in duplicates (technical replicates) by either Thermo Scientific Q Exactive HF-X or Orbitrap Fusion mass spectrometers. Digested samples were subjected to SP3 beads clean-up prior to MS measurement, as described elsewhere[125]. Depending on the presence of cysteine residues in the amino acid composition of the protein substrate, additional steps of reduction and carbamidomethylation using 2 mM DTT and 4 mM iodoacetamide, respectively, were performed during the SP3 peptide clean-up. Briefly, during MS measurements, samples were injected using an Ultimate 3000 RSLC nano pump (both from ThermoFisherScientific). Peptides were loaded and separated by a nanoflow HPLC (RSLC Ultimate 3000) on a C18 nano column (30 cm length, 75 µm internal diameter). Peptides were eluted with a linear gradient of 5%–55% buffer B (80% ACN, 0.1% formic acid) at a flow rate of 300 nl/min for 88 min at 50 °C. MS data collection was done with Thermo Xcalibur and Thermo Xcalibur Instrument Setup v. 4.0.27.19 and Thermo Q Exactive HF Peripheral Devices v. 2.8 SP1 Build 2806 (for Q Exactive HF-X), or with Thermo Xcalibur and Thermo Xcalibur Instrument Setup v. 4.4.16.14 and Tune Application v. 3.4.3072.18 (for Orbitrap Fusion). To acquire MS data in a Data Dependent Acquisition mode, either Top 20 (for Q Exactive HF-X) or Top 30 (for Orbitrap Fusion) precursor ions were used. We acquired one full-scan MS spectrum at a resolution of either 60,000 (for Q Exactive HF-X) or 120,000 (for Orbitrap Fusion) with a normalized automatic gain control (AGC) target value of 1,000,000 and a scan range of 300–1600 m/z. The MS2 fragmentation was conducted using HCD collision energy (35%) with an orbitrap resolution of either 15,000 (for Q Exactive HF-X) or 30,000 (for Orbitrap Fusion). The AGC target value was set up at 100,000 with a max injection time of 128 ms. A dynamic exclusion of 30 s and either 1–6 (for Q Exactive HF-X) or 7 (for Orbitrap Fusion) included charged states were defined within this method.

Recalibrated tandem mass spectra were matched using MSFragger v3.7.0 with a 6 ppm and 5 ppm tolerance on precursor masses for Q Exactive HF-X and Orbitrap Fusion measurements, respectively. Mass tolerance of fragment ions was set at 0.02 Da and 20 ppm for Q Exactive HF-X and Orbitrap Fusion measurements, respectively. Precursor mass tolerance and mass tolerance of fragment ions used in the analysis were based on the performance and quality control run of the two MS equipment. Both MS instruments were successfully used for non-spliced and spliced peptide identification in previous studies (e.g.,[51]). For more details of inSPIRE 1.5 and MSFragger searches see Development and execution of inSPIRE 1.5 section.

The Specht[59], Paes[77] and Roetschke[60] datasets of synthetic polypeptide digestions with proteasomes were measured using either LTQ XL, Q Exactive Plus and Q Exactive Orbitrap or Fusion Lumos Orbitrap mass spectrometers, as described in the original studies. The database of non-spliced and spliced peptide products identified by applying invitroSPI to these datasets is described in[60] (see Source Data section).

## Generation of a contaminant database

To determine the contaminant database to be used in the peptide identification pipeline, protein substrates or 20S proteasomes were individually digested by trypsin (Promega) and chymotrypsin (Roche) in solution with the help of RapiGest (Waters) according to the manufacturer's recommended conditions. Briefly, the samples were reduced with DTT and carbamidomethylated with IAA in RapiGest solution. Subsequently, trypsin or chymotrypsin was added at a protease to substrate ratio of 1:20 to a final concentration 0.1% RapiGest solution and incubated at 37 °C for 16 h. Next, Trifluoroacetic acid (TFA) was added to a final concentration of 0.5% and incubated at 37 °C for 2 h followed by centrifugation at 13,000 rpm, for 30 min. The supernatant was then dried with SpeedVac and resuspended in 2% ACN, 0.05% TFA for MS measurement on Q Exactive HF-FR mass spectrometer. Recalibrated tandem mass spectra were matched using

PEAKS DB v10.6 against the respective expression host proteomes with a 6 ppm tolerance on precursor masses. The mass tolerance of fragment ions was set at 0.03 Da.

## InvitroSPI

All possible *cis*- and homologous *trans*-spliced peptides have been computed as described elsewhere[126]. Mascot search result files were exported and processed using the invitroSPI software[60]. All PSMs suggested by Mascot for a single MS2 scan were mapped to their potential origins in the substrate sequence (see Mapping of peptide sequences section), considering the mass redundancy of leucine (L) and isoleucine (I). PSMs were evaluated based on product type (spliced vs. non-spliced) and based on differences in ion scores to determine the most likely peptide sequence for a given scan as described in[59,60]. Briefly, in case the top-ranked candidate was a non-spliced peptide, the spectrum was assigned as such. In case the top-ranked candidate was a spliced peptide, the spectrum was only assigned if there was no lower-ranked non-spliced peptide with a similar ion score and no ambiguous lower-ranked spliced peptide. Additionally, all assigned PSMs had to have a Mascot ion score larger than 20 and a q-value smaller than 0.05. Ambiguous and low-quality scans were discarded. This hierarchical approach of giving preference to non-spliced peptides over spliced peptides accounts for the difference of database sizes between non-spliced and spliced peptides and increased the reliability of the identification[59,60].

As in our previous studies[59,60], for each substrate digestion, contaminants and peptide synthesis artifacts identified in control samples (either 0 h digestion time or samples with substrates and no proteasomes) were removed as follows: any non-spliced peptide identified in control samples was removed from the final list of identified non-spliced peptides. Any spliced peptide in the control samples, containing the same splice-site as an identified peptide (thus, either identified as such or identified as a longer precursor in control samples) was removed from the final list of identified spliced peptides.

## Development and execution of inSPIRE 1.5

In this study, the original inSPIRE method[72] was modified obtaining inSPIRE 1.5, and used for the identification of spliced and non-spliced peptides in in vitro digestions of proteins. inSPIRE was originally developed to improve PSM identifications in large search spaces, focusing primarily on HLA-I immunopeptidomes[72], and its performance has been tested in that setting with the iBench benchmarking software[100]. inSPIRE is based on the rescoring of PSMs, which is primarily achieved by comparing Prosit-predicted spectra and retention times[94,95] to the experimental values from the corresponding MS2 spectrum. The PSMs from the original search results are then evaluated during rescoring, using Percolator[127,128] with metrics describing the match between experimental results and Prosit predictions. The original inSPIRE also included as filtering steps some of the analytical strategies adopted by some groups[69,70,129] for the validation of the spliced peptides originally published by refs. 61-66,68.

In this study, MS2 spectra were searched three times with MSFragger v3.7.0[96] using a 5 ppm and 6 ppm tolerance on precursor masses for Q Exactive HF-X and Orbitrap Fusion measurements, respectively. Mass tolerance of fragment ions was set at 0.02 Da and 20 ppm for Q Exactive HF-X and Orbitrap Fusion measurements, respectively.

The first search was performed against the substrate protein with non-specific cleavage. Asparagine (N) and glutamine (Q) deamidation, methionine (M) oxidation and N-terminal acetylation were allowed as variable PTMs. Cysteine carbamidomethylation was set as fixed modification for those proteins that contained cysteine residues. Mass calibration was set to "ON and find optimal parameters" and the calibrated mzML files were written to disk.

The second MSFragger search was performed against the pre-calibrated mzML files with a custom database containing all theoretical spliced peptides with no in silico enzymatic cleavages were allowed. This search was carried out with the same mass tolerance and fixed modification settings as the first search but with methionine (M) oxidation as the only variable modification.

The third search was also carried out using the pre-calibrated mzML files. In this case, three custom databases were searched to determine contaminants: (i) a database containing substrate-specific contaminants (see description above), (ii) a database containing 20S proteasome contaminants and (iii) the substrate sequence. All settings were kept consistent with the first search.

The inSPIRE 1.5 provided a number of upgrades compared to the original 1.0 release[72]. The original release focused primarily on the identification of non-spliced peptides in immunopeptidomics with no specialized functionality for spliced peptide identification. inSPIRE 1.5 expanded on the original functionality, primarily for spliced peptide identification but the additional compatibility with MSFragger adds utility for any use case.

For this project, pepXML output files containing putative non-spliced and spliced PSMs from the first two MSFragger searches described in the previous section were passed to inSPIRE 1.5. The PSMs from each search (both target and decoy) were labeled as either spliced or non-spliced. If a peptide could be generated by either hydrolysis or splicing reactions, then only the non-spliced PSM was considered by inSPIRE 1.5. As in the standard execution, peptides containing Asparagine (N) and glutamine (Q) deamidation or N-terminal acetylation were removed from the search results since Prosit was not trained on peptides carrying these PTMs. However, if the PSM was from a non-spliced peptide with these modifications and its MSFragger hyperscore was within 30 % of the best scoring spliced PSM for that MS2 spectrum, then all PSMs for that MS2 spectrum were removed before rescoring.

Prosit prediction of MS2 spectra and indexed retention times for the putative peptides of the remaining PSMs was then performed within inSPIRE 1.5. Prosit indexed retention time (iRT) predictions were fit to observed retention times using linear regression. The regression model was fit on the non-spliced PSMs with the highest spectral angles (>0.9) using 10-fold cross-validation.

The inSPIRE 1.5 also used the origin of a peptide in rescoring, i.e., Percolator obtained information on the spliced or non-spliced character of a given candidate. This information enabled Percolator to penalize spliced PSMs in an automated manner (Supplementary Fig. 3). This allowed the Percolator algorithm to apply more stringent requirements for PSMs assigned to spliced peptides than it did for PSMs assigned to non-spliced peptides, and hence to account for the difference in stratum sequence search spaces (Fig. 3d, Supplementary Fig. 10, Fig. 4a). Compared to inSPIRE 1.0, the number of features used for rescoring was strongly reduced for the spliced peptide use case to simplify the model and increase interpretability. Spearman correlation between Prosit-predicted MS2 spectrum, Prosit spectral angle and the absolute error between observed and predicted iRTs were employed as feature set in inSPIRE 1.5. The conversion back to iRT error enabled rescoring of RAW files together which could have varying retention time gradients.

The smaller feature set and use of iRT error as a feature also enabled rescoring of PSMs from all protein digestions in a single model. This choice was motivated by the desire to improve performance of the model with access to more data and to ensure consistent identification thresholds across all proteins (as observed in Supplementary Fig. 10). We also updated inSPIRE so that the final identifications could be exported using peptide level probabilities rather than PSM level probabilities (this is configurable in inSPIRE 1.5). This was a particularly important update for this use case where the large number of high-confidence PSMs for a single unique non-spliced peptide led to peptide level probabilities being dramatically more stringent. The peptides exported at a 1% FDR were considered for further analysis. All

cognate PSMs for the assigned peptides that passed the 1% PSM-level FDR threshold, were used for quantification.

As a final step of validation in inSPIRE 1.5, we further filtered the assignments to consider non-spliced peptides not identified by the search engine and possible contaminants. For each assigned spliced peptide, all possible isobaric non-spliced peptides were rescored using the weights from the trained Percolator model. The spliced peptide was then removed from the final list of identifications if the competing non-spliced peptide achieved a higher score. The remaining spliced peptides that were assigned in the presence of an isobaric competitor were significantly superior to their best isobaric competitor in terms of MS2 spectra and iRT error (Supplementary Fig. 11). In order to deal with possible contaminants, Prosit spectral prediction was performed for the putative PSMs from the search of the contaminant database (the third MSFragger search detailed above). PSMs that could be better explained by a contaminant (evaluated by comparing Prosit spectral angle between PSMs) were removed from the final list of assignments.

### inSPIRE 1.5 FDR estimation for spliced and non-spliced peptide products

In the inSPIRE 1.5 method, we took several steps for a robust FDR estimation. Firstly, although inSPIRE 1.5 searched each protein digestion dataset separately, we combined all datasets from the 15 digested proteins (17 total digestions given the 3 experiments conducted on Ffh) for the final rescoring within Percolator, thereby allowing greater diversity in the final training. Additionally, we greatly reduced the features used in the inSPIRE 1.5 workflow for this use case (see above). This approach reduces the risk of overfitting that could result from the smaller variation in a single sample.

Furthermore, by using the origin of a peptide as a feature in the inSPIRE 1.5 rescoring we could adjust both the score of a PSM and the FDR estimation for spliced and non-spliced peptides (see the feature weightings in Supplementary Fig. 3). The effect of this adjustment could be seen by the fact that even though the PSMs of spliced peptides were superior in both MS2 spectral quality and retention time prediction (Fig. 3d), they were estimated to have a higher FDR (Supplementary Data 2) of 3.2% compared to an FDR of 0.7% for non-spliced peptides (Supplementary Data 2). These group FDR estimates for spliced and non-spliced peptides were calculated based on the posterior error rate probabilities for spliced and non-spliced peptides (again including adjustment based on the origin of the peptide).

We validated our approach by comparing the standard of our identifications with those identified by ref. 95 in their immunopeptidome analysis using the original Prosit rescoring pipeline (Supplementary Data 2). In that study, a database of 85,919 proteins was searched with 'unspecific cleavage' and assigned non-spliced peptides' PSM had a mean spectral angle (comparison of experimental with Prosit-predicted MS2 spectra) of 0.83[95]. In our study, we observed a similar spectral angle mean for the spliced (0.84) and non-spliced (0.80) peptides' PSMs. This further suggest that the size of the search database was efficiently handled by developing inSPIRE 1.5, with the smallest database of non-spliced and spliced peptide candidates from a single protein having similar spectral angle mean than non-spliced candidates from a massively larger pool of proteins as it is for any immunopeptidome analysis.

### Release of iBench 2.0 for benchmarking of single protein digestion with purified proteasome

iBench 1.0 was primarily developed to benchmark methods for the identification of spliced and non-spliced peptides in immunopeptidomics[100]. In the ideal case, very high confidence PSMs from synthetic peptide measurements could be used as a source for a ground truth dataset (MS2 measurements where the true peptide was known with confidence). The relevant peptides could then be embedded in an in silico only proteome either as full peptide sequences

(mimicking canonical non-spliced peptides) or as two fragments (mimicking the splice reactants of a spliced peptide). The MS2 data can then be reanalyzed with a method to be benchmarked using the modified proteome and performance analyzed. The key read outs are the precision (number of correct identifications divided by the total number of PSMs identified) and the recall (number of correct identifications divided by the total number of possible PSMs that could have been identified) at a method's scoring threshold. Precision-recall curves are generated separating the performance for non-spliced, *cis*-spliced, and (if applicable) homologous *trans*-spliced peptide identifications.

Embedding non-spliced and spliced peptides into a single protein would be a significantly greater computational challenge as it would be near impossible to avoid overlap between peptides without artificially increasing the protein length to be out of touch with reality. Hence, for imitating a single protein digestion, we used a pseudo ground-truth dataset where invitroSPI identifications of spliced and non-spliced peptides in much shorter polypeptide digestions were used. The polypeptide sequences were then concatenated to generate the in silico-only modified protein sequence (Supplementary Fig. 5a). This application came with a major caveat, since the invitroSPI FDR for *cis*-spliced peptides was approximately 4.2% and 6.8% for homologous *trans*-spliced peptides[60]. In order to reduce the risk of misassignments only PSMs with a spectral angle greater than 0.7 were considered, and PSMs assigned to spliced peptides for which an isobaric non-spliced peptide existed were not considered. The polypeptides used to this end were 24–34 residues long and each (40 µM) was digested by 3 µg 20S proteasomes in 100 µl TEAD buffer (Tris 20 mM, EDTA 1 mM, NaN3 1 mM, DTT 1 mM, pH 7.2) at 37 °C and measured by MS (see Source Data Section for details).

As a further in silico experiment to evaluate the precision of inSPIRE 1.5 in a ground-truth dataset that only contained non-spliced peptides, we used a dataset of synthetic peptide library measured by Orbitrap Fusion Lumos mass spectrometer coupled to an Ultimate 3000 RSLC nano pump (both from Thermo Fisher Scientific). Originally, peptides were grouped in eight library batches, with each peptide measured at the concentration of 0.0625 pmol/µl. For each pool, 8 µl were injected in the instrument, thereby measuring 500 fmol of each peptide. For the initial identification of the synthetic peptides used in the synthetic peptide library, we searched RAW files using PEAKS, version 10.6 with precursor mass tolerance of 5 ppm and fragment ion mass tolerance of 0.02 Da. No PTMs were allowed, and the results were exported at an FDR of 1%[72]. We previously used this dataset for developing iBench 1.0[100], benchmarking inSPIRE 1.0 performance in the non-spliced peptide identification[72] and predicting the success of Fmoc-based peptide synthesis[101]. The synthetic peptide library contained 9, 10, or 15 amino acid long peptides ($n = 6876$ unique peptide sequences identified with PEAKS DB) related to CD4[+] and CD8[+] T-cell response to viruses[100]. This library provided a large quantity of very high confidence PSMs and also many peptide sequences that were either substrings of another peptide or contained a significant overlap with another peptide. This latter aspect allowed us to create an in silico-only protein sequence of length 911 residues that contained the sequence of 390 of those synthetic peptides as fully contiguous sequences, which could be used as constructed reference database (Supplementary Fig. 6a). The *in-silico*-only protein length was slightly longer than our longest protein investigated in the study (i.e., IF2, which was 890 residues long), which represented the most challenging circumstances to inSPIRE 1.5 for spliced peptide identification. Indeed, the longer the length of a protein, the larger the number of spliced peptides in the search space. By applying iBench 2.0, we selected the MS2 spectra of those 390 synthetic peptides in the original MS files and generated a constructed ground-truth MS dataset containing only the MS2 spectra of those 390 synthetic peptides. Since they were present as contiguous peptide sequences in the constructed reference

database, they should only be identified as non-spliced peptides by a perfect method.

The constructed ground-truth dataset was then analyzed using MSFragger 3.7 and inSPIRE 1.5 (using an inSPIRE 1.5 model trained on all protein digestion data) using the same precursor mass tolerance and fragment ion mass tolerance settings of the original analysis, and performance measured via PR curve (Supplementary Fig. 6b).

## Development of de novo tools and comparison with inSPIRE 1.5

Since no specialized tools exist for de novo-based identification of spliced peptides in in vitro protein digestions, we developed two simple methods for comparison with inSPIRE. The two de novo methods were motivated by the methods initially proposed by Faridi and colleagues[61] and Mylonas and colleagues[69] with adaptations to in vitro protein digestions. In both cases non-spliced peptides were identified via a standard PEAKS DB search and de novo candidates for spectra not assigned to non-spliced peptides at 1% FDR were mapped to the substrate protein to find possible spliced peptides. These spliced candidates then made up the database for a second PEAKS DB search for spliced peptide sequence assignments (again a 1% FDR cut-off was used). The two methods differed in the cut-offs on peptides in the second PEAKS DB search. For the DN5 method, the top 5 candidates per spectrum with ALC > 50% were considered and checked as spliced peptide candidates for the final database. For the DN1 method, only the top candidate per spectrum with ALC > 80% was considered. The intention of this approach was to provide a more sensitive de novo method (DN5) and a more stringent de novo method (DN1) for comparison with inSPIRE 1.5.

For application of inSPIRE 1.5 in this benchmarking, we reused the model weights from the Percolator model which was used for all assignments in this study.

## inSPIRE 1.5 - aSPIRE pipeline, the semi-quantification and final identification of the peptide products

Label-free quantification of peptide products identified by inSPIRE 1.5 using MS1 ion peak area was performed using the aSPIRE framework. Although this method is not applicable for single peptides[40,103], it is well accepted when analyzing larger proteomics datasets[66,68,77,103,130,131]. The potential bias of this method due to differences in the chemical features of spliced compared to non-spliced peptides has been previously excluded[66].

All PSMs assigned by inSPIRE were quantified using Skyline v22.2.1.417[132] upon removal of contaminants. Contaminants in proteins and proteasome samples were tested by chymotrypsin-trypsin digestions of the substrates alone, and the cognate protein list was used to create a contaminant peptide database used hereafter (see above). After this filtering step, PSM tables were imported in ssl format. A spectral library was built using the BlibBuild function in the BiblioSpec software. Skyline was run in via command line using a Docker image, thereby importing the ssl file, the spectral library and a fasta file with all unique peptide sequences identified. Variable and fixed modifications were imported from the ssl file and ambiguous matches were included. Empty proteins were kept. Refinements were set to auto-select peptides, transitions and precursors, lower boundaries of detections as well as peptide and transition numbers were set to 0. The minimum peak found ratio was set to 0 and the maximum peak found ratio was set to 1. Minimum ion peak area to library spectrum dot products and MS1 precursor peak area to expected isotope distribution dot products were set to 0. Allowed product ion types to generate product ion transitions were y and p. The retention time filter tolerance for a full scan was set to 0.5. Precursor charges 1–6 were allowed, using an Orbitrap precursor mass analyzers at a resolving power of 70,000 at 400 m/z. High selectivity extraction was realized in a retention time window of 0.5 min compared to MS/MS IDs. No enzymatic cleavages were allowed. A maximum of two variable modifications

(C carbamidomethylation, N and Q deamidation, M oxidation or N-terminal acetylation) and two losses were allowed. Peptides of length between 5 and 40 amino acids could be quantified using MS1 intensities without further normalization. MS1 ion peaks were transformed using Savitzky-Golay Smoothing.

The peptide intensities were obtained by summing the total area of MS1 and the total background area over all charges of an identified peptide precursor (MS1 spectrum), including all detected isotopes. Substrate-derived contaminants, which were peptides present in the substrate solution prior to proteasome processing, were removed based on the sequence identity and generation kinetics of identified peptides. All spliced peptides that were identified in the control measurements (0 h time point and no proteasome controls) as such, and as precursors of peptides identified at later time points were removed from the final peptide list, similarly to the approach implemented in invitroSPI (see above). All non-spliced peptides that were identified in the control measurements were, however, only removed from the final peptide list in case a higher MS1 intensity was detected at 0 h time point than at any later time points.

The so obtained quantitative kinetic of peptide generation comprises MS1 intensities at 0, 1, 2, 4 and 24 h of digestion time for all proteins except α-Synuclein. For α-Synuclein, the kinetic comprises 0, 0.25, 0.5, 1, 3, 6 and 24 h. Additionally, data points representing background noise were removed and kinetics were further normalized as follows. A background noise value was obtained for each peptide by extracting the maximum intensity measured at 0 h across all biological and technical replicates. The median of this background noise distribution across all peptides reflected a global cut-off. For a given peptide, all intensity values measured that are smaller than the background noise value for that peptide or than the global cut-off were removed from a kinetic. In all other cases, the background noise value (the maximum of peptide-specific and global background noise value) was subtracted from the measured intensity for each biological/technical replicate. The mean intensity for each time point and biological replicate across technical replicates was reported and is available in the 'aSPIRE output.zip' file of the inSPIRE 1.5 – aSPIRE output in the online repository (see Source Data section).

Peptides that had a missing value, i.e., zero MS1 intensity, after four hours (three hours for α-Synuclein) of digestion time were removed from the final set of quantified peptides.

## Peptide synthesis for validation of assignments

All synthetic peptides used for MS2 spectrum comparison were synthesized using Fmoc solid phase chemistry. The peptides were randomly selected among groups with various features such as the most abundant non-spliced and spliced peptides, cis-spliced peptides either containing a splice-reactant less than 3 residues in length or a splice-reactant that was found as a non-spliced peptide (Supplementary Data 5 for non-spliced peptides, n = 10; Supplementary Data 6 for spliced peptides, n = 22). In each case, we showed a pair plot of the experimental MS2 spectra compared to the synthetic peptide's MS2 spectra, the experimental MS2 spectrum compared to the Prosit-predicted MS2 spectrum, and the synthetic peptide's MS2 spectrum compared to the Prosit-predicted MS2 spectrum. Additionally, for two representative cases where a spliced peptide was clearly identifiable despite the existence of an isobaric non-spliced peptide, we synthesized the spliced peptide and the isobaric non-spliced peptide and compared an experimental MS2 spectrum from which the spliced peptide was identified to the synthetic peptide spectra (n = 2, Supplementary Data 7). Synthetic peptide nomenclature and cognate info are reported in Supplementary Data 10.

## Mapping of peptide sequences
The mapping of peptide sequences was performed as described elsewhere[60]. Briefly, peptide sequences were mapped to a substrate

sequence by exact string matching of the complete peptide product sequence. If this was not possible, the peptide product sequence was split into two splice-reactants at each possible position and the splice-reactants were matched to the substrate sequence. All valid combinations of splice-reactants were kept. Spliced peptides generated by ligation of three or more fragments were not allowed and therefore are not included in our database.

If a peptide sequence could be explained by multiple locations, all locations were reported in the final database. A sequence was assigned as homologous *trans*-spliced only if none of the locations for a given sequence suggested a *cis*-spliced peptide. If a sequence could be explained by both forward and reverse *cis*-splicing, it was denoted as forward/reverse *cis*-spliced peptide, also known as multi-mapper peptides.

### Prediction of MS2 spectra and comparison with measured MS2 spectra

MS spectrum comparison was carried out as described elsewhere[60].

Briefly, Prosit version 2020[94,95] predicts the MS2 spectra given a peptide sequence, precursor charge and calibrated collision energy. A predicted MS2 spectrum can be compared to the detected MS2 spectrum by computing a similarity score. In this study, we used the spectral angle between the L2 normalized spectra, also known as normalized spectral contrast angle[133], which ranges from 0 (very bad match between MS2 spectra) to 1 (perfect match between MS2 spectra). The spectral angle consists of a transformation on the normalized dot product and is the loss metric on which Prosit was trained.

In the comparison of the MS2 spectra measured by MS and predicted by Prosit, we excluded PTM-labeled peptides, peptides shorter than 7 and longer than 30 amino acids. Indeed, Prosit cannot predict MS2 spectra of PTM-labeled peptides[94], and was trained only on carbamidomethylated C. Furthermore, Prosit performance is influenced by peptide length[60,94]. Therefore, all spectral angle comparisons were carried out on peptides shorter than 13 amino acids. Spectral angles of PSMs derived from synthetic polypeptides ($n = 25$) were derived from high-precision MS instruments only as previously described in[60].

### Generation and analysis of simulated background databases of peptide products as controls

In order to identify proteasome specificities, a simulated background database containing a subset of all theoretically possible spliced and non-spliced peptides was generated. This simulated background database was then compared to the database of identified peptide products. Thereby, we could verify whether the identification of spliced and non-spliced peptide characteristics (e.g., splice-reactant, intervening sequence and peptide length, as well as amino acid frequencies) arose from theoretical database structure – and thus were potential analysis artefacts - or from biochemical drivers of the catalytic reaction[59,60].

Briefly, a qualitative simulated background database of spliced peptides was obtained by generating a random set of spliced peptides via repeated sampling with replacement of four positions within the range of the substrate length that denoted the first ($i, j$) and second ($k, n$) splice-reactant. In case the four positions ($i, j, k$ and $n$) form a valid spliced peptide, the sample was accepted. Otherwise, it was rejected. This strategy was repeated until the desired number of random sequences was reached. Here, a qualitative simulated background database size, which was 50-times the size of the database of identified spliced peptide sequences, was chosen. A qualitative simulated background database of non-spliced peptides was obtained by generating all possible non-spliced peptides of a given substrate and sampling from this set. All simulated peptide sequences were re-mapped to the protein sequence of origin to account for potential multi-mapping. For both spliced and non-spliced peptides, only peptides that correspond to the length restriction of inSPIRE 1.5 were considered, to reflect the peptide identification bias in the simulated background databases.

A quantitative simulated background database was generated as follows. The qualitative simulated background database was split into the different product/spliced types. Each sequence in the simulated background database was then assigned a quantity that was sampled uniformly in the range of intensities of detected peptides of the same product/splice type after four hours of digestion (three hours for α-Synuclein).

### Computation of the theoretical sequence space size

The number of possible unmodified spliced and non-spliced peptides that could be derived from a protein sequence in sequence-agnostic fashion constituted the size of the theoretical peptide sequence space[60]. The number $X$ of non-spliced peptides of length $N$ that could theoretically arise from a substrate of length $L$ was:

$$X_{non-spliced} = L - N + 1 \tag{1}$$

To derive the positions of all spliced peptides, we defined four indices $i, j, k$ and $n$ that denoted the first ($i, j$) and second ($k, n$) splice-reactant, respectively. The corresponding number of peptides was calculated via summing over interval ranges that form valid spliced peptides. *Cis*-spliced peptides can be formed via forward or reverse ligation. The number of all forward *cis*-spliced peptides of length $N$ that could theoretically arise from a substrate of length $L$ was:

$$
\begin{aligned}
X_{fwd.cis-spliced} &= \sum_{i=1}^{L-N} \sum_{j=i+L_{ext}-1}^{N-L_{ext}+i-1} \sum_{k=j+2}^{L-N+j-i+2} 1 \\
&= \frac{1}{2}(N - 2L_{ext} + 1)(L - N)(L - N + 1)
\end{aligned}
\tag{2}
$$

$L_{ext}$ denoted the minimal splice-reactant length and was set to 1 per default. Analogously, the number of theoretically possible reverse *cis*-spliced peptides was calculated as:

$$
\begin{aligned}
X_{rev.cis-spliced} &= \sum_{k=1}^{L-N+1} \sum_{n=k+L_{ext}-1}^{N-L_{ext}+k-1} \sum_{i=j+1}^{L-N+n-k+2} 1 \\
&= \frac{1}{2}(N - 2L_{ext} + 1)(L - N + 1)(L - N + 2)
\end{aligned}
\tag{3}
$$

To calculate the number of theoretical homologous *trans*-spliced peptides in an in vitro scenario where a single protein was digested with purified proteasomes, the following formula was used:

$$
\begin{aligned}
X_{trans-spliced} = &-1 + \frac{2}{3}L_{ext}^3 + L_{ext}^2(-1 - N) + \frac{5}{6}N + N^2 - \frac{5}{6}N^3 \\
&+ L\left(-1 + L_{ext}(2 - 2N) + N^2\right) + L_{ext}\left(\frac{7}{3} - 3N + 2N^2\right)
\end{aligned}
\tag{4}
$$

### Computation of substrate-specific cleavage and splicing strengths

SCS-$P_1$ and PSP-$P_1$ were calculated using the MS1 intensities (quantities) of peptides after four hours of digestion. For α-Synuclein, the MS1 intensity after three hours of digestion was taken. In the case of a multi-mapper peptide, i.e., a peptide that maps to several substrate locations (see above), the intensity value of this peptides was adjusted by the number of possible peptide origins.

For each amino acid residue of a given substrate sequence, the summed quantity of all non-spliced peptides containing this residue at their $P_1$ position was denoted as site-specific cleavage strength (SCS-$P_1$). Similarly, the summed intensity of all spliced peptides carrying this residue at the C-terminus of their first splice-reactant (P1) was denoted as site-specific splicing strength (PSP-$P_1$) (Supplementary Fig. 15). Cleavage and splicing strength values for a given residue were then

normalized by their respective sums in a local, 29 amino acid long, window surrounding the current residue, resulting in percentage distributions of SCS-$P_1$ and PSP-$P_1$ over the substrate sequence.

## Protein feature prediction and representation in HLA-I immunopeptidomes

Prediction of structural disorder was done using SLIDER[88] (http://biomine.cs.vcu.edu/servers/SLIDER/), which predicts the presence of long disordered segments, and IUPred3[89,134] (http://iupred3.elte.hu), which predicts the location and presence of disordered protein segments. A SLIDER score larger than 0.538 was chosen as indicator that a given protein had a long disorder segment, in agreement with[25]. An IUPred3 score of 0.4 was used as a cut-off value for a disorder residue[25,89,135,136]. The ratio between either intrinsically disordered region lengths and the whole protein length was computed accordingly (Fig. 2b). Secondary structure elements were predicted using the DECIPHER R package[137].

To compute the number of proteins containing disordered regions in HLA-I immunopeptidomes we again predicted the SLIDER score for every protein in the UniProt human database. We then downloaded all human peptides from HLA-I ligand assays from IEDB and filtered for peptides annotated to bind either HLA-A* or HLA-B* alleles. These peptides were then matched to their source protein with its corresponding SLIDER score. The antigens from the UniProt human database were then separated into 3 groups of proteins; proteins from which peptides were identified from HLA-A* expressing cell lines, proteins from which peptides were identified from HLA-B* expressing cell lines, and proteins from which no peptide was identified for either allele in the IEDB.

## Generation of amino acid logo plots

The computation of quantitative amino acid logo plots based on identified and quantified peptides (Fig. 6d, e) was achieved as follows. The intensity of peptides detected after four hours of digestion time (three hours for α-Synuclein) was extracted, where the intensity of peptides with more than one possible substrate origin (multi-mapper peptides, see above) has been adjusted by the number of possible substrate origins. For each amino acid at each position ($P_6$ to $P_{6'}$, also see Fig. 1a–d and Fig. 6d, e) the $\log_{10}$-transformed intensities of all identified spliced/non-spliced peptides that carried the given amino acid at the given position were summed. Subsequently, the obtained quantities were normalized by the sum over each position, thereby yielding a position-weight-matrix (PWM). The same procedure was repeated on the quantitative simulated background database (see description above). Finally, the quantitative amino acid usage plots were obtained by calculating the Jensen-Shannon (JS) divergence between $P$ and $Q$ following the example in ref. 138:

$$D_p = \frac{1}{2}\sum_a P_{a,p} \cdot \log_2\left(\frac{P_{a,p}}{M_{a,p}}\right) + \frac{1}{2}\sum_a Q_{a,p} \cdot \log_2\left(\frac{Q_{a,p}}{M_{a,p}}\right) \quad (5)$$

where $D_p$ is the height of each position $p$ $P_6$ to $P_{6'}$. $M$ denotes the mixture distribution of $P$ and $Q$ and was derived as follows:

$$M = \frac{1}{2}(P + Q) \quad (6)$$

The height of each amino acid at a given position $r_{a,p}$ was calculated as:

$$r_{a,p} = \begin{cases} 0 \, \forall \, P_p = Q_p \\ \frac{P_p - Q_p}{\sum_a |P_{a,p} - Q_{a,p}|} \, \forall \, P_p \neq Q_p \end{cases} \quad (7)$$

Finally, the height $D_{a,p}$ of a given amino acid $a$ at a given position $p$ was calculated as:

$$D_{a,p} = D_p \cdot r_{a,p} \quad (8)$$

In Fig. 6d, $P$ denotes the peptide product PWM and $Q$ denotes the simulated background PWM, while in Fig. 6e, $P$ denotes the non-spliced peptide product PWM and $Q$ denotes the spliced peptide product PWM. Amino acid logos were then visualized using the *ggseqlogo* R package[139].

## Calculation of coverage profiles and comparison of peptide sequence motifs inside/outside hotspot regions

Peptide hydrolysis and splicing coverages were obtained by extracting the intensities of peptides detected after 4 h of digestion (3 h for α-Synuclein). The intensity of peptides with more than one possible substrate origin (multi-mapper peptides, see above) was adjusted by the number of possible substrate origins. For each substrate residue, the intensities of all peptides that contained this residue were $\log_{10}$-transformed and then summed. The quantity distribution across all substrate residues was then min/max-scaled and smoothed using R's smooth.spline() function[140], and min/maxed-scaled for a second time.

We next assessed the probability of an amino acid occurring at a specific position provided that the position is located in inside or outside a hotspot region. For that, we extracted the normalized coverage values of all substrate residues that were detected as $P_1$ of non-spliced/spliced peptides for the analysis of non-spliced peptides and the N-terminal splice-reactant of spliced peptides. For spliced peptides with multiple sequence origins (multi-mappers), all potential origins were kept. The 50% of those residues with the highest coverage were denoted as hotspot residues, whereas the other 50% were denoted as non-hotspot residues. We then extracted the amino acids in the window surrounding the respective hotspot/non-hotspot residues ($P_6$ to $P_{-6}$). The probability of an amino acid at a specific position $R$ (e.g., A at $P_1$) provided that this position was located in a hotspot region $P(R|H)$ was calculated according to Bayes rule:

$$P(R|H) = \frac{P(H|R) \cdot P(R)}{P(H)} \quad (9)$$

$P(R)$ is the qualitative amino acid-position weight matrix containing the frequencies of all amino acids at all positions surrounding the detected $P_1$ positions. $P(H)$ is a scalar and denotes the fraction of hotspot residues among all detected $P_1/P_{1'}$ residues ($P(H)=0.5$). $P(H|R)$ is the ratio of summed coverages for a given $R$ inside hotspot regions compared to all regions. $P(R)$ and $P(H|R)$ are matrices that were normalized so that the sum over each position is 1. Analogously, we calculated the probability for all amino acids/positions outside hotspot regions:

$$P(R|\bar{H}) = \frac{P(\bar{H}|R) \cdot P(R)}{P(\bar{H})} = \frac{P(\bar{H}|R) \cdot P(R)}{1 - P(H)} \quad (10)$$

where $P(\bar{H}|R)$ is the ratio of summed coverages for a given $R$ outside hotspot regions compared to all regions. The amino acid logos displayed in Fig. 7c, e were obtained by calculating the JS divergence (see above) between $P(H|R)$ or $P(R|\bar{H})$ and their respective randomized backgrounds. Latter were obtained by randomizing coverage values while maintaining the substrate sequence order. The residues mapping to the upper 50% of the coverage values were denoted as hotspot and the lower 50% as non-hotspot regions. This procedure was repeated 100-times to yield randomized $P(R|H)_{rnd}$ and $P(R|\bar{H})_{rnd}$. In Fig. 7d, JS divergence between $P(R|H)$ for non-spliced peptides and $P(R|H)$ for

N-terminal splice-reactants was computed. Accordingly, in Fig. 7f, JS divergence between $P(R|H)$ for non-spliced peptides and $P(R|\bar{H})$ for N-terminal splice-reactants was computed. The amino acid logos displayed in Fig. 7g were obtained by calculating the JS divergence between $P(R|H)$ and $P(R|\bar{H})$ for non-spliced peptides and N-terminal splice-reactants, respectively.

The frequency of $P_1$ residues in hotspot/non-hotspot regions (Fig. 7a) was calculated as follows. Taking substrate residues that were detected as $P_1$ of non-spliced and spliced peptides, they were assigned to hotspot/non-hotspot regions based on their corresponding coverage as described above. The hotspot residue with the lowest coverage value was used as a cutoff. The subset of the entire substrate sequence that had a coverage of this threshold or higher was denoted as hotspot sequence stretch. Vice versa, the remainder of the substrate sequence was denoted as non-hotspot sequence stretch. The number of $P_1$s of non-spliced and spliced peptides inside and outside hotspot regions were divided by the length of the respective hotspot/non-hotspot sequence stretch.

The substrate amino acid frequency (Fig. 7b) was obtained by splitting the substrate sequence into hotspot and non-hotspot regions based the joint coverage with spliced and non-spliced peptides: the 50% of the residues with the highest coverage were assigned to the former, and the 50% of the residues with the lowest coverage to the latter, regardless of whether these residues were detected as $P_1$ of any peptide products. JS divergences were calculated relative to the overall substrate amino acid frequency.

### Statistical analysis

All statistical tests have been done in R or Python. To identify significant difference between groups, a Wilcoxon test was applied. A $p$ value < 0.05 was considered as statistically significant. For correlation between features Pearson's product moment correlation coefficient (PCC) was computed. Test for association between paired samples was performed in Supplementary Fig. 8 with resulting $p$ values obtained via algorithm AS 89, on Fisher-transformed PCC. All kinetics digestion experiments were performed with the same batch of 20S standard proteasomes (from HeLa cell lines) in two independent assays (biological replicates) and measured by MS2 in duplicate, i.e., 2 technical replicates for each biological replicate.

### Software

Figures postprocessing was done with Adobe Illustrator v27.1. Picture postprocessing was done with Adobe Photoshop v23.2.1. Analyses were carried out in R v4.1.1 and Python v3.9.

MS analysis was done using MSFragger v3.7.0, inSPIRE v1.5, Skyline v21.2, aSPIRE v1.0, Mascot v2.7.0.1, invitroSPI v1.0 and PEAKS v10.6. Benchmarking and ground-truth dataset generation was done by applying iBench 2.0. IDP features were computed with the online software SLIDER (http://biomine.cs.vcu.edu/servers/SLIDER/#References) and IUPred3 (https://iupred3.elte.hu/).

For in-house software, see Code Availability section.

### Data availability

The Roetschke database[60] of identified spliced and non-spliced peptide produced by 20S proteasomes in in vitro digestions of synthetic polypeptides has been deposited in the Figshare repository[141]. The cognate MS files are available at the PRIDE repository[142] with the dataset identifier PXD016782[59], PXD021339 and PXD025893[77], and PXD025995[60]. The MS files of the 15 protein digestions are available at the MassIVE online repository with the dataset identifier MSV000092631. The inSPIRE 1.5 – aSPIRE outputs are available at the MassIVE online repository with the dataset identifier MSV000092631. The MS files of the synthetic polypeptide digestions used for iBench 2.0 benchmarking are available at the PRIDE repository with the dataset identifier PXD044451. The MS files of the synthetic peptides, used for iBench 2.0 benchmarking of inSPIRE 1.5 in a spliced peptide-free ground-truth dataset, are available at the PRIDE repository with the dataset identifier PXD031812. The search files and inSPIRE/iBench output files are available at the MassIVE online repository with the dataset identifier MSV000092631. The Supplementary Data 1–10 are provided with this paper and are summarized in the file 'Description of Additional Supplementary Files'. Source data are provided with this paper.

### Code availability

The algorithm generating all possible *cis-* and homologous *trans-*spliced peptides was originally described by Liepe et al.[126]. InvitroSPI method is available on GitHub (https://github.com/QuantSysBio/invitroSPI)[143]. The inSPIRE software has been implemented with Python and is available on GitHub (https://github.com/QuantSysBio/inSPIRE)[86]. The aSPIRE software is available on GitHub (https://github.com/QuantSysBio/aSPIRE)[87].

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

## Acknowledgements

We thank: Gesellschaft fuer wissenschaftliche Datenverarbeitung mbH Goettingen (GWDG) for support and access to the GWDG GPU-cluster; the MPI Mass spectrometry core facility for the support and infrastructure; M. Wegstroth (MPI-NAT) for technical help with the preparation of α-Synuclein; L. L. Köpping (MPI-NAT) for assistance in aspects of the invitroSPI execution; M. Rodnina and I. Wohlegmuth (MPI-NAT) for providing the purified proteins PDF, RF1, EF-G, EF-Ts and Ffh; S. Lorenz (MPI-NAT) for providing the purified proteins HUWE1, UbcH7, Ube2K and Ube2S; H. Hillen and F. Finke (University Medicine Göttingen and MPI-NAT) for providing the purified protein LPR130; A. Stützer (MPI-NAT) for providing the purified proteins H2A, H2B, H3 and H4. This work was financed in part by: Cancer Research UK [C67500/A29686], National Institute for Health Research (NIHR) Biomedical Research Centre based at Guy's as well as St Thomas' NHS Foundation Trust and King's College London and/or the NIHR Clinical Research Facility, Blood Cancer UK [Ref. 22009], CRUK City of London Centre (CoL) Award and CRUK-CoL development fund [CTRQQR-2021/100004] to M.M.; ERC-StG 945528 IMAP to J.L.; MOE Tier 2 grant from Ministry of Education of Singapore (MOE2017-T2-2-099) to H.L.; the Max Planck Society to J.L., H.U., A.C. and S.B. W.T.S. is supported by the European Union's Framework Programme for Research and Innovation Horizon Europe (2021–2027) under the Marie Skłodowska-Curie Grant Agreement No. 101065466. H.P.R. was funded by the Manfred Eigen-Förderstiftung (Principles of Cancer Research - Stipend for exceptional, independently working young scientists) and by King's College London as part of the "Neuro-Immune Interactions in Health & Disease Wellcome Trust PhD Programme. B.F.T. is supported by the National University of Singapore Research Scholarship. J.A.C. is supported by the International Max-Planck Research School (IMPRS) for Genome Science. This work was supported by the Francis Crick Institute which receives its core funding from Cancer Research UK (CC0102), the UK Medical Research Council (CC0102), and the Wellcome Trust (CC0102). For the purpose of Open Access, the author has applied a CC BY public copyright licence to any Author Accepted Manuscript version arising from this submission.

## Author contributions

WTS, HPR, JAC, MM and JL developed the project and wrote the manuscript. WTS, HPR and JAC performed the data analysis, which was supervised by MM and JL. WTS, HPR, NCC and BFT performed and measured the biochemical experiments, which was supervised by HL, MM and JL. MR and RP performed proteomics assays. HU supervised the proteomics and the MS measurements. SB, FH and AC purified and provided substrates and proteasomes.

## Competing interests

The authors declare no competing interests.

## Additional information

[1]Research Group of Quantitative and Systems Biology, Max-Planck-Institute for Multidisciplinary Sciences, 37077 Göttingen, Germany. [2]Centre for Inflammation Biology and Cancer Immunology & Peter Gorer Department of Immunobiology, King's College London, SE1 1UL London, UK. [3]Research Group of Molecular Immunology, Francis Crick Institute, NW1 1AT London, UK. [4]Immunology Programme, Life Sciences Institute; Immunology Translational Research Program and Department of Microbiology and Immunology, Yong Loo Lin School of Medicine, National University of Singapore, Singapore 117456, Singapore. [5]Research Group of Bioanalytical Mass Spectrometry, Max-Planck-Institute for Multidisciplinary Sciences, 37077 Göttingen, Germany. [6]Department of Structural Dynamics, Max-Planck-Institute for Multidisciplinary Sciences, 37077 Göttingen, Germany. [7]Department of NMR-based Structural Biology, Max-Planck-Institute for Multidisciplinary Sciences, 37077 Göttingen, Germany. [8]Research Group of Structural Biochemistry and Mechanisms, Max-Planck-Institute for Multidisciplinary Sciences, 37077 Göttingen, Germany. [9]Institute of Clinical Chemistry, University Medical Center Göttingen, 37075 Göttingen, Germany. [10]These authors contributed equally: Wai Tuck Soh, Hanna P. Roetschke, John A. Cormican. [11]These authors jointly supervised this work: Juliane Liepe, Michele Mishto. ✉e-mail: jliepe@mpinat.mpg.de; michele.mishto@kcl.ac.uk

