## [Peer Review File · Nature Communications]

REVIEWER COMMENTS

Reviewer #1 (Remarks to the Author):

Trans-splicing: the term is used inconsistently in the literature, as pointed out by the authors. Trans-splicing between different proteins is a much more obscure effect and should not be confused with the trans-splicing discussed in this paper. Maybe a different name would be better?

Introduction:

There are in my view too many self references in the introduction and some refs are not included. It is still debated what the importance of spliced peptides is. To my knowledge there are few examples of spliced peptides where a specific role in immunity could be established experimentally. I think the focus should be more on these hard facts and less on the high throughput approaches.

Since T cells are promiscuous and because of effects such as epitope spreading it is difficult to prove that the T cell expansion was caused by a specific peptide and not another. Therefore what does it really mean if a T cell recognises a spliced peptide?

Also, the dark matter of immunopeptidomics is not only formed by spliced peptides: there are many noncanonical peptides such as lncRNA, alternative splicing, PTMs, circular RNA, SNP, Indel, frameshifts, gene fusion, viral and bacterial peptides and many more. It is hard to unambiguously pinpoint the origin of a peptide and nearly impossible to do this by MS only. It would be nice to have a frank discussion about this in the introduction and discussion. What is required to definitively prove that a peptide is indeed a PCPS?

I suspect that the numbers of spliced peptides reported by the large scale MS studies by Liebe et al. and Faridi et al. were far too high. I'm not saying that spliced peptides don't exist or they don't have a role, but one has to be very careful when MS searching the very large search space. Here the authors should give credit to the papers by Mylonas et al, Rolfs et al, Lichti, which were published in less prominent journals but helped to advance the field by performing more careful analysis of MS/MS data for spliced peptides. Many of the ideas and concepts first published in these papers are now part of the inSPIRE software. Even the Prosit immunopeptidomics paper (Wilhelm et al) had some examples, where their predictor did not agree with the peptides from Liebe et al. I'm happy to see that the new software includes both Prosit and Percolator. It would be nice to see more discussion about these MS/MS search issues in the introduction.

Why did you 'only' use 10 proteins for your investigation? It would be interesting to have more variety in the proteins.

Could you relate the concentration of your 4 digested proteins to their concentration in real cells? If I understood correctly, your proteins are not ubiquitinated and I wonder how they are recognized by the proteasome and how realistic this situation is compared to real cells?

77% of the peptide volume is from non-spliced peptides in your in vitro assay. How does this value change if you use less protein material, do all spliced peptides disappear first or do they stay at 23%? Maybe you could use the data for the proteins where you detected less peptides to investigate this. Do you think these percentages of spliced/non-spliced peptides are similar in real cells, where many different proteins compete for proteasomes?

Can the hotspots you observe be explained by proteasomal cleavage rules?

Would it be possible to check whether IDPs are more represented in the immunopeptidome (e.g in HLA atlas). This could give support to your hypothesis that they are preferentially targeted by proteasomes.

Did you use an FDR/qvalue threshold for Precolator? If yes which one?

Reviewer #2 (Remarks to the Author):

This interesting manuscript by Soh and colleagues explores 20S proteasomal digestion of four human intrinsically disordered protein (IDP) substrates (tau, alpha-synuclein, IL-37beta and annexin A1) – of note several other proteins were not able to be digested by the 20S proteasome a point not really followed up on. For instance, does reductive alkylation of say ovalbumin make which results in loss of tertiary structure make this a better substrate for the 20S proteasomes?

Overall this is a well written and generally well thought out manuscript. The complex and in depth informatics used to assess PCPS is a hallmark of this team and based on previously published algorithms (mostly developed for synthetic peptide digests) with several important nuances applied to the IDP digests. All together a large dataset and thoughtful and provocative analysis. I have number of specific queries for the authors;

(i) Prosit spectral prediction is used as a powerful tool to confirm peptide identification for both linear and spliced peptides. Alternative approaches for spliced peptide identification have used de novo sequencing – how does this differ wrt to outcomes given similar principles guide de novo assignment of MS/MS spectra and ms/ms spectral predictions by tools like Prosit? Have the authors compared a de novo approach to their Inspire-aspire workflow?

(ii) The authors state (p2 lns 67-69) “The 20S proteasomes are also active as such, both in the intracellular and extracellular space (16-18). They preferentially process intrinsically disordered proteins (IDPs), which contain large unstructured segments or even completely lack a defined tertiary structure in their native state, and have been estimated to represent up to 30% of the intracellular proteome” – Is it really the case that 20S proteasomes exist in isolation and as such they prefer IDP substrates? Perhaps this simply represents an access issue to the 20S proteasome core? Presumably other molecules are involved in the processing of more ordered substrates and the more mature forms of the proteasome?

(iii) Areas of the Introduction could use some effort to improve flow rather than reading as a compendium of previous studies to improve the readability.

(iv) The low proportion of trans-spliced peptides may also reflect the special case nature of the detected trans-spliced peptides (i.e. where an overlap in splice reactants exist). In a more complex system one could imagine trans-splicing being more readily detected and therefore more abundant?

(v) Lns 333-334 “Together, it suggests that the shorter the length of the fragment that is bound to the proteasome’s active site, the higher is the chance of PCPS’ – since the authors discuss the origin of much controversy – the immunopeptidome” it would be useful to consider that full length proteins may not always be the source of proteasomal substrates – defective ribosome initiation products and other non-canonical sources may predispose to short substrates and impact on the efficiency of PCPS and the specificity of the splice junctions? Some comments would be useful.

(vi) Ln 337 "Firstly, the N-terminal splice-reactants were, on average, shorter than their C-terminal counterpart (Fig. 5e)". Does this bear true in immunopeptidome derived sequences – even the ones that have been confirmed functionally by van den Eynde, Yewdell, Faridi and others?

(vii) Ln 389-90; "where amino acids preferentially used for peptide splicing (such as G and C), were disfavored for peptide hydrolysis and vice versa (e.g., D, L, I; Fig. 6c)". These are important insights but do can these preferences be used to map HLA-peptide cis-spliced sequences produced to date?

(viii) My last few points have focused on HLA-bound peptides, my reason for this emphasis is that this is probably one of the main physiological impacts of such processing. Are there other roles perhaps for the longer degradation products that may correlate with the measured degradation products? Would such species be detectable by proteomics' for instance? [Perhaps more for speculation than measurement]

Reviewer #3 (Remarks to the Author):

The manuscript by Soh et al. investigates the peptide splicing capabilities of the proteasome. Specifically, the study focuses on proteomic analysis of the peptides generated by the degradation of 4 intrinsically unstructured proteins. They propose that 21% and 2% of the produced peptides are cis-spliced and trans-spliced peptides. In general, I find much of the data presented to be quite preliminary, often not convincing and open to alternative interpretations.

I give below some suggestions for how the data might be made more convincing:

In general, as the field is so controversial, extra rigorous analysis is required. Making a decisive conclusion from the analysis of only 4 proteins is far fetch. Only 4 proteins were investigated and not 10 as the authors claim in the introduction.

The biochemical results shown in Figures S1 and S2 are extremely sloppy, making the entire work unreliable:

Fig. S1a - why does a band corresponding to proteasome subunit alpha-4 appear even if the proteasome is absent?

In Fig. S2a - why are enolase levels reduced after 24 hours if the proteasome is absent?

In Fig. S2a - bands of the proteasome are detected even in the absence of the proteasome.

In Fig. S2b - bands corresponding to the 20S proteasome are not detected even though the proteasome is in the sample.

In Fig. S12 additional bands appear in the sample of the purified proteasome? To which proteins do they correspond? How can we exclude the possibility that they are the ones that cause splicing and not the proteasome?

In Figure 3e the authors analyze a splice peptide [G][QLGKNEEGAPQE] which actually fits in mass to the non-spliced a-synuclein peptide [QLGKNEEGAPQEG]. The authors should have examined the non-spliced options and compare their scores, especially given the extremely low intensity of the b ions. In addition, heavy peptides spiked into the sample, prior to LC-MS/MS should have been used for validation, in all cases.

The authors do not explain what is the mechanism for peptide conjugation. How conjugation is kinetically catalyzed in the presence of water.

Careful analysis of the LC-MS/MS spectra in the supplementary files indicates a low correlation between peptides and counterpart synthetic peptides. The majority of the peptide fragments

(generated by proteolysis of intact proteins) are not assigned, while synthetic peptides peaks that are assigned are not detected in the peptide fragment spectra. This reduces the reliability of data analysis. The theoretical database that this study relies on, is as expected very large, leading to a high false discovery rate. Given this drawback, which cannot be overlooked, I would have expected that the authors display the data using the more stringent engine (Figure S4, invitroSPI) to present more reliable results. Moreover, as one of the aims of this study is the development of the inspire software and its advantages. To convince the reader that the performance of the software is superior, more comparisons to invitroSPI or other software should be made.

It was very hard to follow the flow of the manuscript. The authors use an array of unique terms, that are introduced only shortly to the readers, legends are very simplistic and do not assist the reader in comprehending the fine details of the results. Moreover, the more significant part of this study focuses on comparisons to other studies that the reader has little access to (or requires the reader to read two papers in parallel).

We would like to thank the Reviewers and the Editor for their constructive criticisms. We addressed all their comments in the revised version of the manuscript.

Summarizing the main changes in the revised manuscript, we:

1. tripled the number of protein substrates investigated in the study and digested by proteasomes, thereby providing a larger dataset of degraded proteins and a database of peptide products as foundation of our analysis;
2. introduced a significant change in the method used for the identification of non-spliced and spliced peptide products in the digestions, to address several criticisms and comments of the reviewers. Indeed, the inSPIRE version used in the original manuscript (based upon inSPIRE 1.0; published in Mol. Cell. Prot. 2022) has been substituted with inSPIRE 1.5., wherein:
 - we added support for rescoring of MSFragger search results. The use of MSFragger in place of Mascot means that all tools in the computational pipeline are open-source/open-access, increasing reproducibility and accessibility. Furthermore, the use of MSFragger over Mascot showed improved performance in our benchmarking analysis;
 - we combined all results from all protein digestion for the generation of the final Percolator input in inSPIRE 1.5. This leads to more robust and standardized FDR estimation than calculations based on a single protein digestion;
 - we improved the retention time error by calculating iRT difference rather than the actual difference. While previously this error was calculated based on fitting the iRT standard to the experimental results, we now fit the experimental results to the iRT standard. This creates a fairer metric for judging PSM quality across samples;
 - final assignments are now provided with peptide level probabilities. This is a more stringent cut-off than the PSM level. During the manuscript revision, we understood that this strategy is more appropriate for our experimental setting where the large number of replicates and the high possibility of canonical peptides being detected across multiple scans lead to a non-standard distribution of Percolator scores;
 - we used the Spearman correlation as a feature in *in vitro* protein digestion mode rather than matched coverage. This feature has been given far more weight in the Percolator feature importance table, now reported in **Fig. S3**.

All these changes improved the inSPIRE performance (see **Fig. 1 for Reviewers**), but also led to differences from the PSMs reported in the original manuscript. We believe that these stricter cut-offs provide a suitable response to any concerns raised over spliced peptide identification. We also carried out a comparison of PSM features and method performance by comparing inSPIRE 1.5 to invitroSPI and two *de novo* approaches, as requested by Reviewers (see **Fig. S5**).

3. We developed, released and applied iBench 2.0, which is the latest version of iBench (Cormican et al., 2022), for a systematic benchmarking.

4. We introduced various new analyses to strengthen the confidence of the readers and reviewers in the identification of spliced peptides (e.g., see **Fig. S3-S5, S8-S9**)

In addition to this summary, we would like to reply point by point (in blue) to the comments (in *italics*) as follows:

Reviewer #1 (Remarks to the Author):

Trans-splicing: the term is used inconsistently in the literature, as pointed out by the authors. Trans-splicing between different proteins is a much more obscure effect and should not be confused with the trans-splicing discussed in this paper. Maybe a different name would be better?

Our reply

We thank the Reviewer for the suggestion. We modified the text defining heterologous trans-splicing (i.e. occurring between two different proteins as described by Purcell and Faridi in immunopeptidomes,) and homologous trans-splicing (i.e. occurring between two molecules of the same protein). In this study, only homologous trans-spliced peptides were investigated. E.g., see revised **Fig. 1**.

Introduction:

There are in my view too many self references in the introduction and some refs are not included. It is still debated what the importance of spliced peptides is. To my knowledge there are few examples of spliced peptides where a specific role in immunity could be established experimentally. I think the focus should be more on these hard facts and less on the high throughput approaches.

Our reply

We thank the Reviewer for the criticism. We updated the references of that section and put emphasis on the immunological evidence of spliced epitopes. We also added a brief section about the immunological evidence for hybrid insulin peptides, which are spliced epitopes, too. See lines **98-124**.

Since T cells are promiscuous and because of effects such as epitope spreading it is difficult to proof that the T cell expansion was caused by a specific peptide and not another. Therefore what does it really mean if a T cell recognizes a spliced peptide?

Our reply

We acknowledge the point raised by the Reviewer. We added a sentence in the manuscript to comment on that. In general, it depends on the aim of the study. If a TCR recognizes a spliced peptide, it means that the spliced peptide is presented by MHC-I. We and our collaborators previously performed specific experiments to exclude that cross-recognition could be the cause of the CD8 T cell response (e.g., Platteel et al., Cell Rep. 2017). Furthermore, the Van den Eynde lab performed many experiments along the same lines to show T cell responses specific against spliced peptides. We reported these references in the revised Introduction. See line **99-104** and **112-114**.

Also, the dark matter of immunopeptidomics is not only formed by spliced peptides: there are many noncanonical peptides such as lncRNA, alternative splicing, PTMs, circular RNA, SNP, Indel, frameshifts, gene fusion, viral and bacterial peptides and many more. It is hard to unambiguously pinpoint the origin of a peptide and nearly impossible to do this by MS only. It would be nice to have a frank discussion about this in the introduction and discussion. What is required to definitively proof that a peptide is indeed a PCPS?

Our reply

We agree with the Reviewer that this is a very important point when analyzing the peptide repertoire of HLA-I immunopeptidomes, which was not systematically addressed as far as we know. However, we do not think it is pertinent to the present study since, here, we analyzed the digestion of single purified proteins, thus the antigen source of the identified peptides is known. They are whole proteins (or larger domains of a protein) and do not derive from any cryptic origins. We have mentioned this in the revised Introduction, which represents a further advantage of the type of experimental set up we have employed in the study to decipher the features of peptide hydrolysis and peptide splicing by 20S proteasomes as such. We thank the Reviewer for the good suggestion. See lines **144**.

I suspect that the numbers of spliced peptides reported by the large scale MS studies by Liebe et al. and Faridi et al. were far too high. I'm not saying that spliced peptides don't exist or they don't have a role, but one has to be very careful when MS searching the very large search space. Here the authors should give credit to the papers by Mylonas et al, Rolfs et al, Lichti, which were published in less prominent journals but helped to advance the field by performing more careful analysis of MS/MS data for spliced peptides. Many of the ideas and concepts first published in these

papers are now part of the inSPIRE software. Even the Prosit immunopeptidomics paper (Wilhelm et al) had some examples, where their predictor did not agree with the peptides from Liepe et al. I'm happy to see that the new software includes both Prosit and Percolator. It would be nice to see more discussion about these MS/MS search issues in the introduction.

Our reply

We agree with the Reviewer that the spliced peptides reported in Liepe et al., Science 2016 and 2019 and Faridi et al., 2018 have been intensively re-investigated by other groups, who arrived often to different conclusions in terms of spliced peptide frequency. We reported some of those studies in the revised Introduction, although they were already summarized in the cited reviews in the original manuscript. See lines **127-135**.

Some of the inSPIRE 1.5 steps have previously been used by others to test the accuracy of the spliced peptide identification. In fact, in the last years, we learned from the critical evaluation of our work by esteemed colleagues and used their critical angles to improve our (initially faulty) strategy for spliced peptide identification. We mention this in the description of inSPIRE 1.5 (see lines **1319-1321**) as well as in the Discussion (see lines **552**).

Regarding the database size and the statistical issues connected to that, we fully agree with the Reviewer that this is a very important point, which has not been systematically addressed so far, at least to our knowledge. MS rescoring approaches such as Percolator have been used to tackle the so-called large search space problem in MS, which provided the basis of inSPIRE 1.5. We added a brief section the introduction about this topic, although we think that a specific study on this matter is deeply needed.

In the original manuscript, we addressed this issue for the specific experimental conditions of this study (i.e., the digestion of either a single protein or a single polypeptide) in **Fig. S3** and reported the generation efficiency for each peptide type in in **Fig. 4g**. In the revised manuscript, we reported the theoretical database size for non-spliced and spliced peptides side by side to their generation efficacy to improve the accessibility of the results to the reader (now **Fig. 4a**).

In this revision we also expanded on the methods used to combat the enlarged search space of spliced peptide candidates. In particular, we highlight the penalization of spliced peptide assignments compared to non-spliced within Percolator as shown in **Fig. S3**. The effects of these can be seen in the distribution of the Prosit spectral angles and Prosit predicted retention time errors for identified spliced and non-spliced peptides (new **Fig. S8**).

Why did you 'only' use 10 proteins for your investigation? It would be interesting to have more variety in the proteins.

Our reply

Bearing in mind that only a handful of proteins have been investigated through digestion by 20S proteasomes in the last 30 years, we were excited to have the results from 4 proteins digested out of 10 tested. However, to address the reviewer's comment we extended the revised manuscript to 27 proteins, of which 15 showed to be efficiently processed by 20S proteasomes under our experimental conditions. This leads to a robust increase of the substrate diversity (we now have both human and bacterial sequences, among others) and of the number of identified peptide products (16,219 non-spliced and 2,428 spliced rather than 3,692 non spliced and 1,737 spliced peptides in the original manuscript).

Could you relate the concentration of your 4 digested proteins to their concentration in real cells? If I understood correctly, your proteins are not ubiquitinated and I wonder how they are recognized by the proteasome and how realistic this situation is compared to real cells?

Our reply

In the revised version of the manuscript, we were able to obtain 15 protein substrates that can be digested by 20S proteasomes. For a systematic study of substrate digestion by 20S proteasomes, we employed a proteasome : protein ratio (ranging from 1:5 to 1:100) that varied from protein to protein depending on the degradation rate. We aimed to have a similar relative degradation rate of the substrates. This strategy has been used by us and others in the degradation of polypeptides, for the same reason. In a preliminary evaluation of the ratio of proteasome to specific targets (used in this study), we obtained on average a range from 5 : 1 to 1:1, because proteasomes are abundant in cells. However, we think that this would be misleading, because there are also many different protein substrates for proteasomes (and IDPs) in cells. Therefore, in our experimental set up we privileged the *in vitro* degradation rate as reference.

As better explained in the revised manuscript, the ubiquitinated proteins are usually recognized by the 19S regulatory particles (specifically, Rpn10, Rpn1 and Rpn13 subunits) of the 26S proteasome. In this study we used human 20S proteasomes, which is the catalytic core particle responsible for the proteolytic activity, without the 19S complex. The mechanism by which the 20S proteasomes recognizes non-Ub proteins is still an unresolved issue, which we think

would need a dedicated study to be addressed. However, 20S proteasomes have been shown to digest disordered proteins (doi.org/10.3390/biom4030862). In the revised version of this manuscript, we have had screened 27 protein substrates and observed a similar conclusion that protein substrates that can be digested by 20S proteasomes have a higher intrinsically disordered score. In addition, it was found that a substantial proportion of proteasomes in mammalian cells are 20S proteasomes and were able to process more than 20% of the cellular proteins in cell extracts (DOI: 10.3390/biom4030862). A recent study has provided evidence that a disordered protein was processed by 20S proteasomes in cells. (DOI: 10.1038/s41467-021-26427-0). Remarkably, the same study also showed that 20S proteasomes are capable of degrading non-Ub as well as Ub proteins. In addition, catalytic core regulators (CCRs) have recently been hypothesized to impact on 20S proteasome gate without affecting the catalytic subunits (doi: 10.1038/s41467-023-38404-w.).

In addition, the proteasome is a tightly regulated barrel with the catalytic sites inside the barrel, which should preserve their catalytic activity. Recently, we also speculated that the barrel shape of proteasome and the location of the catalytic sites is promoting peptide splicing by separating inside from outside of the proteasome barrel, which is a feature that is conserved in our *in vitro* experiments (Liepe et al., eLife 2015).

In the revised manuscript we have also performed novel experiments using different concentrations of Ffh as shown in **Fig. S10**, which we believe may partially address the Reviewer's comment.

Based on these elements, we reckon that our *in vitro* system should mimic what happens in cells, since the catalytic core of proteasomes in cells should be well represented in the *in vitro* system.

We have added further information about IDP and proteasome literature regarding those aspects in **line 67-86**.

77% of the peptide volume is from non-spliced peptides in your in vitro assay. How does this value change if you use less protein material, do all spliced peptides disappear first or do they stay at 23%?

Maybe you could use the data for the proteins where you detected less peptides to investigate this. Do you think these percentages of spliced/non-spliced peptides are similar in real cells, where many different proteins compete for proteasomes?

Our reply

Another interesting point raised by the Reviewer. The proteasome : protein ratio used in the generation of the final set of experiment varied from protein to protein depending on the degradation rate. We wanted to have similar (as much as possible) degradation rate among proteins. However, to test the Reviewer's hypothesis, we investigated if the proteasome to target molar ratio used in our *in vitro* digestions could impinge upon the frequency of spliced peptides identified. We selected the Ffh substrate, which gave a relative frequency of spliced peptide of 18.3% (**Table 1**), and performed *in vitro* digestions in which the 20S proteasome concentration was kept constant and the proteasome to target molar ratio was reduced from 1:25 to 1:12.5 and 1:6.2. Reaction volumes containing similar amount of substrate were measured by MS - to avoid bias in the peptide product identification due to different peptide/protein amounts loaded into the LC-MS/MS – and the analysis was carried out by inSPIRE 1.5 – aSPIRE. The Ffh degradation, visualized and quantified on SDS gel, showed a similar degradation rate among the different conditions tested (new **Fig. S10a-c**). This suggested a Vmax state in our experimental conditions that varied over time, in agreement with a study carried out with short fluorogenic peptide substrates (doi: 10.7554/eLife.07545). The relative frequency of unique non-spliced and spliced peptides did not vary considerably between conditions (new **Fig. S10d**). There was a relatively similar coverage of the substrate sequence by the amount of non-spliced and spliced peptide products among the different conditions (new **Fig. S10e**). All these analyses suggested that the molar ratio of proteasome to target should not have impinged upon the catalytic activities of 20S proteasomes in our experimental setting.

Can the hotspots you observe be explained by proteasomal cleavage rules?

Our reply

We thank the reviewer for raising this question, which has triggered an interesting new line of analysis. The presence of hotspots for non-spliced and spliced peptides is one of the most interesting aspects of the study, in our opinion, and it emerged only by the first analysis of whole protein digestions carried out in this study. Also, the fact that these hotspots did not correlate with intrinsically disordered regions or secondary structure elements (new **Fig. 6a-b, Fig. S12**) raised our curiosity.

To address this question, we investigated whether the local amino acid preferences depicted in the new **Fig. 6d** could be linked back to “hotspots” observed on global substrate sequence level (new **Fig. 6e-j**). A detail summary of the outcome of this analysis is reported in the revised Results section. In summary, an analysis of the amino acid preferences and usage inside and outside the hotspot regions did not provide a resolute hypothesis of why hotspot

regions exist, although it pointed toward different peptide hydrolysis and splicing dynamics inside and outside these regions where the sequence motifs surrounding the P₁ position seem to have a different weight in both catalytic reactions. We think that a follow up study focusing on this initial observation, perhaps by digesting and comparing proteins that differ only partially in sequence (both varying hotspots and non-hotspots regions), might be a pathway to fully address the interesting question raised by the reviewer.

Would it be possible to check whether IDPs are more represented in the immunopeptidome (e.g in HLA atlas). This could give support to your hypothesis that they are preferentially targeted by proteasomes.

Our reply

The hypothesis that IDPs can be processed by 20S proteasomes is not ours (e.g., see Myers et al., 2018) as mentioned in the Introduction. We tested this hypothesis in our experimental conditions, and it was nicely confirmed, also in the new set of substrates added to the revised manuscript. We extended the information provide about IDP degradation in the Introduction and Result sections of the revised manuscript.

In addition, we followed the Reviewer's question and added an additional layer of information to our study and to the implication that it can have also for immunologists studying HLA-I immunopeptidomes, in the revised manuscript. To this end, we investigated the Reviewer's hypothesis in the IEDB antigenic peptide database considering HLA-A and HLA-B separately (the HLA Atlas database was not informative enough for this analysis). Antigens represented in both pools of HLA-I complexes had a significantly greater predicted number of long-disorder segments than the group of proteins that were not represented in HLA-I immunopeptidomes (new **Fig. 2c**). The difference in median SLIDER score between groups was comparable to what was observed in the pool of *in vitro* digested and non-digested proteins (new **Fig. 2b**). This could suggest an immunologically relevant role of 20S proteasome-mediated processing of proteins with disordered regions in HLA-I antigen presentation.

However, in the revised manuscript we extended the pool of proteins tested and those degraded by proteasomes. All 15 degraded proteins had some regions predicted to be disorder in absence of partner interactions but only few were defined as classical IDPs. We clarified this aspect in the Results section, and we did not refer to these 15 degraded proteins as IDPs. In agreement with this, the title of the manuscript was changed accordingly.

Did you use an FDR/qvalue threshold for Precolator? If yes which one?

Our reply

We thank the reviewer for raising the importance of the FDR/q-value threshold used for PSM assignment. Indeed, in the original submission we did use a PSM level FDR/q-value cut off of 1% for peptide identifications. In this revision, we used the same 1% FDR/q-value cut off but applied on the peptide identification level, in addition to the improved precision of the inSPIRE 1.5 method. This is an even more stringent requirement, especially for the spliced peptides identified. This can be seen in **Fig. 1 for Reviewers**, particularly in the increased matched coverage (**Fig. 1b for Reviewers**), reduced iRT error and posterior error rate probabilities (**Fig. 1c,d for Reviewers**). We also provide further analysis and discussion on FDR and posterior error rates per stratum (e.g., **File S2**).

We clarified this aspect and expanded the discussion in the revised Methods.

Reviewer #2 (Remarks to the Author):

This interesting manuscript by Soh and colleagues explores 20S proteasomal digestion of four human intrinsically disordered protein (IDP) substrates (tau, alpha-synuclein, IL-37beta and annexin A1) – of note several other proteins were not able to be digested by the 20S proteasome a point not really followed up on.

For instance, does reductive alkylation of say ovalbumin make which results in loss of tertiary structure make this a better substrate for the 20S proteasomes?

Our reply

We agree with the Reviewer that understanding the mechanisms of protein selection as substrates of 20S proteasomes is a very interesting subject matter. The objective of this study is, however, to investigate peptide hydrolysis and splicing from whole proteins, which has not been attempted before.

The presence of disordered regions is clearly a driving force, as also confirmed by the new set of 17 substrates added to the revised manuscript.

The reviewer raised a very interesting aspect about the substrate accessibility to 20S proteasomes hindered by the substrate's tertiary structure. Instead of reductive alkylation, we employed detergent (SDS) to assist the unfolding of the substrate for 20S proteasomes digestion. We demonstrated that Ova can be completely digested by 20S in the presence of SDS as shown in the **Fig. S2** of the original manuscript (now reported in **Fig.2a,b for Reviewers**, where panel **Fig. 2c for Reviewers** is a magnification of panel **a**). However, some controls were not shown in that figure, which could have led to some misunderstanding. To clarify this aspect, in the **Fig.2d for Reviewers**, we presented the effect of SDS on 20S proteasomes alone and Ova alone. Despite that SDS helped in the denaturation of Ova for 20S proteasomes degradation, proteasomes itself also became susceptible to self-proteolysis.

We considered the interest of both Reviewer 2 and 3 on this matter, and have performed and shown new preliminary experiments aimed at clarifying the effect of SDS on protein and proteasome stability in the presence of SDS (see **Fig.2 for Reviewers**). However, we think that this interesting question should be addressed in a study focusing on this aspect of proteasome proteolysis, and thus we have removed any reference to SDS-based protocols in the revised manuscript.

Overall this is a well written and generally well thought out manuscript. The complex and in depth informatics used to assess PCPS is a hallmark of this team and based on previously published algorithms (mostly developed for synthetic peptide digests) with several important nuances applied to the IDP digests. All together a large dataset and thoughtful and provocative analysis. I have number of specific queries for the authors;

(i) Prosit spectral prediction is used as a powerful tool to confirm peptide identification for both linear and spliced peptides. Alternative approaches for spliced peptide identification have used de novo sequencing – how does this differ wrt to outcomes given similar principles guide de novo assignment of MS/MS spectra and ms/ms spectral predictions by tools like Prosit? Have the authors compared a de novo approach to their Inspire-aspire workflow?

Our reply

We thank the Reviewer for this comment. Following the reviewer's line of reasoning, we studied the performance of the inSPIRE pipeline in comparison to two *de novo* methodologies (details of these approaches are provided in the revised Materials & Methods). The two *de novo* methods were motivated by the methods initially proposed by Faridi et al. (Sci Immunol. 2018) and Mylonas et al. (MCP 2018) with adaptations to *in vitro* protein digestions. Furthermore, we developed upon iBench – a software for benchmarking of mass spectrometry identification algorithms (Cormican et al., Proteomics 2022) – developed and released iBench 2.0, which allows benchmarking in this *in vitro* setting.

These results are presented in **Fig. S5** in the revised manuscript. This benchmarking showed clearly improved performance for the inSPIRE 1.5 -aSPIRE workflow. Firstly, in terms of the shape of the precision-recall curve, which indicates that inSPIRE 1.5 better distinguishes between correct and incorrect PSMs for cis-spliced peptides (**Fig. S5c**). Furthermore, the benchmarking indicates that inSPIRE 1.5 is more reliable in terms of FDR estimation – the precision at the applied cut-off as represented by the large dot on the PR curve. This indicates that inSPIRE 1.5 -aSPIRE identifications at 1% FDR were more accurate than those of the *de novo* methods.

(ii) The authors state (p2 Ins 67-69) “The 20S proteasomes are also active as such, both in the intracellular and extracellular space (16-18). They preferentially process intrinsically disordered proteins (IDPs), which contain large unstructured segments or even completely lack a defined tertiary structure in their native state, and have been estimated to represent up to 30% of the intracellular proteome” – Is it really the case that 20S proteasomes exist in isolation and as such they prefer IDP substrates?

Our reply

The hypothesis that IDPs can be processed by 20S proteasomes is not ours (e.g., see Myers et al., 2018) as mentioned in the Introduction. We tested this hypothesis in our experimental conditions, and it was nicely confirmed, also in the new set of substrates added to the revised manuscript. We expanded that part of the Introduction by reporting more evidence reported in literature, for example by citing a recent study showing that both 20S and 26S proteasomes are capable of degrading non-Ub and Ub proteins (doi: 10.1038/s41467-021-26427-0; see **line 67-86**) as well as a recent paper suggesting the involvement of CCRs in 20S proteasome activation (without affecting the catalytic cores; see **line 80**).

Regarding the question about the relative ratio 26S:20S proteasomes in cells (assuming that this is what the reviewer refers to with '*20S proteasomes exist in isolation*'), there has been an estimation that the 20S proteasomes represent 45-75% of all assembled proteasomes in cells (doi: 10.3390/biom11020148). In addition, several studies have shown that 20S proteasomes can be compartmentalized in vesicles, be secreted extracellularly, and were found in extracellular fluid such as serum and plasma. These studies are reported in the revised Introduction.

Perhaps this simply represents an access issue to the 20S proteasome core? Presumably other molecules are involved in the processing of more ordered substrates and the more mature forms of the proteasome?

Our reply

We would speculate along the line of thoughts of the Reviewer, although understanding the mechanisms of selection of proteins as substrates of 20S proteasomes is beyond the scope of this study. We added other details from previous studies in the revised Introduction. See **line 67-86**.

(iii) Areas of the Introduction could use some effort to improve flow rather than reading as a compendium of previous studies to improve the readability.

Our reply

We thank the Reviewer for the suggestion. We worked on the narrative of the Introduction in the revised manuscript.

(iv) The low proportion of trans-spliced peptides may also reflect the special case nature of the detected trans-spliced peptides (i.e. where an overlap in splice reactants exist). In a more complex system one could imagine trans-splicing being more readily detected and therefore more abundant?

Our reply

We thank the Reviewer for the suggestion. In the revised manuscript, we modified the text defining heterologous trans-splicing (e.g., that described by Purcell and Faridi in immunopeptidomes, occurring between two different proteins) and homologous trans-splicing (i.e., occurring between two molecules of the same protein), which is what we analyzed in this study. In our experiments, the generation efficacy between *cis*-spliced and homologous *trans*-spliced peptides did indeed not differ. Because the study does not address heterologous trans-spliced peptides, we preferred to avoid speculations in that direction in the revised manuscript.

(v) Lns 333-334 "Together, it suggests that the shorter the length of the fragment that is bound to the proteasome's active site, the higher is the chance of PCPS' – since the authors discuss the origin of much controversy – the immunopeptidome" it would be useful to consider that full length proteins may not always be the source of proteasomal substrates – defective ribosome initiation products and other non-canonical sources may predispose to short substrates and impact on the efficiency of PCPS and the specificity of the splice junctions? Some comments would be useful.

Our reply

This is a very interesting hypothesis. However, by expanding the pool of proteins investigated in the study, the original observation was changed showing a more complex behavior. Therefore, to reduce the complexity and to better understand this study, we preferred to remove the corresponding figure and we planned to investigate in a specific study both the phenomenon and the possible implications with DRiPs in a specific study.

(vi) Ln 337 “Firstly, the N-terminal splice-reactants were, on average, shorter than their C-terminal counterpart (Fig. 5e)”. Does this bear true in immunopeptidome derived sequences – even the ones that have been confirmed functionally by van den Eynde, Yewdell, Faridi and others?

Our reply

This is an interesting suggestion. However, the community is currently skeptical about any identification of spliced peptides in immunopeptidomes, therefore we do not know what spliced immunopeptidome dataset could be used to check this hypothesis. We strongly believe that there is a need to have a novel method for identification of spliced peptides in immunopeptidomes, and for the benchmarking of its performance against other methods. Until then, it is not possible to test this hypothesis.

While some functionally validated spliced epitopes indeed have a shorter N-terminal splice-reactant as compared to their C-terminal counterpart (e.g. [RTK][QLYPEW] (Vigneron et al. Science 2004), [S][AYGRQVYL] (Platteel et al. EJI 2016), [ISY][AFYKL] (Platteel et al. Cell Rep 2017), [FSD][QLIHLY] (Paes et al. PNAS 2019)), this is also not the case for other epitopes (e.g. [NTYAS][PRFK] (Hanada et al. Nature 2004), [SLPRGT][STPK] (Warren et al. Science 2006), [IYMDGT][ADFSF] (Dalet et al. PNAS 2011), [RSYVPLAH][R] (Michaux et al, JI 2014), [QLYPEW][RTK] (Ebstein et al. Sci Rep 2015)). The number of spliced epitopes investigated using CTL clones by Van den Eynde, Sijts, Purcell, us and Ovaas is too small to draw any meaningful conclusion. Furthermore, post-proteasomal N-terminal trimming of the final epitopes by cytosolic or endoplasmic aminopeptidases needs to be considered, hindering the analysis of splice reactant length in immunopeptidomes. We hope we will soon be able to approach this hypothesis.

(vii) Ln 389-90; “where amino acids preferentially used for peptide splicing (such as G and C), were disfavored for peptide hydrolysis and vice versa (e.g., D, L, I; Fig. 6c)”. These are important insights but do can these preferences be used to map HLA-peptide cis-spliced sequences produced to date?

Our reply

This is a further very interesting suggestion by the reviewer. However, as mentioned above, the community is currently skeptical about any identification of spliced peptides in immunopeptidomes, therefore we do not know what spliced immunopeptidome dataset could be used to check this hypothesis. When reviewing the spliced neopeptides that have been described and functionally validated, some do indeed reflect the amino acid preferences detected *in vitro* for P₁ (Fig. 6d), for instance:

- [NTYAS][PRFK] (Hanada et al. Nature 2004)
- [SLPRGT][STPK] (Warren et al. Science 2006)
- [IYMDGT][ADFSF] (Dalet et al. PNAS 2011)
- [S][AYGRQVYL] (Platteel et al. EJI 2016)
- [LLL**G**][LLTKV] (Faridi et al. CIR 2020)
- [LLLEA][LEQL] (Faridi et al. CIR 2020)

However, there are also examples of spliced neopeptides described in literature that do not follow this pattern of P₁, for example:

- [ISY][AFYKL] (Platteel et al. Cell Rep 2017)
- [FSD][QLIHLY] (Paes et al. PNAS 2019)
- [KLLIL][ELHV] (Faridi et al. CIR 2020)

Latter three epitopes reflect partially the amino acid preferences in positions surrounding P₁, which could explain the observed spliced neopeptide generation (see Fig. 6). However, the number of these functionally validated spliced epitopes is too small to draw any statistically meaningful conclusions. For non-spliced peptides, we do believe that we cannot directly compare the P₁ preferences discovered *in vitro* to immunopeptidome data since the P₁ position corresponding to the peptides' C-terminus, which is strongly shaped by HLA-I-peptide binding affinity and can also undergo post-proteasomal trimming by cytosolic or endoplasmic aminopeptidases.

(viii) My last few points have focused on HLA-bound peptides, my reason for this emphasis is that this is probably one of the main physiological impacts of such processing. Are there other roles perhaps for the longer degradation products that may correlate with the measured degradation products? Would such species be detectable by proteomics' for instance? [Perhaps more for speculation than measurement]

Our reply

There are studies focusing on the role of proteasome-generated peptide products in addition to their role in the MHC-I pathway. We mentioned some of them (including our work on OPN) in the revised manuscript. See **line 84**.

So far, nothing has been investigated with regards to spliced peptides. As mentioned above, the community is still debating the existence of peptide splicing. We hope that this seminal study will help to move the controversy forward.

Reviewer #3 (Remarks to the Author):

The manuscript by Soh et al. investigates the peptide splicing capabilities of the proteasome. Specifically, the study focuses on proteomic analysis of the peptides generated by the degradation of 4 intrinsically unstructured proteins. They propose that 21% and 2% of the produced peptides are cis-spliced and trans-spliced peptides. In general, I find much of the data presented to be quite preliminary, often not convincing and open to alternative interpretations.

I give below some suggestions for how the data might be made more convincing:

In general, as the field is so controversial, extra rigorous analysis is required. Making a decisive conclusion from the analysis of only 4 proteins is far fetch. Only 4 proteins were investigated and not 10 as the authors claim in the introduction.

Our reply

We thank the Reviewer for the suggestion. In the revised manuscript, we investigated 27 proteins, of which 15 were identified as efficiently digested *in vitro* by 20S proteasomes and studied for the production of peptide products. This is by far the largest dataset of *in vitro* digested proteins, which, we hope, can be used by the scientific community to better understand both peptide hydrolysis and peptide splicing.

The biochemical results shown in Figures S1 and S2 are extremely sloppy, gels making the entire work unreliable: Fig. S1a - why does a band corresponding to proteasome subunit alpha-4 appear even if the proteasome is absent?

Our reply

We are sorry for the impression that the original gels shown in **Fig. S1-S2** left to the reviewer. As we explain in the answers below, for most of the reviewer's comment on **Fig. S1-S2**, we did not present all the controls we had generated, which led the reviewer to misunderstand some of the outcome of these preliminary experiments. These controls are discussed further below.

However, we did two errors in preparing the original figures, which were the human alpha 4 subunit band (as spotted by the Reviewer), and the mouse IL-alpha of **Fig. S1a**. We triple checked again all WBs and gels shown in the revised **Fig. S1**, and we are confident that any error in the representation of these results has been corrected. For both proteins, we confirmed that they could not be efficiently digested by 20S proteasomes *in vitro*, and thus the original mistake in the presentation of the gels and blots did not impinge upon the rest of the manuscript. We would like to apologize for the mistake done in the submitted manuscript.

In Fig. S2a - why are enolase levels reduced after 24 hours if the proteasome is absent?

Our reply

We agree with the Reviewer that some controls, which were performed but not shown, should have been provided in **Fig. S2a** to avoid misunderstandings. Yeast Enolase 1 was not degraded by 20S proteasomes under our experimental conditions (i.e. absence of SDS) (original **Fig. S1a**). However, Yeast Enolase 1 alone was seemingly degraded in the presence of SDS. Its "self-degradation" profile resulted in bands resemble but not identical to the 20S proteasome subunits. This aspect is clarified in **Fig. 2c for Reviewers**, with a magnification of **Fig. 2a for Reviewers**. In addition, we show the stability of Enolase and proteasome in presence or absence of SDS in **Fig. 2e for Reviewers**.

We considered the interest of both Reviewer 2 and 3 on this matter, and performed and presented new preliminary experiments aimed at clarifying the impact of SDS on protein and proteasome stability (see **Fig. 2 for Reviewers**). However, we think that this interesting question should be address in a study focusing on this aspect of proteasome proteolysis, and thus have we removed any reference to SDS-based protocols in the revised manuscript.

In Fig. S2a - bands of the proteasome are detected even in the absence of the proteasome.

Our reply

We agree with the Reviewer that the bands within the 20-35 KDa present in lane 5 of the original **Fig. S2a** might look similar to the proteasome subunit bands. However, if the Reviewer could zoom in (see **Fig. 2c for Reviewers**), it becomes clear that they are not the same bands. On the contrary, they are products of Enolase (self?) degradation. As explained above, we preferred to remove Enolase and any reference to SDS-based protocols in the revised manuscript, so as not to distract the reader from the take-home message of this study.

In Fig. S2b - bands corresponding to the 20S proteasome are not detected even though the proteasome is in the sample.

Our reply

Even in this case, we agree with the Reviewer that some controls, which were performed but not presented, should have been provided in **Fig. S2b** to avoid misunderstandings. We observed that 20S proteasomes were self-degraded in the presence of SDS after 24 hours of incubation. Hence, the 20S proteasome subunit bands were not observed. By adding SDS, proteasome stability (as observed also for enolase) is altered resulting in (self?) degradation (see **Fig. 2d-e for Reviewers**). This phenomenon has been described by Wilk and Orłowski in 1983 in one of the first descriptions of the proteasome and its functions (doi.org/10.1111/j.1471-4159.1983.tb08056.x). However, the effect of SDS on the proteasome integrity has long been disregarded, as SDS was commonly used for the linearization of proteins to make them vulnerable to 20S proteasome-mediated proteolysis *in vitro*. We preferred to remove any reference to SDS-based protocols in the revised manuscript, so as not to distract the reader from the take-home message of the study.

In Fig. S12 additional bands appear in the sample of the purified proteasome? To which proteins do they correspond? How can we exclude the possibility that they are the ones that cause splicing and not the proteasome?

Our reply

The purity of our proteasome preparation is in line with (or better than) 20S proteasomes that could be purchased from companies or that were used in other studies (references were already reported). The applied protocol for proteasome purification was presented in Schrader et al., Science 2016 and developed by F. Henneberg and A. Chari, who are among the authors of this study. We added this information in the revised manuscript. To test the Reviewer's hypothesis, we performed an in-gel tryptic digestion of the 3 bands detected between 37 and 100 kDa. They are Actin cytoplasmic 2 (41.7 kDa), Adenylyl cyclase-associated protein 1 (52 kDa), and Transitional endoplasmic reticulum (89.3 kDa). We included this information in the revised **Fig. S15, File S9**. None of the detected proteins are proteases and none of them is capable of catalyzing proteolysis or transpeptidation. We included this information in the Methods of the revised manuscript.

In Figure 3e the authors analyze a splice peptide [G][QLGKNEEGAPQE] which actually fits in mass to the non-spliced a-synuclein peptide [QLGKNEEGAPQEG]. The authors should have examined the non-spliced options and compare their scores, especially given the extremely low intensity of the b ions.

Our reply

We are sorry if the reviewer missed the reference to the original **File S7** (we hope that the reviewers had access to all **File S**), which already addressed this important issue in the original manuscript. Indeed, we agree with the Reviewer that the splice peptide [G][QLGKNEEGAPQE] could have been a misassignment of the non-spliced a-synuclein peptide [QLGKNEEGAPQEG]. This hypothesis was tested in the original manuscript, wherein the non-spliced a-synuclein peptide [QLGKNEEGAPQEG] was reported as PCP11 in the **Table S2** and the at page 2 of **File S7**. As shown in the latter, the MS2 spectrum of the non-spliced peptide had a significantly lower ProSight matched Peaks score than the spliced peptide (0.07 and 0.94, respectively).

In the revised manuscript we improved the data display regarding this reviewer's concern. We expanded this important aspect of inSPIRE 1.5 (*i.e.* the isobaric peptide filter) and showed the MS2 spectrum of the isobaric peptide competitor also in **Fig. 3f** and **File S7**. To note, the isobaric peptide competitor was also identified but with an MS2 related to a precursor with a different retention time. The presence of both peptides can also be seen by the double peak in the chromatogram of the 24h digestion in the revised **Fig. 3f**.

In the description of inSPIRE 1.5 in the revised manuscript, we explained that during development of the inSPIRE-aSPIRE workflow, we were also wary of potential misassignment of spliced peptides to scans of isobaric non-spliced peptides. The inSPIRE 1.5 pipeline for spliced peptide identification applies an extra filtering step to remove spliced peptides assigned if there exists a potentially better matched isobaric non-spliced peptide which was not found by the search engine. In this revision, we extended the analysis of spliced peptide and isobaric competitors in the revised **Fig. 3g, Fig. S9, File S7 and S8**. This investigation shows strong evidence of reliable spliced peptide identification. Firstly, 72% of identified spliced peptides are not isobaric to any non-spliced peptide. For the remaining 28% of spliced peptides identified, the results of the comparison (**Fig. S9**) are very strong in favor of the spliced peptide assignment. The PSMs of the spliced peptides show a higher median spectral angle (0.84 compared to 0.19), lower median iRT prediction error (1.73 compared to 12.8), and higher median Spearman correlation (0.89 compared to 0.24) compared to their non-spliced isobaric competitors.

In addition, heavy peptides spiked into the sample, prior to LC-MS/MS should have been used for validation, in all cases.

Our reply

The use of isotopic labeled synthetic peptides spiked into the sample has advantages as well as disadvantages (e.g., required efficiency of isotopic labelling and the resulting financial cost), and it is often not used in immunopeptidomics. As shown in our reply to the previous criticism, our strategy of comparison of experimentally detected PSMs in the protein digestions with synthetic peptides, focusing on both MS2 spectral angles and retention time, is able to discriminate between non-spliced and spliced peptides.

However, in the revised manuscript we expanded on peptide validation. For example, we added 11 synthetic peptide comparisons to cover examples of PSMs derived from several protein digestions (see **File S5-6**) and we extended the comparison of the MS2 features shown in the **File S3-S4** to all peptides identified in the study. We believe that isotopic labelled synthetic peptides spiked into the digestion sample prior to LC-MS/MS would not provide further confidence additional to the validation strategies already presented.

The authors do not explain what is the mechanism for peptide conjugation. How conjugation is kinetically catalyzed in the presence of water.

Our reply

The most well accepted mechanism of PCPS is transpeptidation. This mechanism has been proposed by Van den Eynde lab, and confirmed by us and others. There are many reviews describing the mechanism. We mentioned transpeptidation in the revised Introduction and in the Discussion, and reported opposing views about the likelihood of transpeptidation to occur.

Briefly, proteases are catalysts for the hydrolysis of the scissile bond. Such a process is theoretically reversible. However, the presence of water has made the reverse reaction kinetically unfavorable. The concept of transpeptidation refers to the attack of the unstable substrate-protease acyl intermediate bond by the N-terminal amine group of a peptide rather than by a nucleophilic water molecule. The energy barrier of amidation can be overcome by enzymes. Indeed, there are known naturally occurring proteases that can drive proteolysis and amidation, such as sortases, butelase and asparaginyl endopeptidases. These proteases drive amidation by retaining the substrate in their S'-pockets, thereby blocking water access to the substrate-protease acyl intermediate. This, in turn, allows for the subsequent nucleophilic attack by peptides in close proximity. The 20S proteasome has a barrel-shaped structure that can shield its contents from the surrounding environment as well as retaining peptide products could facilitate the amidation.

This text could be added to the manuscript if the Editor and reviewers think it is not too detailed for the wide readership of Nature Communications.

Careful analysis of the LC-MS/MS spectra in the supplementary files indicates a low correlation between peptides and counterpart synthetic peptides. The majority of the peptide fragments (generated by proteolysis of intact proteins) are not assigned, while synthetic peptides peaks that are assigned are not detected in the peptide fragment spectra. This reduces the reliability of data analysis.

Our reply

We thank the Reviewer for performing a manual evaluation of the MS2 spectral angle of peptides identified in the digestions compared to the cognate synthetic peptides and their potential alternative isobaric non-spliced peptide competitors (now **File S7-8**).

To systematically address this point, first, we computed the MS2 spectral angle and Spearman correlation of all those peptides that were identified in the digestions and compared to the cognate synthetic peptides. The outcome is now reported in **Fig. 3e** to simplify the access to this key information to the readers, whereas the details of each peptide are reported in **File S5-6**. The median spectral angle and spearman correlation between non-spliced peptides that were identified in the digestions and the cognate synthetic peptides is 0.82 and 0.74. The median spectral angle and spearman correlation between spliced peptides that were identified in the digestions and the cognate synthetic peptides is 0.93 and 0.92. This shows a similar trend to our metric calculations on the revised **Fig. 3d-e**). This confirmed the precision of our PSM assignments.

Second, in addition, in the revised manuscript we reported the predicted and experimental MS2 spectra of all peptides identified, both non-spliced and spliced, in **File S3-S4**, and the distribution of spectral angles, Spearman correlation and iRT error between measured and Prosit predicted MS2 spectra in **Fig. 3d**. As apparent in the latter figure panel,

spliced peptides showed better features than non-spliced peptides, confirming the reliability of our identifications. Furthermore, using a 1% global FDR cut-off the spliced peptides had a spectral angle mean higher than the non-spliced peptides assigned in the analysis of immunopeptidomics data by Wilhelm *et al.* (2021), where the search space spanned non-spliced peptides from 85,919 proteins with 'unspecific cleavages'. This confirms that our standard for identification is in the range of, or exceeding studies such as the Wilhelm *et al.*'s study. We added this info in **line 330-331**.

Third, in this revision, we extended the analysis of spliced peptides and their isobaric non-spliced competitors in the revised **Fig. 3g, Fig. S9, File S7 and S8**, as described above.

The theoretical database that this study relies on, is as expected very large, leading to a high false discovery rate.

Our reply

In the original manuscript we addressed this issue for the specific experimental conditions of this study (i.e., the digestion of either a single protein or a single polypeptide) in **Fig. S3** and reported the generation efficacy for each peptide type in **Fig. 4g**. By splitting the results between the main body and the supplementary material in the original submission, we might have confused the reviewer. We would like to apologize for that. In the revised manuscript, we reported the theoretical database size for non-spliced and spliced peptides (**Fig. 4a**) in the same figure where the generation efficacy is shown (**Fig. 4g**) to improve the accessibility of the results to the reader.

Furthermore, while we agree with the reviewer that naïve database expansion can lead to a greatly inflated false discovery rate, in the inSPIRE-aSPIRE workflow we take a number of steps to avoid this. Primarily, we utilize the fact that inflation of the target database also leads to inflation of the decoy database. Hence, the product type of a peptide as a feature in our Percolator rescoring. We can see that this feature penalizes spliced peptide assignments over non-spliced peptide assignments as shown in the weights of Percolator features (this weighting against spliced peptide assignments is now shown in the revised **Fig. S3**). In order to understand the impact of these measures in the context of database size, we further investigated spectral angle distributions of peptides identified at varying q-value/False Discovery thresholds (**File S2**). In particular, we would point out that the mean spectral angle at our chosen FDR cut-off is greater than that of the peptides identified in the immunopeptidomics data analyzed by Wilhelm *et al.* 2021.

In our analysis of the posterior error probabilities from Percolator (where spliced/non-spliced type is also used in estimation) we estimate a false discovery rate of 3.2% on our spliced peptide identification. This estimate compares favorably to the estimated false discovery rate of established methods for spliced peptide identification in polypeptide digestions (estimated to be between 4.2% and 5.3% in our recent paper by Roetschke *et al.*, 2023). This is despite the fact that we are dealing with the larger search space in protein digestions compared to polypeptide digestions.

Given this drawback, which cannot be overlooked, I would have expected that the authors display the data using the more stringent engine (Figure S4, invitroSPI) to present more reliable results.

Our reply

We think that the data reported in the original **Fig. S4** mislead the Reviewer. Indeed, in that figure we reported that InvitroSPI identified fewer non-spliced peptides. However, this is not due to the fact that it is more stringent than inSPIRE in the identification of peptides in protein digestions. On the contrary, InvitroSPI has a lower recall, which leads to the identification of a smaller number of non-spliced peptides (new **Fig. S4**). inSPIRE is a more sensitive tool – using deep learning enabled Prosit prediction for assignments as opposed to relying on Mascot identifications as performed by invitroSPI. The reliability of inSPIRE for non-spliced identification was thoroughly benchmarked in our recent publication where we presented the original tool (Cormican *et al.*, 2022).

We would further argue that inSPIRE is in fact more stringent in its identification of spliced peptides. This can be seen across the 3 proteins analyzed and compared with both tools (**Fig. S4**). The distribution of spectral angles for PSMs of spliced peptides identified with invitroSPI is lower compared to inSPIRE 1.5, indicating a lack of precision/stringency in this experimental setup. The larger peptide yield by applying inSPIRE 1.5 rather than invitroSPI was due to a higher recall of peptides – particularly for spliced peptides - rather than a drop in the precision, because the distribution of the spectral angles between measured and Prosit-predicted MS2 spectra for *cis*- and *trans*-spliced peptides assigned by inSPIRE 1.5 was significantly higher as compared to invitroSPI. While spectral angles of spliced peptides assigned by invitroSPI are lower than for non-spliced peptides, this is entirely different to results obtained with inSPIRE 1.5 (**Fig. S8**) where the distribution of both Prosit spectral angles and retention time prediction error is better for spliced peptides compared to non-spliced peptides.

Moreover, as one of the aims of this study is the development of the inspire software and its advantages. To convince the reader that the performance of the software is superior, more comparisons to invitroSPI or other software should be made.

Our reply

We thank the Reviewer for the criticism. Bearing that in mind, we decided to further improve inSPIRE in the revised manuscript by removing Mascot (which is not freely available) and using the (more performant) open-source tool MSFragger instead. We think that this will improve the accessibility of inSPIRE 1.5 for the readers of the journal and beyond. This significant change in the method led to more stringent identification of spliced peptides by inSPIRE 1.5 compared to the original version (named inSPIRE 1.4) as shown in **Fig. 1 for Reviewers**.

We agree with the reviewer that the introduction of the inSPIRE software for spliced peptide identification should be thoroughly examined. Following the line of thoughts of the Reviewer, in the revised manuscript we developed and released a new version of our software iBench (Cormican et al., 2022) and applied it to benchmarking inSPIRE 1.5 with two simple *de novo* implementations. The performance benchmark is reported in the revised **Fig. S5**, and shows how inSPIRE 1.5 outperforms the *de novo* methods.

Firstly, in terms of the shape of the precision-recall curve, which indicates that inSPIRE is better at distinguishing between correct and incorrect PSMs for *cis*-spliced peptides (**Fig. S5c**). Furthermore, the benchmarking indicates that inSPIRE is more reliable in terms of FDR estimation – the precision at the applied cut-off as represented by the large dot on the PR curve. This indicates that inSPIRE 1.5 - aSPIRE identifications at 1% FDR were more accurate than those of the *de novo* methods.

With regards to a comparison with invitroSPI, as in our reply to the previous point, we would like to point to **Fig. S4**.

It was very hard to follow the flow of the manuscript. The authors use an array of unique terms, that are introduced only shortly to the readers, legends are very simplistic and do not assist the reader in comprehending the fine details of the results. Moreover, the more significant part of this study focuses on comparisons to other studies that the reader has little access to (or requires the reader to read two papers in parallel).

Our reply

We thank the reviewer for this feedback. In the revised manuscript, we spent more words to describe the features of the peptide product database generated by invitroSPI (Roetschke et al., 2023), which we think is what the Reviewer meant by 'the other studies'. We also improved the figure captions by providing more details. In addition, we wrote an entire new section describing the development of the new inSPIRE 1.5 and iBench 2.0, whose old versions were only cited in the original manuscript. This should help the readers to avoid 'jumping' from paper to paper to appreciate the narrative of the present study.

e

Figure 1 for Reviewers. Comparisons of PSMs assigned for inSPIRE 1.4 (i.e., the version used in the original submission) and inSPIRE 1.5 across protein digestions. (a) The mean values of the spectral angle between the Prosit predicted spectrum of spliced and non-spliced peptide assigned by inSPIRE 1.5 and 1.4 for each degraded protein. (b) The mean values of the matched coverage between the Prosit predicted spectrum of spliced and non-spliced peptide assigned by inSPIRE 1.5 and 1.4 for each protein. The higher values of inSPIRE 1.5 for the spliced peptides confirms the higher precision of inSPIRE 1.5 for the identification of spliced peptides than the inSPIRE 1.4, which was used in the original manuscript. (c) The mean values of the matched coverage between the Prosit predicted retention time (fit from iRT to retention time of the raw file) spliced and non-spliced peptide assigned by inSPIRE 1.5 and 1.4 for each degraded protein. The lower values of inSPIRE 1.5 for the spliced peptides confirms the higher precision of inSPIRE 1.5 for the identification of spliced peptides than the inSPIRE 1.4, which was used in the original manuscript. (d) The mean values of the posterior error probabilities of PSMs from which spliced and non-spliced peptides assigned by inSPIRE 1.5 and 1.4 for each degraded protein. The lower values of inSPIRE 1.5 for the spliced peptides confirms the higher precision of inSPIRE 1.5 for the identification of spliced peptides than the inSPIRE 1.4, which was used in the original manuscript.

Figure 2 for Reviewers. Proteolysis of yeast enolase 1 and chicken OVA by human 20S proteasomes in presence of SDS. (a,b) Degradation of yeast enolase 1 (a) and chicken OVA (b) by human 20S standard proteasomes in presence or absence of SDS 0.01% in the buffer for 0 and 24 h. Both substrates and 20S standard proteasomes subunits are visible in the Coomassie blue staining of 4-12% NuPAGE gels. This is the original Fig. S2 in the submitted manuscript. (c) Zoom in of the lanes 5 and 6 reported in (a) to appreciate that the bands in lane 5 are not proteasome subunit bands but rather degradation products perhaps due to self catalysis in presence of SDS. (d) Controls to show Ova and proteasome stability in presence of SDS for 0 h and 24 h. A clear degradation of 20S proteasomes but not OVA appeared after 24h perhaps due to self catalysis. (e) Controls to show Enolase and proteasome stability in presence of SDS for 0 h and 24 h. A clear degradation of both proteins appeared after 24h perhaps due to self catalysis.

Reviewers' Comments:

Reviewer #1:

Remarks to the Author:

Dear Authors

thank you for addressing my comments and improving the manuscript. I just have some minor comments left:

To produce homologous trans-spliced peptides two proteins of the same sequence have to be processed consecutively. This may well be the case in the in-vitro assays, but how likely is this in vivo, where many different proteins may compete for proteasomes? These trans-spliced peptides are already quite rare, and one could expect that they may disappear in a real sample. Could you briefly elaborate on that?

Lines 187, 196, 359, ...: if you say 'significantly' you should indicate a p-value. Otherwise, leave out 'significantly'.

Figure 3: Why do you show statistical significance in Figure 4 and but not in Figure 3?

Figure 4f: Maybe use a log scale for better visibility of the distributions?

Figure 4d: for cis-spliced peptides the protein and synthetic distributions look very similar, but the difference seems to be significant at 0.05. Same for Figure 5b. Wouldn't it make sense to use several stars (e.g. * : < 0.05, **: < 0.01, and *** : < 0.001) to indicate when distribution differences are barely significant or very significant?

Figure 4g: all these stars are a bit confusing. Maybe coloring the stars and the corresponding lines in the same colors would increase the readability of the figure

Figure 5a: not clear what the upper stars refer to (protein digestion vrs protein digestion or all vrs all). Please clarify.

Reviewer #2:

Remarks to the Author:

I thank the authors for their thorough and thoughtful responses and am satisfied with the revised version of the manuscript.

Reviewer #4:

Remarks to the Author:

This paper focuses on analyzing the peptide products of 15 selected proteins processed by the 20S proteasome in vitro. The identification of cis- and trans-spliced peptides suggests that the 20S proteasome can catalyze peptide splicing. The results indicated that the occurrence of peptide splicing is much less than peptide hydrolysis as the majority of peptides identified are non-spliced peptides. While the current study has provided solid evidence on the capability of 20S proteasome to produce splicing peptides in vitro, how the 20S proteasome carries out such function remains unclear. Although the authors suggest that non-spliced and spliced peptides were not randomly localized but skewed

toward peptide product hotspots, their sequence analyses did not reveal any clear correlation of hot spot or IDP regions with non-spliced and spliced forms. Moreover, it is unclear how reproducible and robust is peptide splicing catalyzed by the 20S proteasome in vitro and whether 20S proteasome generates similar splicing products in vivo. Furthermore, whether the identified splicing peptides share any sequence similarities to known HLA-I peptides were not discussed, and the biological significance of the splicing peptides detected in this work is unclear. Since the authors aim to resolve a controversial view in the field, the proteomic results have to be carefully examined and validated. Here are some specific points that need to be addressed:

1. Since the paper relies on unambiguous peptide identification, the result accuracy needs to be carefully examined. It would be helpful if different proteases are used to generate control datasets to examine the data analysis pipeline developed in this study.
2. For any proteomic analysis, biological replicates are required. It is unclear whether any biological replicates at the optimal digestion time point have been included for the 15 selected proteins. If so, how many biological replicates were included and what is the data reproducibility of the 15 proteins when using the same 20S proteasome?
3. It seems that proteasomes from different sources with different purities were used in this study. What are the overlaps of the identified peptides from the same protein substrates when different 20S proteasomes were used? This could be used for result validation.
4. 20S proteasomes have three protease activities: trypsin-like, chymotrypsin-like and PGPH activities. Among the identified non-spliced and spliced peptides, how do their cleavage sites correlate with the known proteasome activities? What is the distribution among non-spliced and spliced peptides?
5. How many protein entries were used for each database searching? If only one protein of interest is included, this could lead to the FDR much higher than originally anticipated. How was FDR assessed for spliced and non-spliced peptides? They need to be considered differently to have better FDR estimation.
6. It seems that Mascot peptide score 20 was used as the cutoff criteria. This threshold is most likely insufficient for unambiguous identification of proteasome processed peptides, especially for long peptides. For unknown sequences, stringent criteria would be required to improve the confidence of peptide identification.
7. As the authors indicated that peptide hydrolysis is the main outcome of the 20S proteasome, what is the peptide sequence overlap between non-spliced and spliced peptides?
8. How do the identified spliced and non-spliced peptides from the 15 proteins compare to published data?
9. It was stated that "MS2 spectra were searched three times with MSFragger v3.7.0 94 using a 5 ppm and 6 ppm tolerance on precursor masses for Q Exactive HF-X and Orbitrap Fusion measurements". It is unclear why 5 ppm and 6 ppm mass errors were used as the different parameters for database searching? Did 1 ppm difference make a discriminative difference?
10. What is the experiment variance for the results listed in Table 1?

REVIEWERS' COMMENTS

Reviewer #4 (Remarks to the Author):

The authors have adequately addressed my concerns. The current content is suited for publication.